



# Modeling the response of Greenland outlet glaciers to global warming using a coupled flow line–plume model

**Johanna Beckmann**[1], **Mahé Perrette**[1], **Sebastian Beyer**[1,2], **Reinhard Calov**[1], **Matteo Willeit**[1], **and Andrey Ganopolski**[1]

[1]Climate Impact Research (PIK), Member of the Leibniz Association, P.O. Box 60 12 03, 14412 Potsdam, Germany
[2]Alfred Wegener Institute, 27570 Bremerhaven, Germany

**Correspondence:** Johanna Beckmannn (beckmann@pik-potsdam.de)

**Abstract.** In recent decades, the Greenland Ice Sheet has experienced an accelerated mass loss, contributing to approximately 25 % of contemporary sea level rise (SLR). This mass loss is caused by increased surface melt over a large area of the ice sheet and by the thinning, retreat and acceleration of numerous Greenland outlet glaciers. The latter is likely connected to enhanced submarine melting that, in turn, can be explained by ocean warming and enhanced subglacial discharge. The mechanisms involved in submarine melting are not yet fully understood and are only simplistically incorporated in some models of the Greenland Ice Sheet. Here, we investigate the response of 12 representative Greenland outlet glaciers to atmospheric and oceanic warming using a coupled line–plume glacier–flow line model resolving one horizontal dimension. The model parameters have been tuned for individual outlet glaciers using present-day observational constraints. We then run the model from present to the year 2100, forcing the model with changes in surface mass balance and surface runoff from simulations with a regional climate model for the RCP8.5 scenario, and applying a linear ocean temperature warming with different rates of changes representing uncertainties in the CMIP5 model experiments for the same climate change scenario. We also use different initial temperature–salinity profiles obtained from direct measurements and from ocean reanalysis data. Using different combinations of submarine melting and calving parameters that reproduce the present-day state of the glaciers, we estimate uncertainties in the contribution to global SLR for individual glaciers. We also perform a sensitivity analysis of the three forcing factors (changes in surface mass balance, ocean temperature and subglacial discharge), which shows that the roles of the different forcing factors are diverse for individual glaciers. We find that changes in ocean temperature and subglacial discharge are of comparable importance for the cumulative contribution of all 12 glaciers to global SLR in the 21st century. The median range of the cumulative contribution to the global SLR for all 12 glaciers is about 18 mm (the glaciers' dynamic response to changes of all three forcing factors). Neglecting changes in ocean temperature and subglacial discharge (which control submarine melt) and investigating the response to changes in surface mass balance only leads to a cumulative contribution of 5 mm SLR. Thus, from the 18 mm we associate roughly 70 % with the glaciers' dynamic response to increased subglacial discharge and ocean temperature and the remaining 30 % (5 mm) to the response to increased surface mass loss. We also find a strong correlation (correlation coefficient 0.74) between present-day grounding line discharge and their future contribution to SLR in 2100. If the contribution of the 12 glaciers is scaled up to the total present-day discharge of Greenland, we estimate the midrange contribution of all Greenland glaciers to 21st-century SLR to be approximately 50 mm. This number adds to SLR derived from a stand-alone ice sheet model (880 mm) that does not resolve outlet glaciers and thus increases SLR by over 50 %. This result confirms earlier studies showing that the response of the outlet glaciers to global warming has to be taken into account to correctly assess the total contribution of Greenland to sea level change.

## 1 Introduction

Sea level rise (SLR) is one of the major threats to humanity under global warming, and approximately one-fourth of the recent SLR can be attributed to the Greenland Ice Sheet (GrIS) (Chen et al., 2017). In the future projections of SLR, the GrIS is not only one of the major potential contributors but also a significant source of uncertainty. Two processes are largely responsible for the GrIS contribution to SLR: (1) dynamic mass loss due to retreat and acceleration of outlet glaciers (40 %) and (2) increased surface melt induced by atmospheric warming (60 %) (Khan et al., 2014; Van Den Broeke et al., 2016). The first process, which is most pronounced for marine-terminating outlet glaciers (Moon et al., 2012), is potentially caused by an increase in submarine melting, which can in turn be attributed to a warming of the subpolar North Atlantic ocean, induced by circulation changes, and increased subglacial discharge (Straneo and Heimbach, 2013). The maximum contribution of increased surface melt is estimated to range between 50 and 130 mm by the year 2100 (Fettweis et al., 2013). Due to the possibility of applying relatively high-resolution regional climate models, confidence in this estimate has increased in recent years (van den Broeke et al., 2017). The contribution of the dynamic mass loss, however, remains highly uncertain because processes related to the response of marine-terminating Greenland glaciers are still not properly represented in the contemporary GrIS models (Straneo and Heimbach, 2013; Khan et al., 2014; Goelzer et al., 2017).

The principal objective of this paper is to quantify the response of marine-terminating outlet glaciers to future submarine melting and to analyze whether the impacts of ice–ocean interaction on SLR are comparable to long-term changes in surface mass balance (SMB).

In order to assess Greenland's contribution to future sea level rise, several different model strategies have been proposed. The most common method is to use three-dimensional ice sheet models, tuned to present-day conditions, and apply future climate change projections based on global or regional climate models. However, such models still have relatively coarse spatial resolution and cannot properly resolve most of the outlet glaciers that terminate in Greenland's fjords. Peano et al. (2017) investigated the five biggest ice streams and outlet glaciers in Greenland with a 3-D ice sheet model at a resolution of 5 km (Seddik et al., 2012) and Gillet-Chaulet et al. (2012) included improved model physics by using a full-Stokes approach and refined resolution over fast-flow regions with adaptive mesh techniques. Their setup, however, did not yet allow simulation of glacier retreat. Most of the ice sheet simulations also do not describe the interaction between glaciers and the ocean explicitly, but in some cases, for instance in Fürst et al. (2015), ocean melting is parameterized indirectly by increasing the basal sliding factor as ocean temperature increases. For the RCP scenario 8.5, they calculated a SLR between 155 and 166 mm at the year 2100 for

the entire ice sheet applying atmospheric and oceanic forcing. For regional settings, 3-D models with a simple ocean melting parameterization were applied to study the historical (last 20–30 years) retreat of Jakobshavn Isbrae (Muresan and Khan, 2016; Bondzio et al., 2017). A more advanced treatment of submarine melt rate was used by Vallot et al. (2018). They coupled a plume model based on the Navier–Stokes equations with a full-Stokes ice sheet model. With this off-line coupling, glacier dynamics for one melt season were simulated for Kronebreen in Svalbard.

Another method, followed by Nick et al. (2013), is to simulate single-outlet glaciers individually using a one-dimensional (1-D) flow line model. Nick et al. (2013) performed simulations for four outlet glaciers that collectively drain about 22 % of the total solid ice discharge of the Greenland Ice Sheet. Assuming proportionality between the future contribution to SLR and present-day ice discharge, Nick et al. (2013) scaled up results obtained from four glaciers to the total estimate of all Greenland outlet glaciers, which resulted in a range between 65 and 183 mm by the year 2100. Taking this one step further, Goelzer et al. (2013) used the results from Nick et al. (2013) in a 3-D coarse-resolution ice sheet model. They applied the 1-D glacier thinning and grounding line retreat scenarios as an external, pre-calculated forcing in the grid cells at the ice sheet boundary. Since only four glaciers had been simulated in the 1-D model, they mapped the forcing from the original glaciers onto all Greenland's other marine-terminating outlet glaciers with a nearest-neighbor approach. The incorporation added only 8 to 18 mm SLR on top of the stand-alone 3-D ice sheet model simulation. Goelzer et al. (2013) argued that the rather small contribution compared to Nick et al. (2013) is caused by the full retreat of the small marine-terminating glaciers in the 3-D ice simulations within a short timescale. When fully retreated, they do not experience any ice–ocean interaction any more. This loss of ice–ocean interaction, however, is neglected by the upscaling-method from Nick et al. (2013) and therefore leads to higher numbers of total SLR.

Since we are especially interested in the impacts of ice–ocean interactions on glacier dynamics and want to investigate numerous glaciers, we followed an approach similar to that of Nick et al. (2013) but for different glacier types and with one notable improvement: for calculations of the vertically distributed submarine melt, we use a turbulent plume parameterization following Jenkins (2011). According to this parameterization, the submarine melt rate depends not only on ambient water temperature in fjords but also on seasonally varying subglacial discharge, shape and angle of the glacier tongue. The first idealized simulations of a coupled flow line–plume model were carried out by Amundson and Carroll (2018) by using the maximum melt rate as a frontal ablation factor to account for undercutting plus calving of tidewater glaciers, demonstrating the potential impact of the subglacial discharge on glacier dynamics. While their study emphasizes the importance of subglacial discharge, their model

setup does not allow for the evolution of floating tongues. Thus, we follow a different approach (Sect. 2) to simulate the glacier profile more realistically.

We perform simulations for 12 representative Greenland glaciers (compared to four in Nick et al., 2013). This enabled us to test the assumption used in Nick et al. (2013) that the contribution of individual Greenland outlet glaciers to SLR is proportional to their present-day discharge and therefore the total contribution of Greenland outlet glaciers can be obtained by scaling up contribution of individual glaciers proportionally to the entire present-day discharge of all outlet glaciers. In particular we derived a proportional factor between present-day grounding line discharge and future SLR using results of simulations for all 12 glaciers. We also estimated the uncertainties in the contribution of Greenland glaciers to SLR resulting from uncertainties in calving and ocean melt parameters and ocean warming.

The paper is structured as follows. First, we describe the coupled flow line–plume model and then how the input data were preprocessed together with the experimental setting and climate change scenarios. Finally, we present the results of our model simulations for present-day and future scenarios.

## 2 The coupled flow line–plume model

Most of Greenland's outlet glaciers terminate in fjords that are connected to the ocean. Inside these fjords, observations of upwelling plumes along the edges of glaciers have drawn attention to the potential importance of submarine melting. Consequently, considerable efforts in the modeling of submarine melt rate have been undertaken by using high-resolution 3-D and 2-D ocean general circulation models that are tuned to or parameterized after the buoyant-plume theory (Sciascia et al., 2013; Xu et al., 2013; Slater et al., 2015, 2017; Cowton et al., 2015; Carroll et al., 2015). However, such models are computationally too expensive and therefore impractical for simulating the response of the entire GrIS to climate change on centennial timescales. At the same time, recent studies demonstrate that the simple line plume model by Jenkins (2011) is an adequate tool to simulate plume behavior (Jackson et al., 2017) and to determine submarine melt rates for marine-terminated glaciers (Beckmann et al., 2018). Since the plume model is significantly less computationally expensive than 3-D ocean models, it represents an alternative approach to introduce ice–ocean interaction into the GrIS model and still maintain the model's ability to perform a large set of centennial-scale experiments. Simulating the glacier dynamics with 3-D ice sheet models requires very high spatial resolution ($\ll 1\,\mathrm{km}$) resulting in high computational cost (e.g., Aschwanden et al., 2016) and so far they cannot be used for centennial timescales. To reduce the computational cost we instead use a depth- and width-integrated one-dimensional ice flow model (Enderlin and Howat, 2013; Nick et al., 2013) coupled to a line plume model (Beckmann

et al., 2018). Unlike Amundson and Carroll (2018), who used the maximum melt rate as a frontal ablation factor for tidewater glaciers, we take into account the entire profile along the submerged part of the outlet glacier to calculate the submarine melt rate with the plume model (Fig. 1).

### 2.1 Glacier model

The governing equations of the 1-D glacier model (Fig. 1) include mass conservation:

$$\frac{\partial H}{\partial t} = -\frac{1}{W}\frac{\partial (UHW)}{\partial x} + \dot{B} - \dot{M}, \tag{1}$$

where $H$ is ice thickness, $t$ is time, $U$ is the vertically averaged horizontal ice velocity, $W$ is the width and $x$ is the distance from the ice divide along the central flow line. $\dot{B}$ and $\dot{M}$ are the surface mass balance and the submarine melt rate of the glacier.

The conservation of momentum involves a balance between longitudinal stress, basal shear stress and lateral stress on the one hand, and driving stress on the other:

$$2\frac{\partial}{\partial x}\left(H\nu\frac{\partial U}{\partial x}\right) - A_{\mathrm{s}}\left[(H - \frac{\rho_{\mathrm{w}}}{\rho_{\mathrm{i}}}h_{\mathrm{b}})U\right]^{q}$$
$$- \frac{2H}{W}\left(\frac{5U}{EAWW_{\mathrm{s}}}\right)^{\frac{1}{3}} = \rho_{\mathrm{i}}gH\frac{\partial h_{\mathrm{s}}}{\partial x}, \tag{2}$$

where $h_{\mathrm{s}}$ denotes the ice surface height, $h_{\mathrm{b}}$ the depth of glacier below sea level, and $\rho_{\mathrm{i}}$ and $\rho_{\mathrm{w}}$ the ice and seawater density, respectively. The sliding law follows Nick et al. (2010) with the basal sliding coefficient $A_{\mathrm{s}}$ and the velocity exponent $q$, and the lateral stress involves a nondimensional width-scaling parameter $W_{\mathrm{s}}$. The lateral stress term likewise used by Nick et al. (2013), Enderlin and Howat (2013), and Schoof et al. (2017), and originally derived by Van Der Veen and Whillans (1996), is necessary to account for lateral resistance in fast-flowing, laterally confined glaciers typical for Greenland. Finally, the rate factor $A$ and the enhancement factor $E$ determine the viscosity $\nu$

$$\nu = (EA)^{\frac{1}{3}}\left|\frac{\partial U}{\partial x}\right|^{-\frac{2}{3}}. \tag{3}$$

Calving occurs when surface crevasses propagate down to the water level (Nick et al., 2013). Crevasse depth $d_{\mathrm{s}}$ is calculated from the resistive stress $R_{xx} = 2\left(\frac{1}{A}\frac{\partial U}{\partial x}\right)^{1/3}$, as ice stretches, and can be enhanced by freshwater depth $d_{\mathrm{w}}$:

$$d_{\mathrm{s}} = \frac{R_{xx}}{\rho_{\mathrm{i}}g} + d_{\mathrm{w}}\frac{\rho_{0}}{\rho_{\mathrm{i}}}, \tag{4}$$

where $\rho_{0}$ is the freshwater density. The glacier front continuously advances over time, as the accumulated flux leaving the last grid cell is recorded and the calving front is advanced whenever the accumulated volume reaches the volume of a

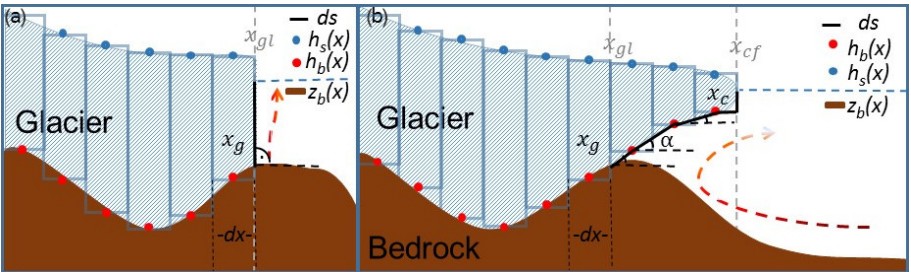

**Figure 1.** Visualization of the 1-D glacier model with the staggered grid for **(a)** a tidewater glacier and **(b)** a glacier with a floating tongue. Red dots indicate where the depth of glacier base $h_b$ is defined and blue dots where surface elevation $h_s$ of the glacier is defined. They are calculated at $dx/2$ – the half width of each grid cell. Last grounded cell has the coordinate $x_g$ and last floating cell has the coordinate $x_c$. The grounding line $x_{gl}$ is determined at the border of the last grounded cell, where the flotation criterion is not yet achieved. After the grounding line, the calculation of submarine melt along the distance $ds$ (thick, black line) is performed with the line plume model. For a floating tongue **(b)** every grid cell may have a different angle for the slope of glacial base while for a tidewater glacier **(a)** the angle is set to 90°. The bedrock elevation $z_b$ (brown, thick line) is equal to $h_b$ for the grounded part and is deeper for the floating part of the glacier.

grid cell (assuming same thickness). Glacier front advance and calving are the two competing processes that determine the calving front position.

The boundary condition at the top of the glacier catchment is $U(x = 0) = 0$. At the calving front $x_{cf}$ (Fig. 1b), the balancing of the longitudinal stress with the hydrostatic seawater pressure and the incorporation of the flow law of ice yields longitudinal stretching

$$\left.\frac{\partial U}{\partial x}\right|_{x=x_{cf}} = EA\left[\frac{\rho_i g H}{4}\left(1 - \frac{\rho_i}{\rho_w}\right)\right]^3. \tag{5}$$

The model employs a stretched horizontal grid with a horizontal resolution of 100 m, where velocity is calculated at midpoints. At each time step of 3.65 d, the grid is stretched to keep track of the grounding line position, which is determined by the flotation criterion

$$H_{float} \leq |z_b|\frac{\rho_w}{\rho_i}, \tag{6}$$

where $z_b$ is the bedrock depth. Glacier thickness $H$ and bedrock depth $z_b$ of each cell interface are determined by linear interpolation between the cell-centered values. Grid stretching is performed so that there is always a cell edge at the interpolated grounding line position ($x_{gl}$ in Fig. 1). The new calving front position is determined so that the total glacier volume is not modified by interpolation. For every new point in the interior, model variables are interpolated from the previous grid. The first grid point at the ice divide remains unchanged. If points on the new ice grid lie outside the ice domain of the previous ice grid, as is typically the case for the last ice cell before the calving front, ice thickness from the last grid cell is extended to the new ice cell (the calving front advances).

The code is written is FORTRAN, following the numerical procedure of Enderlin et al. (2013). The main differences compared to their original MATLAB code (available at

https://sites.google.com/site/ellynenderlin/research, last access: 8 August 2019) are that we include a subgrid-scale treatment of the calving front boundary and an improved treatment of the submarine melting.

## 2.2 Plume model

The plume model equations account for a uniformly distributed subglacial discharge along the grounding line of a glacier, and contain the evolution of the plume thickness $D$, velocity $V$, temperature $T$ and salinity $S$ along the direction of the plume.

$$q_s' = \dot{e} + \dot{m} \tag{7}$$

$$(q_s V)' = D\frac{\Delta\rho}{\rho_0} g \sin(\alpha) - C_d V^2 \tag{8}$$

$$(q_s T)' = \dot{e}T_a + \dot{m}T_b - C_d^{\frac{1}{2}} V \Gamma_T (T - T_b) \tag{9}$$

$$(q_s S)' = \dot{e}S_a + \dot{m}S_b - C_d^{\frac{1}{2}} V \Gamma_S (S - S_b) \tag{10}$$

The volume flux of the plume $q_s = DU$ (expressed per unit length in the lateral direction, i.e., $m^2 s^{-1}$) is described by Eq. (7). It can increase by the entrainment of ambient seawater $\dot{e}$ and by melting $\dot{m}$ of ice from the glacier front. Equation (8) describes the balance between buoyancy flux and the drag $C_d U^2$ of the glacier front. The buoyancy flux is proportional to the relative density contrast $\frac{\Delta\rho}{\rho_0}$ between plume water and ambient water in the fjord (subscript a). This density contrast is linearly parameterized as $\beta_S(S_a - S) - \beta_T(T_a - T)$. The drag also results in a turbulent boundary layer (subscript b) at the ice–water interface, where melting occurs, and heat and salt is exchanged by (turbulent) conduction–diffusion. The temperature $T$ and salinity $S$ of the plume (Eqs. 9, 10) are determined by the entrainment of ambient water and the addition of meltwater, as well as by conduction fluxes at the ice–water interface (i.e., between boundary layer and plume). The entrainment rate is calculated as $\dot{e} = E_0 U \sin(\alpha)$, pro-

portional to plume velocity and glacier slope, with the entrainment coefficient $E_0$.

The submarine melt rate along the path of the plume $\dot{m}$ is determined by solving the equations of heat and salt conservation at the ice–water interface:

$$\dot{m}L + \dot{m}c_i(T_b - T_i) = c C_d^{\frac{1}{2}} V \Gamma_T (T - T_b), \qquad (11)$$

$$\dot{m}(S_b - S_i) = C_d^{\frac{1}{2}} V \Gamma_S (S - S_b), \qquad (12)$$

where $T_i$ and $S_i$, $c_i$ are the temperature, salinity and specific heat capacity of the ice and $c$ the specific heat density for seawater. At the ice water interface the freezing temperature $T_b$ is approximated as a linear function of depth $Z$ ($Z < 0$) and salinity of the boundary layer $S_b$:

$$T_b = \lambda_1 S_b + \lambda_2 + \lambda_3 Z, \qquad (13)$$

with $Z = Z_0 + x \cdot \sin(\alpha)$, where $Z_0$ is the depth (negative) at the grounding line ($x = 0$). The algorithm for solving the set of equations and a list of all parameter values is provided in Beckmann et al. (2018).

The plume is a 1-D model and therefore can neither simulate variability across the calving front (Fried et al., 2015) nor account for fjord-wide circulation (Slater and Straneo, 2018) across and outside plumes. However, the width-averaged melt rates – as required for the 1-D glacier model – can be simulated with the 1-D plume model (Beckmann et al., 2018). We set the entrainment parameter $E$ to 0.036, as suggested by Beckmann et al. (2018). Since the plume model in some cases underestimates and in others overestimates submarine melt rates (Beckmann et al., 2018), we also scale the simulated melt rate profile by a constant factor $\beta$, which we treat as an additional tuning parameter within the range 0.3–3 possibly different for each glacier (see Sect. 4.1). The plume model employs a fine spatial resolution of about 1 m.

## 2.3 Coupling between glacier and plume model

As mentioned in the beginning of this section, the submarine melt rate profile is calculated along the entire glacier profile in contact with the ocean (tidewater glacier or with floating tongue, Fig. 1a and b) and converted into a thickness loss for each of these glacier cells.

Submarine melting volume flux is calculated for each cell and is applied as a vertical thinning rate on the floating tongue ($x_{g+1}...x_c$), or on the last grounded cell ($x_g$) in the case of tidewater glaciers (no floating tongue). The melt rate $\dot{m}$ is integrated from the grounding line (position $x_{gl}$) along the bottom face of the floating tongue (if any), and along the calving face (position $x_{cf}$) up to sea level (Fig. 1), or to the top height of the risen plume (which can stop below sea level). The total submarine melt rate over the glacier tongue (if any)

for one outlet glacier is given by

$$M = \int \dot{m}(s)\,\mathrm{d}s = \int_{x_{gl}}^{x_{cf}} \dot{m}(h_b(x)) \cdot (\cos\alpha)^{-1}\mathrm{d}x$$

$$+ \int_{h_b(x_{cf})}^{0} \dot{m}(z)\mathrm{d}z, \qquad (14)$$

where $s$ is the distance coordinate along the tongue bottom and the vertical calving face, $h_b$ denotes bottom ice elevation and $\cos\alpha$ is the variable tongue slope (calculated from the relation $\tan\alpha = \frac{\partial h_b}{\partial x}$). The integral is partitioned over various glacier cells (or only one cell ($x_g$) in the case of a tidewater glacier, where the first integral term is zero since $x_{gl} = x_{cf}$). This total submarine melt rate, on a cell-by-cell basis, is substituted in (the discrete form) of Eq. (1):

$$M_i = \int_{x_{i-\frac{1}{2}}}^{x_{i+\frac{1}{2}}} \dot{m}(s)\,\mathrm{d}s + \varepsilon_i \int_{h_b(x_{cf})}^{0} \dot{m}(z)\mathrm{d}z, \qquad (15)$$

where $\varepsilon_i$ is 1 if $i$ represents the last ice cell ($x_i = x_c$), or 0 otherwise. The submarine melt rate $\dot{M}$ per units of length for each glacier cell ($\mathrm{d}x$) in Eq. (1) is

$$\dot{M}_i = \frac{M_i}{\mathrm{d}x}. \qquad (16)$$

If there is no floating tongue, submarine melting is applied to the last grounded cell, otherwise it is applied starting from the first floating cell.

Thus the submarine melt rate reduces the thickness of the glacier cell. A reduced thickness at the first floating cell or last grounded cell leads to grounding line retreat since the grounding line position is determined by interpolation of the ice thickness above flotation at each time step. Calving front retreat can be reached by melting/thinning the last floating ice cell completely or by calving, which increases with thinning.

Since the plume model does not allow for negative values of $\alpha$, its minimum value is set to $10^{-6}$. If the plume already ceases before reaching the calving front $x_{cf}$, we numerically introduce a minimal background melting determined by the last melt rate value before the plume ceased. At the calving front we calculate a second plume that starts at $h_b(x_{cf})$ with the initial minimum default discharge value of $10^{-6}\,\mathrm{m^3\,s^{-1}}$ to assure a background frontal melting.

Subglacial discharge $Q$ for each glacier was computed off-line from the output of simulations with the ice sheet model SICOPOLIS (Calov et al., 2018), which includes explicit treatment of basal hydrology (Sect. 3.3). It is applied to the line plume (Fig. 2), assuming a uniform distribution of subglacial discharge along the width of the grounding line: $q_s = Q(W)^{-1}$. It is assumed that plume properties (velocity,

temperature, salinity and thickness) adapt instantaneously to changes in the glacier's shape, subglacial discharge, temperature and salinity profiles of ambient water. The glacier and plume model exchange information at every time step of the glacier model (Fig. 2).

## 3   Model input

To simulate the response of the glacier–plume model to future global warming we considered the potential changes of surface mass balance (SMB) and submarine melting. To this end, for each glacier, we derived data sets for three forcing factors from the year 2000 till 2100: spatially distributed SMB (Sect. 3.3), subglacial discharge (Sect. 3.3) and fjord water temperatures (Sect. 3.4). For changes in SMB we used anomalies from the simulation with the regional climate model MAR, forced by global general circulation model (GCM) MIROC5 for the RCP8.5 scenario. In our previous study (Calov et al., 2018) we used the same SMB changes to force the 3-D ice sheet model SICOPOLIS. Now we use results of this simulation to compute the subglacial discharge for each glacier from simulated surface runoff. Changes in ocean temperature were included by applying a linear warming trend, derived from several different CMIP5 models. On every time step the three forcing factors were provided as data input and forced the glacier–plume model (Fig. 2). While for each glacier the future evolution of the subglacial discharge and ocean temperature were firmly prescribed in the data sets, the SMB input was interactively corrected for the surface elevation feedback and thus considered the glacier surface height on each time step. The upcoming subsections describe the choice of glaciers, how the geometry for the 1-D model was derived and how the corresponding forcing factors were determined and applied.

### 3.1   The choice of glaciers

In this study we modeled 12 well-studied Greenland outlet glaciers of different sizes and located in different regions of Greenland (Fig. 3). One criterion for this selection was that the glaciers should represent different types of ice flows and different environmental conditions. We also included small marine-terminating glaciers to assure a more realistic up-scaling (Goelzer et al., 2013). In addition, for most of the chosen glaciers, Enderlin and Howat (2013) estimated submarine melting-to-calving ratios (grounding line mass flux lost by submarine melting divided by mass loss of calving), which were used as an additional constraint on the choice of modeling parameters.

### 3.2   Glacier geometry

For each individual glacier, bedrock elevation and width were determined by analyzing cross sections taken at regular intervals along the glacier flow, generally covering a large portion of the glacier catchment area (Perrette et al., in prep). In each cross section, the procedure comes down to calculating a flux-weighted average for bedrock elevation, ice velocity $U$, and thickness $H$ and choosing the glacier width $W$ such that the flux $F$ through the cross section is conserved, i.e., $W = F/(UH)$. We used the BedMachine v2 data for bedrock topography (Morlighem et al., 2014). Fjord bathymetry was extended manually by considering available data (Mortensen et al., 2013; Schaffer et al., 2016; Dowdeswell et al., 2010; Syvitski et al., 1996; Rignot et al., 2016). For ice velocity we use data from Rignot and Mouginot (2012). The resulting glacier profiles are depicted in Fig. 6.

### 3.3   Subglacial discharge and glacier surface mass balance

To force the plume model, we used monthly averaged subglacial discharge. Subglacial discharge represents the sum of basal melt (melt under the grounded ice sheet), water drainage from the temperate layer and surface runoff. The former two sources were computed directly by the ice sheet model SICOPOLIS (Calov et al., 2018). In reality surface runoff can travel along the ice surface until it either reaches an existing connection to the bedrock (e.g., crack) or accumulates in a supraglacial lake that eventually drains, making a new connection. However, these processes are complex and still poorly understood. This is why in the relatively coarse-resolution (5 km) ice sheet model (Calov et al., 2018), these small-scale processes were neglected and it was assumed that runoff penetrates directly down to the bedrock. The surface runoff and SMB anomalies for future scenarios are taken from experiments with the regional climate model MAR (Fettweis et al., 2013) and corrected for the future surface elevation change (Calov et al., 2018). The entire basal water flux (runoff, basal melt and water from the temperate layer) was routed by the hydraulic potential using a multi-flow direction flux-routing algorithm, as described in Calov et al. (2018). All water transfer was assumed to be instantaneous. Water that passes through the boundary of the prescribed SICOPOLIS ice mask was assigned to the closest glacier within a maximum distance of 50 km. This maximum distance is necessary in areas where only a few named glacier positions are available (mostly in the south of Greenland) and the distance between glaciers is large. For most of the coastline, especially in the area of our selected glaciers, this distance had no effect on the results. We did not separately study the uncertainty in subglacial discharge related to this approach, but rather accounted for this uncertainty implicitly through the uncertainty of the scaling coefficient $\beta$ for the submarine melt rate (see Sect. 4.1).

In our future scenarios, when simulating subglacial discharge, we accounted for changes in surface runoff, basal melt and ice sheet elevation. The routing end points, which determine the amount of subglacial discharge for each

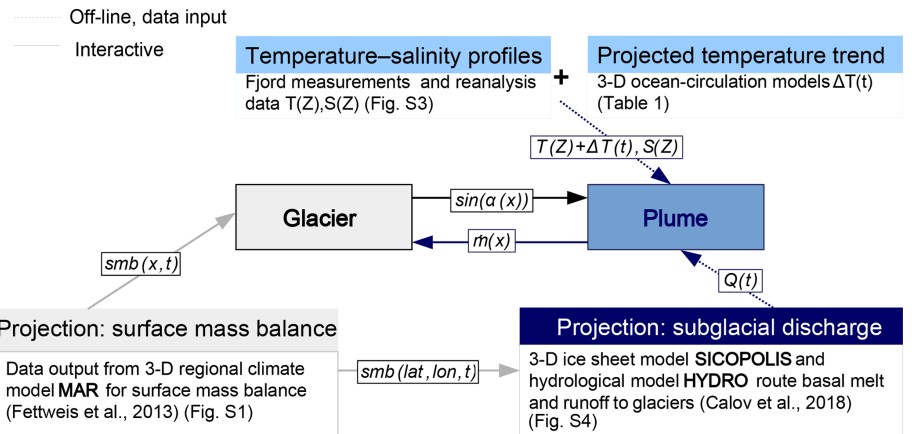

**Figure 2.** Visualization of the experimental setup. In the center the coupled glacier–plume model exchange information on the glacier geometry ($\sin(\alpha)$) and the calculated submarine melt rate ($\dot{m}$) on every time step for every glacier grid point $x$. To force the coupled model for global warming (RCP8.5), changes in SMB ($smb(x,t)$), ocean temperature ($T(Z) + \Delta T(t)$) and subglacial discharge ($Q(t)$) are considered via data input. While the SMB changes act on the glacier part, the changes in subglacial discharge and ocean temperature are used to recalculate the submarine melt rate by the plume part of the model. The future evolution of subglacial discharge and ocean temperature is prescribed firmly in the data sets (off-line) that force the plume part, whereas changes in SMB are corrected for the surface elevation feedback and therefore regard changes in the ice surface height interactively. SMB is derived from MAR data (Fettweis et al., 2013), also used to derive the subglacial discharge for each glacier by Calov et al. (2018).

glacier, however, were set constant to the present-day position of the ice sheet margin. For neighboring glaciers with a competing catchment area, a strong ice sheet retreat may strongly affect the distribution of the subglacial discharge between those glaciers (Lindbäck et al., 2015). This effect was not included in this study.

In this study, we used a single scenario for future surface runoff and SMB change, namely, a simulation with the regional model MAR nested in the global GCM MIROC5 model forced by the RCP8.5 scenario. Among the CMIP5 models, MIROC5 simulates climate change, which led to a medium contribution of GrIS to future SLR (Calov et al., 2018). To correct for the general circulation model's biases in surface runoff and SMB, we used an anomalous approach by adding future anomalies in surface runoff and SMB simulated by MAR nested into the MIROC5 model to the reference climatology (reference period 1961–1990) simulated by MAR forced by ERA-Interim reanalysis data. We also corrected model surface runoff and SMB for changes in surface elevation by applying the gradient method of Helsen et al. (2012) as described in Calov et al. (2018). The surface runoff $R$ over the ice sheet (SICOPOLIS) is determined as

$$
\begin{aligned}
R(x, y, t_{\text{monthly}}) = {} & R_{\text{MAR(REAN)}}^{\text{Clim 1961–1990}}(x, y) \\
& + (R_{\text{MAR(MIROC)}}(x, y, t_{\text{monthly}}) \\
& - R_{\text{MAR(MIROC)}}^{\text{Clim 1961–1990}}(x, y)) \\
& + \left(\frac{\partial R}{\partial z}\right)_{\text{MAR(MIROC)}}(x, y, t) \\
& \Delta h_{\text{s}}(x, y, t_{\text{monthly}}),
\end{aligned} \tag{17}
$$

where the runoff $R(x, y, t)$ on every grid cell $(x, y)$ at any time $t$ on a monthly time step was calculated by the climatological mean from 1961 to 1990 of MAR (forced by reanalysis data) $R_{\text{MAR(rean)}}^{\text{Clim 1961–1990}}(x, y)$ plus the anomaly of the runoff relative to the climatological mean for the same period of time obtained by MAR forced with MIROC5 $(R_{\text{MAR(CMIP5)}}(x, y, t) - R_{\text{MAR(CMIP5)}}^{\text{Clim 1961–1990}}(x, y))$. For ice surface evolving in time $\Delta h_{\text{s}}(x, y, t) = h_{\text{s}}^{\text{obs}}(x, y) - h_{\text{s}}(x, y, t)$, the vertical gradient $\left(\frac{\partial R}{\partial z}\right)_{\text{MAR(MIROC)}}(x, y, t)$ determined for every time step was additionally applied to account for the increase in surface runoff. The observed surface elevation $h_{\text{s}}^{\text{obs}}$ of the ice sheet was taken from Bamber et al. (2013). Negative runoff values were set to zero. The correction of runoff for elevation change can be important in some case since, as it was shown in Amundson and Carroll (2018) for tidewater glaciers, large and rapid changes in glacier volume can lead to a high increase in runoff due to surface lowering.

For the present-day condition, the SMB for the glaciers was calculated from relaxation to observed surface elevation $h_{\text{s}}^{\text{obs}}$, with a different relaxation timescale $\tau$ for each glacier (see Sect. 4.1):

$$
\dot{B} = \frac{h_{\text{s}}^{\text{obs}} - h_{\text{s}}}{\tau} \quad \text{in m yr}^{-1}. \tag{18}
$$

With the latter equation we calculated the present-day SMB during the spin-up experiment, similarly to Calov et al. (2018). For future scenarios, we added the anomaly of the SMB (relative to the year 2000) to the present-day SMB. The anomaly for each grid cell of the glacier was computed from interpolation of the MAR anomaly of the centerline of the

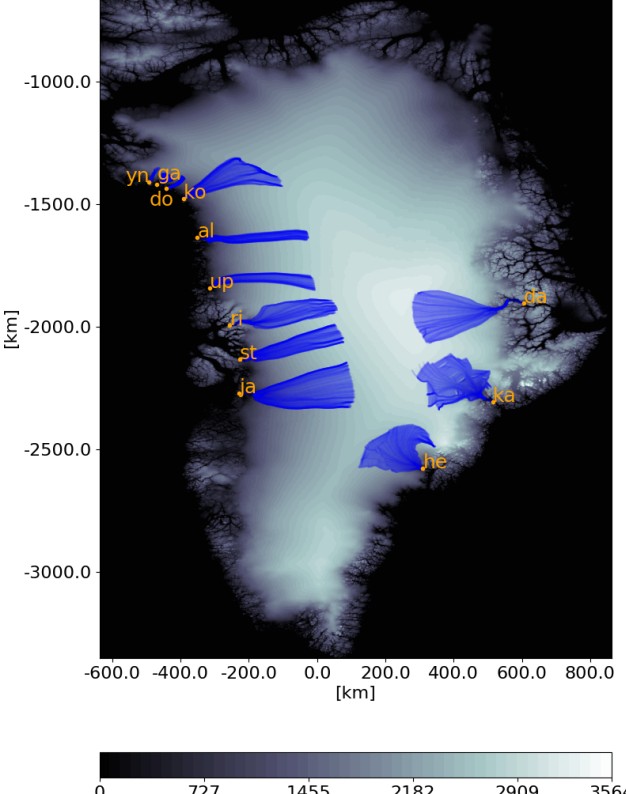

**Figure 3.** Terminus location (orange dot) with the catchment area (blue) of the 12 investigated glaciers: Alison Glacier (al), Daugaard-Jensen Glacier (da), Docker Smith Glacier (do), Gade (ga) Helheim Glacier (he), Jakobshavn Isbrae (ja), Kangerlussuaq Glacier (ka), King Oscar Glacier (ko), Rink Isbrae (ri), Store Glacier (st), Upernavik Isstrom N (up), Yngvar Nielsen Glacier (yn).

individual glacier and additionally corrected for the glacier elevation change similarly to the surface runoff (Eq. 17), but for the SMB calculation, $\Delta h_s$ is the glacier elevation change compared to present day, assuming that the derived glacier shape from the present-day data set is for the year 2000.

$$\text{SMB}(x, t) = \text{SMB}^{2000}_{\text{MAR(MIROC)}}(x)$$
$$+ (\text{SMB}_{\text{MAR(MIROC)}}(x, t) - \text{SMB}^{2000}_{\text{MAR(MIROC)}}(x))$$
$$+ \left(\frac{\partial \text{SMB}}{\partial z}\right)_{\text{MAR(MIROC)}} (x, t) \quad \Delta h_s(x, t) \quad (19)$$

The time series of cumulative SMB (without surface correction) and the annual subglacial discharge for each glacier are shown in the Supplement (Figs. S1 and S2).

### 3.4 Fjord temperature and salinity profiles: CTD measurement and ocean reanalysis data

Determining vertical temperature and salinity profiles in Greenland fjords, which are the input for the plume model, is a challenging task. Measurements inside Greenland fjords

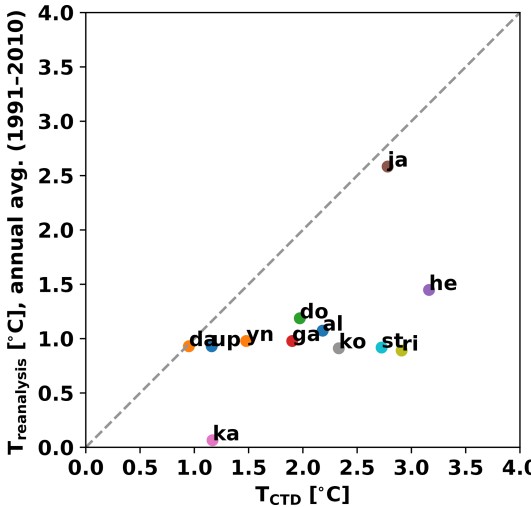

**Figure 4.** Temperature at the grounding line depth of CTD measurements closest to glacier front, inside the fjords (x axis) and temperatures reconstructed from reanalysis data (y axis) from the nearest possible grid cell for all 12 glaciers: Alison (al), Daugaard-Jensen (da), Docker Smith (do), Gade (ga), Helheim (he), Jakobshavn Isbrae (ji), Kangerlussuaq (ka), King Oscar (ko), Rink Isbrae (ri), Store (st), Upernavik Isstrom N (up), Yngvar Nielsen (yn).

are rare and do not cover all of them. For some fjords, several conductivity–temperature–depth (CTD) measurements exist, but they are mostly infrequent and often not performed close enough to the calving front. It is also important to note that $T-S$ profiles obtained from CTD measurements have to be treated with caution because they represent only a "time shot" of fjord properties, which vary significantly in time (Jackson et al., 2014).

However, the question arises of how to treat fjords where no CTD measurements are available. A possible solution is to use ocean reanalysis data. Here we use the TOPAZ Arctic Ocean Reanalysis data (http://marine.copernicus.eu/services-portfolio/access-to-products/?option=com_csw&view=details&product_id=ARCTIC_REANALYSIS_PHYS_002_003, last access: 8 August 2019) (Xie et al., 2017) and compare them with existing CTD measurements. Note that for all 12 glaciers used in this study the CTD measurements from the adjacent fjord are available and we use them throughout our experiments as the preferred temperature–salinity profile ($T-S$ profile). To make assumptions on potential impacts of the differences between reanalysis and CTD profiles on the glacier response to climate change we investigate both types of ocean data (reanalysis and measurements). Figure 4 illustrates the strong bias towards colder temperatures for the reanalysis data set. The detailed description on how the temperature profiles were derived from the reanalysis data and illustrations of both temperatures profiles that forced the glacier–plume model can be found in the Supplement.

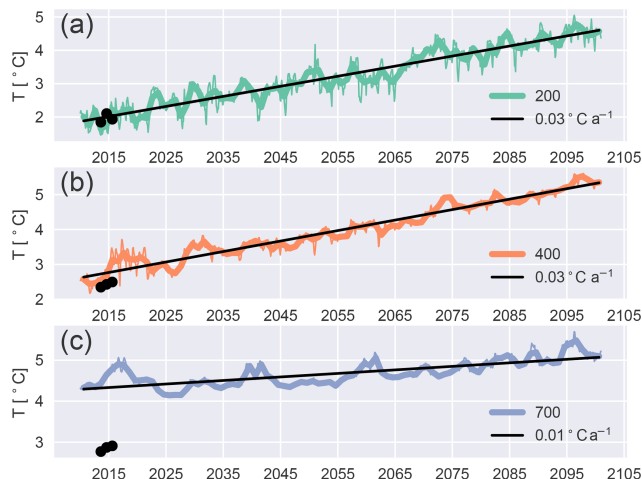

**Figure 5.** Monthly ocean temperature and centennial trend from the CMIP5 model MPI-ESM-LR in the closest grid cells to the fjord of Rink Isbrae that have a depth of at least **(a)** 200 m, **(b)** 400 m and **(c)** 700 m. Black dots show CTD measurements at the same depth but inside the fjord.

**Table 1.** Minimal and maximal ocean temperature trend (rounded to the nearest degree Celsius) over 100 years for three CMIP5 models derived from the grid cells closest to each glacier fjord with minimum 400 m depth. Detailed information are listed in Table S1, Supplement.

| Glacier name | $\Delta T_{\min}$ (°C/100 yr) | $\Delta T_{\max}$ (°C/100 yr) |
|---|---|---|
| Daugaard-Jensen | 3 | 5 |
| Helheim Glacier | 2 | 3 |
| Jakobshavn Isbrae | 2 | 4 |
| Kangerlussuaq Glacier | 3 | 4 |
| Rink Isbrae | 1 | 3 |
| Store Glacier | 1 | 3 |
| King Oscar Glacier | 1 | 3 |
| Alison Glacier | 1 | 3 |
| Upernavik Isstrom N | 1 | 3 |
| Yngvar Nielsen | 1 | 3 |
| Docker Smith Glacier | 1 | 3 |
| Gade Glacier | 1 | 3 |

Thus, for each glacier, we have two temperature profiles (CTD and reanalysis) that are used to simulate the present-day submarine melt rate in the spin-up experiment. We did not consider a seasonal cycle since this would only be represented in the upper surface layer, which is of less importance for the calculation of the submarine melt rate. For future simulations, we prescribed simple scenarios for the ocean temperature anomalies ($\Delta T$) based on minimal and maximal temperature trends simulated by several CMIP5 models (GFDL ESM2G, MPI-ESM-LR and HadGEM2-CC). The trend is added uniformly to all the temperature profiles (both CTD and reanalysis) for the future simulations.

$$T(Z, t) = T(Z)_{\mathrm{CTD/REAN}} + \Delta T_{\min/\max} \cdot t \qquad (20)$$

To determine this temperature trend $\Delta T$ for each CMIP5 model we used the grid cell closest to each fjord but with a depth larger than 400 m. The temperature trends were approximated by linear regression as illustrated in Fig. 5. The figure also shows the big discrepancy between the model temperatures and CTD measurement at 700 m depth, which was the motivation to use 400 m depth only. The temperature trends and cell locations for each glacier and CMIP5 model are listed in Table S1 of the Supplement, while the resulting minimal and maximal temperature trends of these trends for each glacier $\Delta T_{\min/\max}$ are listed in Table 1.

## 4 Experimental setup

### 4.1 Selection of model parameters and model spin-up

Model calibration and spin-up for each glacier has been made in two steps. First, the stand-alone glacier model (without

the plume parameterization) was pre-calibrated to best match observed surface elevation, grounding line position (accuracy $\pm 2$ km has been required) and velocity profile assuming a constant prescribed submarine melt rate. Dynamic parameters $E$, $W_s$, $A_s$ and $q$ (Eq. 2) were varied for this purpose (affecting basal shear stress, lateral stress and calving front boundary condition), along with the freshwater depth in crevasses $d_w$ and the constant melt rate $m$, for each glacier separately. The values of dynamic parameters and relaxation timescales for each glacier are listed in the Supplement Table S2.

Once the four dynamic parameters and the relaxation timescale are set in our pre-calibration, we performed a set of spin-up experiments with the coupled glacier–plume model for each glacier. In the spin-up experiments the submarine melt rate is now simulated interactively by the plume model, which requires subglacial discharge and temperature and salinity profiles as input data. We used monthly subglacial discharge for the year 2000 derived from SICOPOLIS (Sect. 3.3). Vertical temperature and salinity profiles in these experiments were taken from the reanalysis data, averaged over the time interval 1990–2010 or from recent CTD data, and held constant in time (Fig. S3, Supplement). Nonetheless, in the spin-up experiments the submarine melt rate is not constant since changes in the grounding line depth and shape of the floating tongue (if present) affect the submarine melt. We chose the year 2000 as the quasi-equilibrium initial state for "future" climate change simulations since the mass loss of GrIS during the last decade of the 20th century was rather small (ca. 0.1 mm yr$^{-1}$ in sea level equivalent) compared to that observed in the first decade of the 21st century (Vaughan et al., 2013).

We generate an ensemble of model realizations by varying two model parameters: freshwater depth in crevasses $d_w$ and the plume linear scaling parameter $\beta$ (factor in a range from 0.3 to 3 that multiplies the simulated melt rate profile), which control calving rate and submarine melting, respectively. We run the coupled model for each combination of these two parameters over 100 years, so that the glacier at the end of the simulation was close to an equilibrium state and we exclude model versions which simulated the grounding line position more than 2 km distant from the observed one or which display a low-frequency oscillatory behavior with advancing glacier front over the last 20 years of simulations. The list of the parameter range and number of valid realizations for CTD and reanalysis data can be found in the Supplement, Table S3. The partition between calving and submarine melting was available from Enderlin and Howat (2013) for some glaciers, and was used as an additional constraint for the model parameter combinations.

## 4.2 Future climate scenarios

For all future simulations, we used valid combinations of model parameters and corresponding initial conditions obtained at the end of the 100-year spin-up runs. The anomalies of SMB were derived from the regional climate model MAR simulations as described in Sect. 3.3 (Fig. S1, Supplement). To compute future ocean temperature, we use the minimal and maximal ocean temperature trends (Table 1) added to the temperature–depth profiles for each glacier (Sect. 3.4). We prescribe the subglacial discharge for each glacier offline with a monthly time step from the output of the ice sheet model SICOPOLIS. The yearly subglacial meltwater discharge is depicted in Fig. S2. Figure 2 illustrates the data input required for each glacier to simulate their response to future atmospheric and oceanic warming.

All forcing scenarios were applied for the years 2000–2100. In addition, we run the model for 100 years with zero anomalies of temperature, SMB and subglacial discharge to determine unforced model drift.

To express ice volume loss in sea level rise equivalent we used the multiplication factor $l$ under the assumption of oceans occupying $A_{\text{ocean}} = 360 \times 10^6 \, \text{km}^2$:

$$l = \frac{\rho_{\text{ice}}}{\rho_{\text{fw}} A_{\text{ocean}}}, \tag{21}$$

leading to a SLR of $2.55 \times 10^{-3}$ mm for $1 \, \text{km}^3$ of ice volume $V_{\text{SLR}}$ with the density of ice $\rho_{\text{ice}} = 917 \, \text{kg m}^{-3}$, and fresh water $\rho_{\text{fw}} = 1000 \, \text{kg m}^{-3}$.

The contributing ice volume $V_{\text{SLR}}$ is determined by the lost ice volume above flotation from each glacier.

## 5 Results

### 5.1 Present-day state

The simulated present-day glacier thickness and velocity profiles for the different submarine melting and calving ratios are depicted in Fig. 6 with a close-up of the grounding line in Fig. S5, Supplement. Note that we allow for small floating termini since many tidewater glaciers still evolve on a seasonal scale and glacier fronts are also mostly undercut and thus missing a pure vertical cliff without any floating terminus (Bevan et al., 2012; Straneo et al., 2016; Rignot et al., 2015). Each line in the figure corresponds to a different combination of model parameters $\beta$ and $d_w$ listed in Table S3, Supplement. We found that for some glaciers, the grounding line demonstrates a high sensitivity to the melting/calving ratio, while others are primarily controlled by their bedrock topography and have relatively small variations in their grounding line position over the whole melting / calving ratio range. Upernavik Isstrom N (north) is an example of the latter case (Figs. S9 and S10, Supplement). In general, we observed higher velocities at the glacier terminus when higher calving rates were applied. Thus, if a glacier is not strongly buttressed by a sill or lateral resistance, different values of velocity at the glacier terminus due to different $d_w$ values strongly affect the equilibrium grounding line position. Such behavior points to the crucial role of the bedrock topography for glacier dynamics. The simulated realistic velocity profiles (Fig. 6) for Gade Glacier and Jakobshavn Isbrae lead to glaciers slightly thinner than observed. For Jakobshavn Isbrae we were only able to achieve stable states using $T-S$ profiles from the reanalysis data set since CTD measurements showed significantly warmer temperatures and the resulting high submarine melt rate led to the retreat of the glacier on the retrograde (upstream deepening) bedrock.

Table 2 provides a comparison of simulations with observational data derived by Enderlin and Howat (2013). Only the glaciers King Oscar and Docker Smith showed a grounding line flux $\text{Flx}_{gl}$ matching the observational data. All other glaciers have smaller grounding line fluxes than in Enderlin and Howat (2013). However, it should be noted that many glaciers have accelerated since 2000, so it is not clear whether the fluxes reported by Enderlin and Howat (2013) are directly comparable with our equilibrium fluxes. Additionally, Enderlin and Howat (2013) derived submarine melt rates for the floating termini of the glaciers only since they could not account for melting of vertical glacier fronts due to limitations of their methodological approach. For a direct comparison to Enderlin and Howat (2013), we calculate the melt flux (MeltFlx, Table 2) of the simulated glaciers by only considering the mass loss from the floating tongue induced by submarine melting. The ratios of submarine melting to grounding line discharge of our simulations lie within the uncertainty ranges determined by Enderlin and Howat (2013).

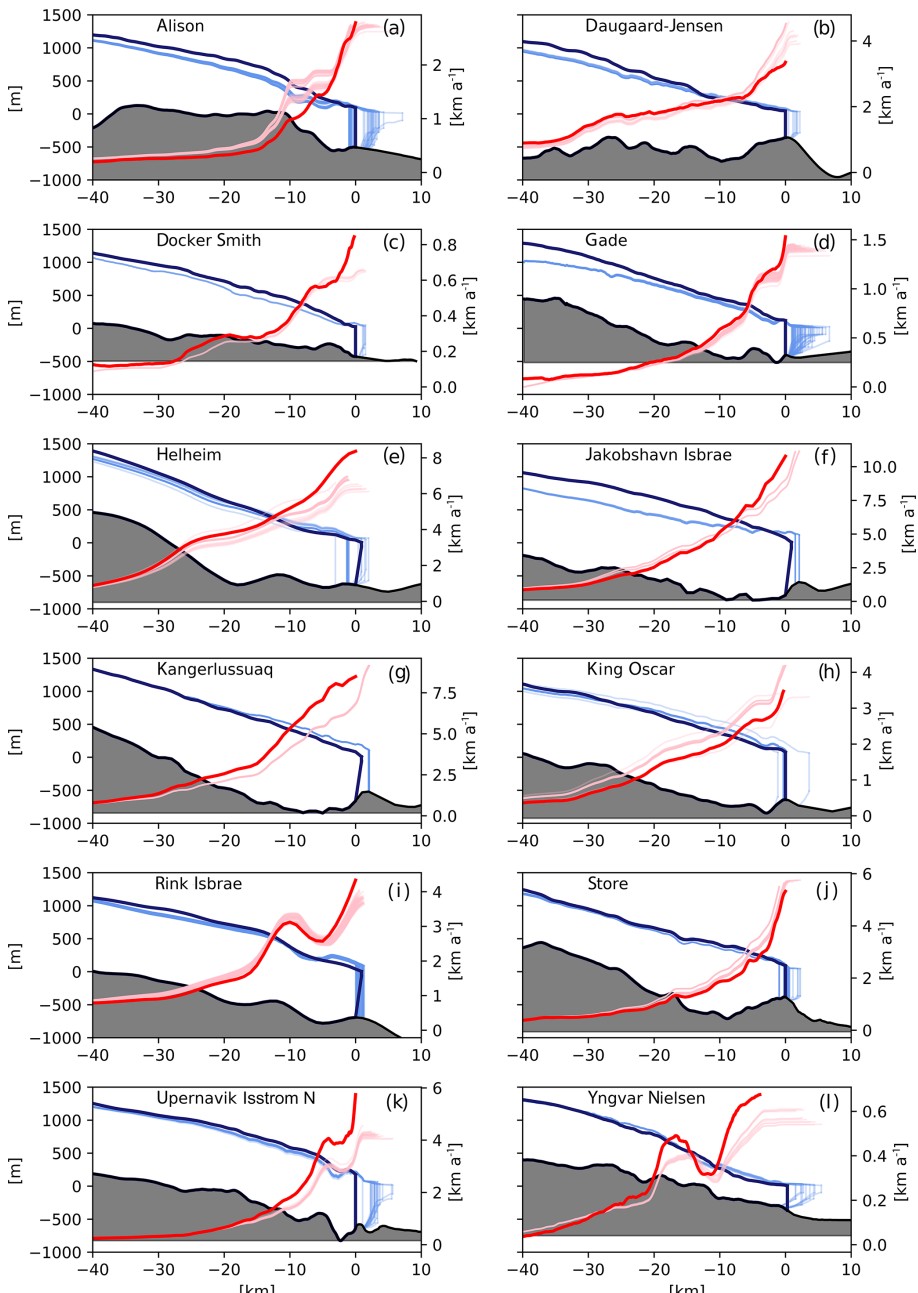

**Figure 6.** Simulated steady state after spin-up. Glacier elevation (light blue) and velocity profile (light red) for the last 40 km to the grounding line depicted together with observational data (dark blue and dark red) by Morlighem et al. (2014) and Rignot and Mouginot (2012). Bedrock data are derived by the flux-weighted average over the whole catchment area. The number of simulations is given in Table S3, Supplement.

However, these uncertainties are quite large and thus allow a broad parameter combination range for some glaciers. For Jakobshavn, a high calving flux was needed in order to obtain a realistic present-day velocity profile for the coupled glacier–plume model (Fig. 6). This results in simulated glacier profiles without any floating terminus (MeltFlx = 0), which is not consistent with Enderlin and Howat (2013). Therefore, this simulated glacier does not match the ratio of submarine melting to grounding line discharge given in Enderlin and Howat (2013) (MeltFlx$^{\star E}$/Flx$_{\mathrm{gl}}^{\star E}$ Table 2). The high calving flux required to obtain the precise grounding line position might result from an error in bedrock data or a problem with the flux-weighted averaging. The simulated Daugaard-Jensen Glacier only has a stable position with submarine melt rates lower than in Enderlin and Howat (2013).

**Table 2.** Each investigated glacier with the mean grounding line discharge from observation $\text{Flx}_{\text{gl}}^{\star E}$ (Enderlin and Howat, 2013) and from the stable-state simulations $\text{Flx}_{\text{gl}}$ as well as the number of stable simulations (No.). The melt flux range for floating termini from all present-day simulations MeltFlx and from the observational data MeltFlx$^{\star E}$ is calculated with the error ranges in Enderlin and Howat (2013) but with the condition $0 < \text{MeltFlx}^{\star E} < \text{Flx}_{\text{gl}}^{\star E}$. The respective ratio of melt flux /grounding line discharge (MeltFlx/$\text{Flx}_{\text{gl}}$) as a percentage is listed for the simulation and observations ($^{\star E}$) and indicates how much ice that flows over the grounding line is lost by submarine melting. The sign * indicates glaciers for which the melt rate partition of the simulation does not overlap with the range of Enderlin and Howat (2013). Melt fluxes are derived for floating tongues and thus MeltFlx = 0 indicates tidewater glaciers (no floating tongue). Store Glacier is not examined in Enderlin and Howat (2013).

| Glacier | $\text{Flx}_{\text{gl}}^{\star E}$ $10^9$ $(\text{m}^2\,\text{yr}^{-1})$ | $\text{Flx}_{\text{gl}}$ $10^9$ $(\text{m}^2\,\text{yr}^{-1})$ | MeltFlx$^{\star E}$ $10^9$ $(\text{m}^2\,\text{yr}^{-1})$ | MeltFlx $10^9$ $(\text{m}^2\,\text{yr}^{-1})$ | MeltFlx$^{\star E}$/$\text{Flx}_{\text{gl}}^{\star E}$ (%) | MeltFlx/$\text{Flx}_{\text{gl}}$ (%) | No. |
|---|---|---|---|---|---|---|---|
| Alison | 6.83 | 6.25–6.55 | 0.82–6.41 | 0.00–4.77 | 12–94 | 0–76 | 54 |
| Daugaard-Jensen* | 9.34 | 7.82–8.44 | 4.12–9.34 | 0.00–2.06 | 44–100 | 0–26 | 22 |
| Docker Smith | 1.06 | 1.05–1.07 | 0.00–0.87 | 0.22–0.66 | 0–82 | 21–62 | 5 |
| Gade | 4.85 | 2.63–2.81 | 0.00–4.85 | 0.17–2.14 | 0–100 | 6–77 | 55 |
| Helheim | 29.16 | 22.84–25.94 | 0.19–6.90 | 0.00–8.39 | 1–24 | 0–36 | 28 |
| Jakobshavn Isbrae* | 43.03 | 36.81–37.14 | 21.11–32.91 | 0.00–0.00 | 49–76 | 0–0 | 11 |
| Kangerlussuaq | 38.80 | 24.51–24.58 | 0.00–6.83 | 0.00–0.00 | 0–18 | 0–0 | 39 |
| King Oscar | 11.86 | 10.34–12.86 | 3.06–6.28 | 0.00–2.64 | 26–53 | 0–26 | 16 |
| Rink Isbrae | 10.95 | 11.20–11.73 | 0.00–6.85 | 0.00–0.00 | 0–63 | 0–0 | 64 |
| Store | – | 10.55–11.29 | – | 0.00–1.73 | – | 0–16 | 67 |
| Upernavik Isstrom N | 17.12 | 7.48–7.84 | 5.81–11.20 | 0.03–5.92 | 34–65 | 0–78 | 21 |
| Yngvar Nielsen | 0.69 | 0.53–0.56 | 0.00–0.69 | 0.08–0.42 | 0–100 | 15–76 | 11 |

## 5.2 Future simulations

After obtaining the present-day state (year 2000), we then ran the model ensemble with all valid combinations of the parameters $\beta$ and $d_{\text{w}}$ for 100 simulation years, applying MAR SMB anomalies, monthly subglacial discharge and two scenarios for ocean temperature change (minimum and maximum) as forcing. All results shown here have a small model drift subtracted from the calculated values, to ensure that the simulated SLR is a response to the climate change signal. The glaciers' response to climate change strongly depends on the combination of model parameters and scenarios, resulting in high uncertainty ranges. The simulations that led to a median range[1] SLR for each glacier are depicted in Fig. 7. After 100 years, some glaciers retreat entirely and become land-terminating (Alison, Daugaard-Jensen, Kangerlussuaq, Store), while others barely show a change in the position of the grounding line (Helheim). The individual contribution of each glacier to SLR for the median range (see footnote 1) SLR experiments is shown in Fig. 8a. Jakobshavn Isbrae shows the most significant contribution to SLR, due to the big catchment area and large retreat, followed by Kangerlussuaq Glacier due to its full retreat.

When forced by comprehensive climate change scenarios (changes in SMB with the surface elevation feedback, ocean temperature $T$ and subglacial discharge $Q$) the median estimate for SLR contribution from all 12 glaciers is about 18 mm (17.9 mm) at the year 2100 (Fig. 8a and b, blue

---

[1]Median for an odd number of simulations, the first value of higher half for an even number of simulation.

curve). To quantify the role of the individual forcing factors, we performed an additional set of simulations with the same model versions corresponding to the median SLR response (18 mm) but applying the three different forcing factors in sequence. With the same model version we rerun the experiment for each glacier omitting changes in subglacial discharge (denoted "SMB + $T$" in Fig. 8b, pink curve) and omitting changes in subglacial discharge and ocean temperatures (denoted "SMB" in Fig. 8b, sum of the brown, orange and yellow areas). The total effect of SMB change on SLR is decomposed into "static" (brown), "dynamic" (yellow) and "dynamic effect with elevation correction" (orange). Static effect was computed as the cumulative integral of SMB anomalies over the fixed present-day catchment and elevation of individual glaciers. As Fig. 8b shows, this component is close to zero, which is explained by the geometry of the glaciers' catchment area, where the ablation area is much smaller than accumulation area. For some glaciers, the cumulative SMB over the glacier's catchment even increased towards the end of the 21st century due to increased precipitation over accumulation area (Fig. S1, Supplement). The SMB forcing used to force the glacier model comes from the original MAR output data (Fig. 2), in which no surface–elevation feedback is considered since the ice sheet surface is considered constant over time. In the dynamic (yellow) experiment we force the glacier model with the original MAR output data (no surface–elevation feedback). Thus most SLR due to SMB change alone occurs through the dynamical processes – thinning and acceleration of glaciers – which in turn affects the calving rate and grounding line retreat. In the third experi-

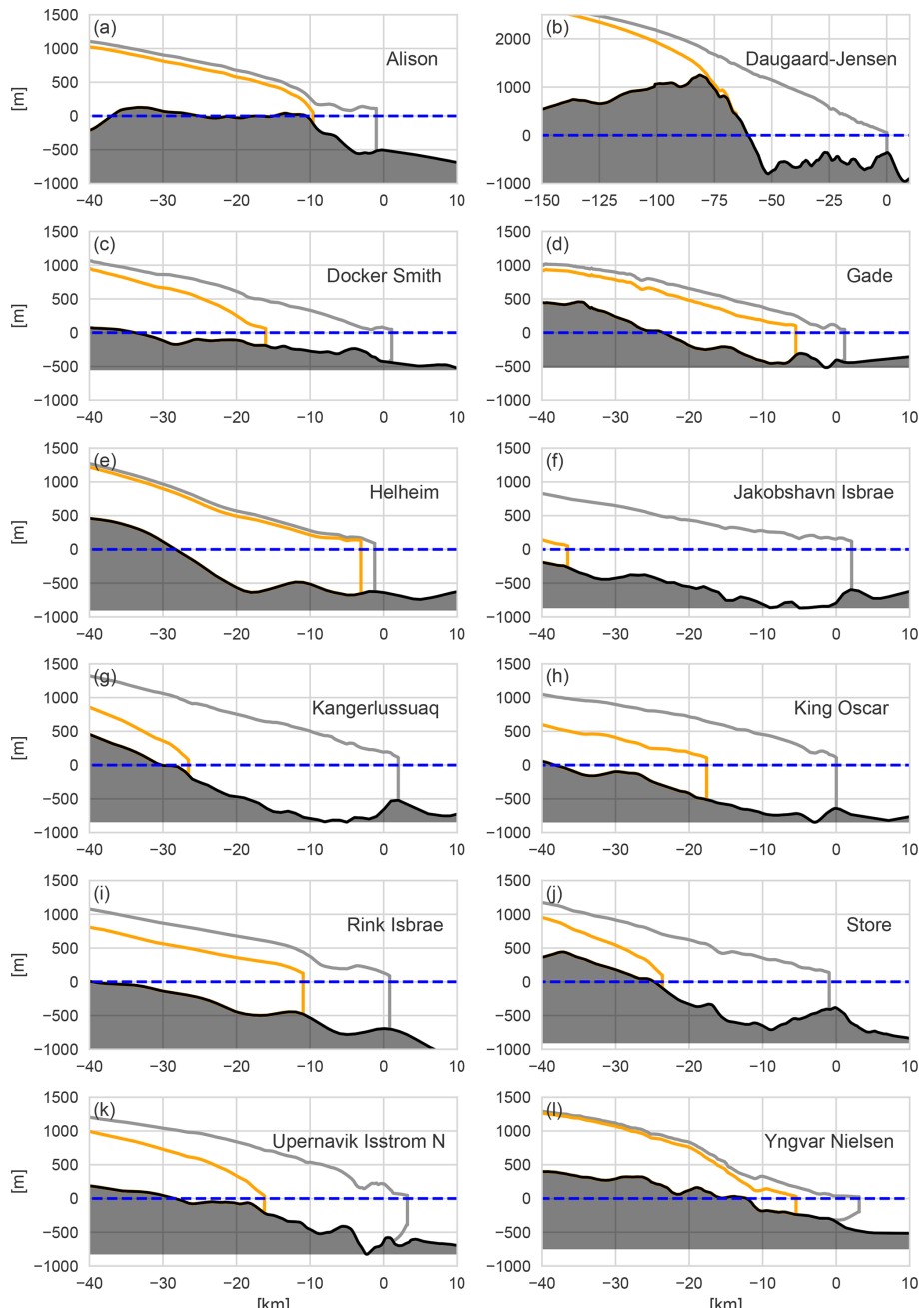

**Figure 7.** Retreat of median range (see footnote 1) SLR scenario for RCP8.5 forcing scenarios (SMB, ocean temperature and subglacial discharge) for all 12 glaciers in 2100 (orange). Corresponding initial states are depicted in grey. Daugaard-Jensen showed full retreat of over 60 km.

ment (orange) we consider the surface–elevation feedback by correcting the SMB forcing for the elevation change on every time step (Eq. 19). The resulting additional SLR (Fig. 8b, orange) is negligible compared to the dynamic response (yellow curve). Thus, the surface elevation feedback has only a minor effect on the glaciers' response to SMB change, which is not the case for the entire GrIS (Calov et al., 2018) where this effect is important. As explained above, we attribute

30 % of the 18 mm SLR to the response to changes in SMB alone. The remaining 70 % of SLR is thus caused by the response to ocean warming and increased subglacial discharge (Fig. 8b, blue and pink area together). We found that both factors, ocean warming and increased subglacial discharge, are of comparable importance for SLR (by comparing the blue and pink curves in Fig. 8b). These estimates are valid only for the cumulative SLR of all 12 glaciers. Each individ-

**Table 3.** Median and first and third quartiles of SLR contribution from each glacier under RCP8.5 scenario (SMB, subglacial discharge and ocean temperature (min and max)). Values are corrected from drift.

| Glacier | SLR (mm) | | |
| --- | --- | --- | --- |
| | Median | First quartile | Third quartile |
| Alison | 0.26 | 0.26 | 0.30 |
| Daugaard-Jensen | 2.73 | 2.12 | 2.84 |
| Docker Smith | 0.18 | 0.15 | 0.19 |
| Gade | 0.17 | 0.14 | 0.30 |
| Helheim | 0.41 | 0.38 | 0.85 |
| Kangerlussuaq | 3.00 | 2.96 | 3.26 |
| King Oscar | 2.89 | 1.83 | 3.61 |
| Rink Isbrae | 1.10 | 0.79 | 1.38 |
| Store | 1.05 | 0.40 | 1.16 |
| Upernavik Isstrom N | 0.85 | 0.63 | 0.98 |
| Yngvar Nielsen | 0.03 | 0.03 | 0.03 |
| Jakobshavn Isbrae | 5.22 | 3.30 | 7.65 |
| Sum | 17.90 | 12.99 | 22.55 |

**Table 4.** Median and first and third quartiles of grounding line retreat from each glacier under RCP8.5 scenario (SMB, subglacial discharge and ocean temperature (min and max)). Values are corrected from drift.

| Glacier | Grounding line retreat (km) | | |
| --- | --- | --- | --- |
| | Median | First quartile | Third quartile |
| Alison | 9.21 | 8.69 | 10.77 |
| Daugaard-Jensen | 60.80 | 28.99 | 62.21 |
| Docker Smith | 15.13 | 14.23 | 16.49 |
| Gade | 5.85 | 4.62 | 15.17 |
| Helheim | 1.52 | 1.10 | 9.63 |
| Kangerlussuaq | 28.52 | 28.44 | 28.53 |
| King Oscar | 17.65 | 14.61 | 18.63 |
| Rink Isbrae | 11.07 | 10.90 | 11.18 |
| Store | 17.59 | 3.99 | 23.21 |
| Upernavik Isstrom N | 17.43 | 12.79 | 17.72 |
| Yngvar Nielsen | 4.69 | 4.28 | 5.22 |
| Jakobshavn Isbrae | 38.57 | 19.85 | 40.53 |
| Avg | 19.00 | 12.71 | 21.61 |

ual glacier may respond differently to the individual forcing factors. For instance, King Oscar Glacier (Fig. 9) is slightly gaining mass with the SMB forcing alone and shows a retreat by 10 km and a contribution of 1 mm to SLR due to ocean warming. When the increase in subglacial discharge is added to the ocean warming, the glacier retreats another 10 km and contributes additionally 2 mm to SLR. At the same time, Yngvar Nielsen Glacier (Fig. 10) is already retreating significantly in the experiment with the SMB forcing alone. Ocean warming and increased subglacial discharge also contribute to SLR, but for Yngvar Nielsen the largest SLR contributor is the SMB change. The different dynamic responses of glaciers can be clearly seen for Rink Isbrae and Store Glacier: both have approximately the same SMB forcing (Fig. S1) but the unstable position of Store Glacier (on the tip of a steep sill, Fig. 6) causes the glacier to be more vulnerable to mass changes at the glacier terminus and when pushed to the retrograde bed the glacier automatically retreats and thus contributes to additional SLR. The dynamic response leads to a significantly higher SLR for Store Glacier than for Rink Glacier.

Above we discussed only median range scenarios, but the uncertainty ranges are crucial when projecting future SLR. Therefore, Fig. 11 shows the first and third quartiles together with the median values of the individual glacier's contributions to SLR for all sets of valid model realizations and full forcing (SMB + $T$ (max/min) + $Q$) against the simulated present-day glacier discharge. Their potential SLR and grounding line retreat are listed in Tables 3 and 4. Figure 11 shows a correlation between present-day grounding line discharge and the contribution to future SLR for individual glaciers. Jakobshavn and King Oscar show the largest spread.

The only "uncertainty" in the ocean forcing is contained in the choice of the ocean temperature trend ($T_{min}$ or $T_{max}$). Thus, to analyze whether the uncertainty ranges in SLR result primarily from the uncertainty range in the forcing or from the uncertainties of the model parameters or the melting-to-calving proportion (uncertainty ranges in $\beta$ and $d_w$) we show in Fig. 12 results of experiments forced only by $T_{min}$ or $T_{max}$ ocean warming scenarios. Figure 12 shows a small future SLR and uncertainty range related to glaciers responding to stand-alone SMB forcing (except for Jakobshavn Isbrae). Since the SMB forcing is the same in all simulations, the spread originates from the differences in initial states caused by different $d_w$ and $\beta$ combinations, and thus different melting-to-calving proportions.

For King Oscar Glacier, the negative SLR originates from the increase in SMB in this region under the RCP8.5 scenario (Fig. S1). Including the forcing factors of submarine melt, $T$ and $Q$ leads to a relatively high SLR contribution and high SLR uncertainty ranges for the King Oscar, Kangerlussuaq, Rink and Daugaard-Jensen glaciers (Fig. 12, shown by the blue columns). Since these high uncertainties also arise with the same forcing (only $T_{min}$ or $T_{max}$), we attribute the major source of uncertainty to the different combinations of the model parameters $d_w$ and $\beta$. For each experiment, we also investigated whether the choice of using CTD measurements or reanalysis data for the initial ocean temperature profile had an impact on the potential SLR (Fig. S11, Supplement). If we neglect Jakobshavn and King Oscar glaciers (no valid simulations with CTD profiles available), only Helheim glacier showed a stronger increase in SLR when reanalysis data were used to construct $T$–$S$ profiles. For the rest of the glaciers the choice of using reanalysis data or CTD data for $T$–$S$ profiles shows only minor differences in SLR.

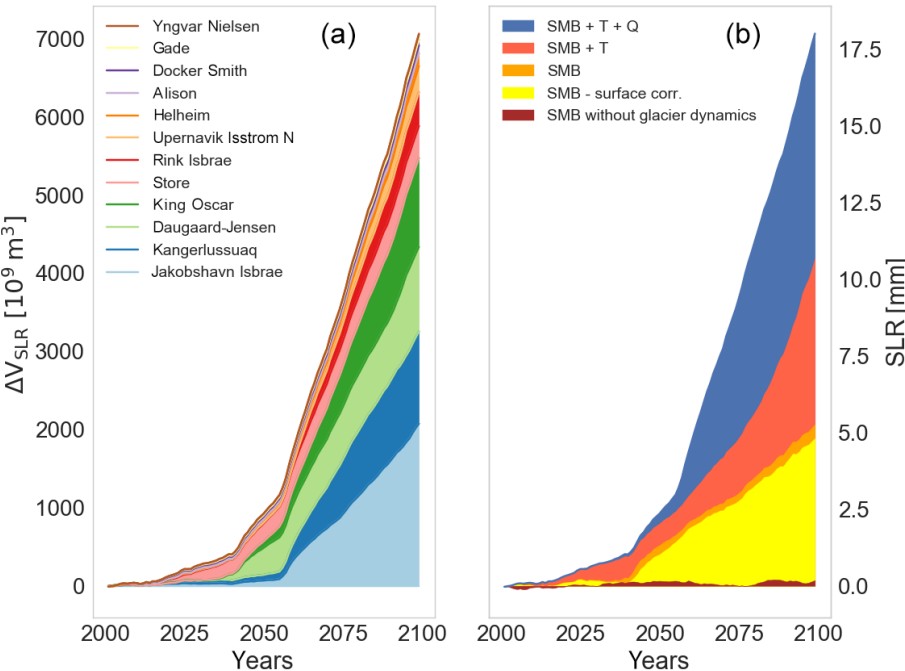

**Figure 8.** Cumulative sea level rise of median range (see footnote 1) SLR scenario from Fig. 7 for all 12 glaciers. **(a)** Individual glaciers' response to a complete future forcing scenario (SMB, subglacial discharge $Q$ and ocean temperature $T$). **(b)** The role of individual forcing factors for all glaciers. The dynamics response of all 12 glaciers forced by $SMB + T + Q$ (blue), $SMB + T$ (pink), SMB forcing only (orange) and SMB without the surface elevation feedback in the glacier model (yellow). The static cumulative SMB anomaly over fixed present-day glacier domains and surface heights from MAR for all 12 glaciers (brown).

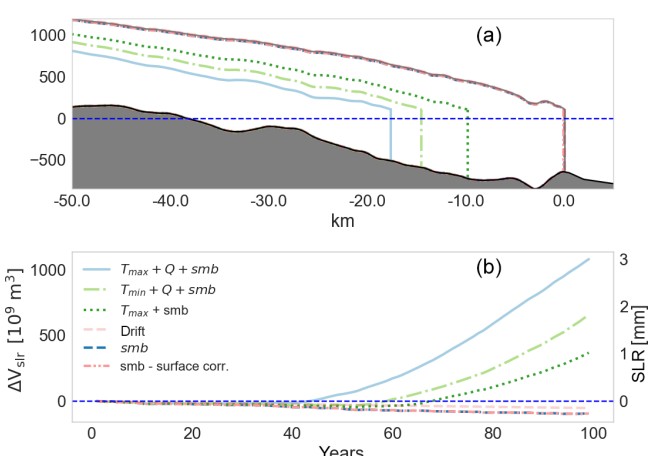

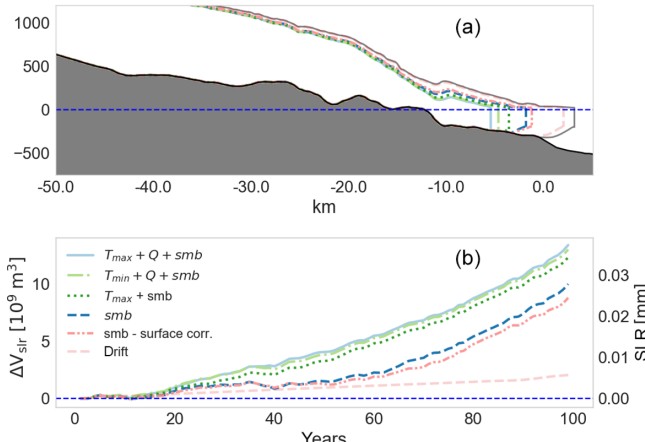

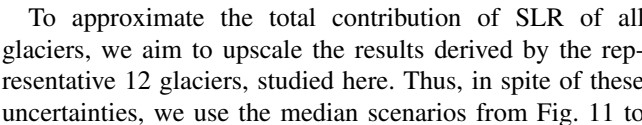

**Figure 9. (a)** King Oscar Glacier with a representative medium-SLR retreat scenario applying forcing factors as subglacial discharge $Q$, ocean temperature $T$, and surface mass balance SMB with and without accounting for surface elevation correction (SMB − surface corr.) for the medium-SLR scenario. The corresponding SLR of each experiment is displayed in panel **(b)**.

**Figure 10. (a)** Yngvar Nielsen Glacier with a representative medium-SLR retreat scenario applying forcing factors as subglacial discharge $Q$, ocean temperature $T$, and surface mass balance SMB with and without accounting for surface elevation correction (SMB − surface corr.) for the medium-SLR scenario. The corresponding SLR of each experiment is displayed in panel **(b)**.

To approximate the total contribution of SLR of all glaciers, we aim to upscale the results derived by the representative 12 glaciers, studied here. Thus, in spite of these uncertainties, we use the median scenarios from Fig. 11 to

estimate the relationship between present-day glacial discharge and contribution to SLR for the year 2100 by fitting a linear regression determined with the least-square method. The derived slope ($0.12 \, \mathrm{mm \, km^{-3} \, yr}$) is statistically signif-

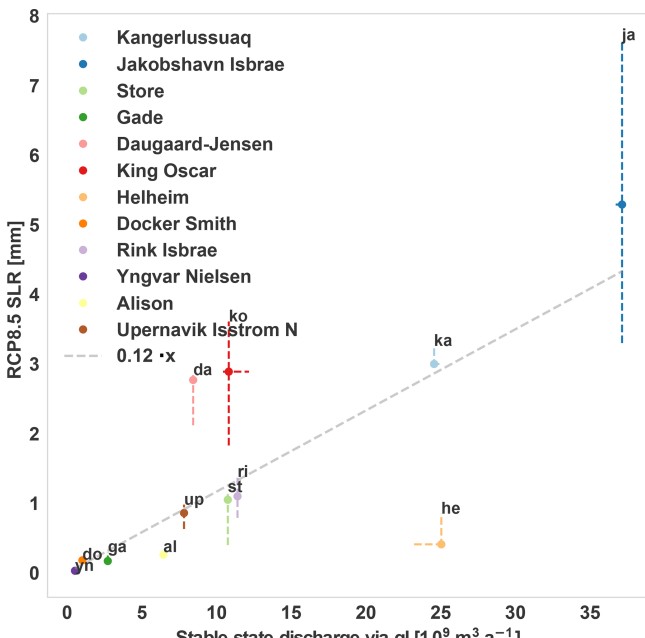

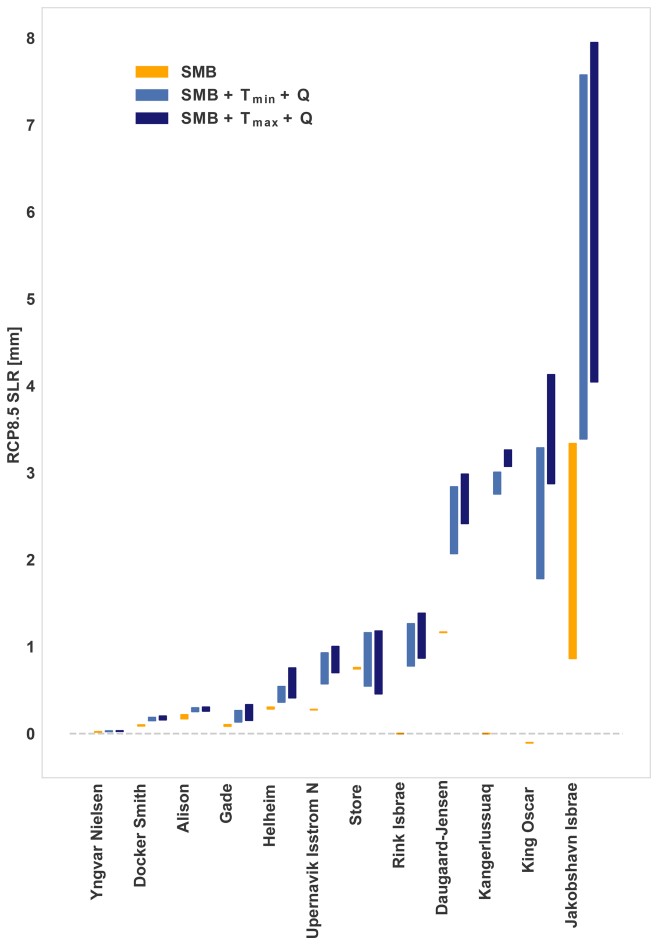

**Figure 11.** First to third quartiles (median indicated with a dot) of contribution to SLR at 2100 under RCP8.5 for each glacier from Table 3 as a function of the present-day grounding line discharge. The future simulations were forced by changes in SMB, subglacial discharge, and minimal and maximal ocean temperature trend (Table 1). Grey dashed line indicates a linear regression obtained with an ordinary least-square method from the median values. Slope and $p$ value are $0.12 \, \text{mm} \, \text{km}^{-3} \, \text{yr}$ and $9 \times 10^{-5}$, respectively. The correlation coefficient is 0.74.

**Figure 12.** First to third quartiles of contribution to SLR for each glacier. Future RCP8.5 scenarios were forced with either SMB changes only (orange) or changes in SMB, ocean temperature ($T_{\text{min}}$ and $T_{\text{max}}$) and subglacial discharge (blue).

icant ($p$ values $< 0.01$) and has a correlation coefficient of 0.74. With this slope and the total flux of all outlet glaciers ($\sim 450 \, \text{Gt} \, \text{yr}^{-1}$; Enderlin et al., 2014; Rignot et al., 2008), the simple linear relationship would imply a total SLR contribution of roughly 5 cm (54 mm) from all Greenland outlet glaciers at the year 2100. This upscaling method is very sensitive to the choice and number of glaciers as Fig. 13 shows. When choosing only four glaciers (as in Nick et al., 2013) to determine the slope of the regression line, the slope can range between 0.03 and $0.16 \, \text{mm} \, \text{km}^{-3} \, \text{yr}$ (by picking the four glaciers that led to the most extreme cases, Fig. 13). This leads to an uncertainty range of roughly 15–80 mm, overlapping with the higher uncertainty range of Nick et al. (2013) (65–183 mm). Due to the low sample size of four glaciers, the resulting regression line is however not statistically significant in the case for the smaller slope of 0.03. Nonetheless, the experiment underlines the importance of choosing a sufficiently large sample size and representative types of glaciers.

## 6   Discussion and conclusions

For 12 selected outlet glaciers of the GrIS, we investigated their potential contribution to SLR during the 21st century for the RCP8.5 scenario. To study the role of future changes in SMB, ocean temperature and subglacial discharge, we used a 1-D flow line model which includes a surface crevasse calving law and is coupled to the 1-D line plume model of Jenkins (2011). In our model, the calving flux can be altered by choosing a parameter for the freshwater depth in crevasses, and the submarine melt rate can be changed by a scaling factor. We also used two different initial temperature–salinity profiles – one derived from reanalysis data and another from in situ measurements inside the fjords. For the present-day simulations, we varied the submarine melting and the calving parameter to obtain a glacier profile similar to observations. For all outlet glaciers, we were able to achieve a reasonable agreement between the simulated and observed present-day profiles. However, for the Jakobshavn

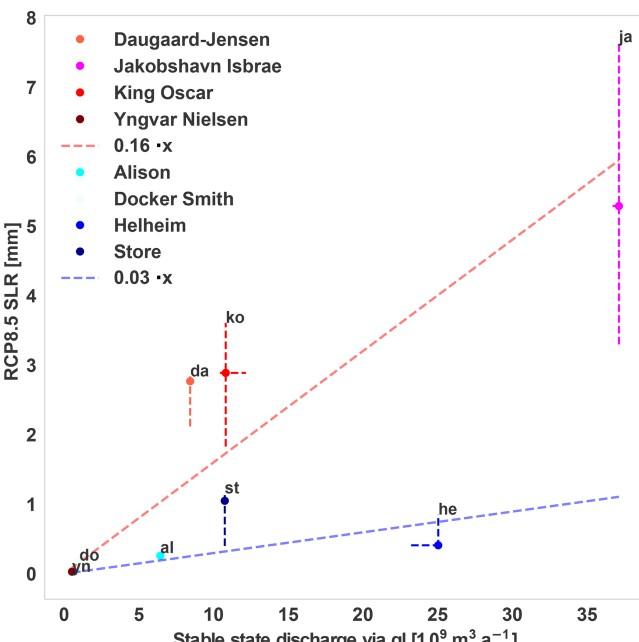

**Figure 13.** First to third quartiles (median indicated with a dot) of contribution to SLR under RCP8.5 for eight glaciers from Table 3 as a function of the present-day grounding line discharge. The future simulations were forced by changes in SMB, subglacial discharge, and minimal and maximal ocean temperature trend (Table 1). Dashed lines indicate a linear function of the present-day grounding line discharge in future SLR for 2100 obtained with an ordinary least-square model from the median values for four glaciers only that result in a high slope (reddish colors) and that result in a low slope (blueish colors). For the reddish glaciers, the resulting slope and $p$ value are $0.03\,\mathrm{mm\,km^{-3}\,yr}$ and 0.18, respectively, and for the blueish glaciers $0.16\,\mathrm{km^{-3}\,yr}$ and 0.01.

Isbrae, the simulated ratio of submarine melt and grounding line discharge does not agree with that derived by Enderlin and Howat (2013), as this ice stream could not develop a floating terminus in our simulations. The melt ratio derived by Enderlin and Howat (2013) could also not be achieved for Daugaard-Jensen.

In order to simulate the future glacial contribution to SLR under the RCP8.5 scenario, we prescribed changes in SMB and subglacial discharge based on results of the regional climate model MAR. Anomalies of ocean temperatures from CMIP5 climate models were used to generate minimum and maximum scenarios for the ocean temperature change until the year 2100. Simulated SLR contributions for the year 2100 compare well to values from Nick et al. (2013) for Jakobshavn Isbrae. The conservative estimates of Jakobshavn Isbrae contribution to SLR obtained with the 3-D model of Bondzio et al. (2017) also lie within our uncertainty range. For Kangerlussuaq Glacier our estimates of SLR contribution exceed the estimates of Nick et al. (2013) by 2 mm, while for Helheim Glacier our SLR estimates are below the estimates of Nick et al. (2013). In our simulations all glaciers

experience a grounding line retreat which is found by Nick et al. (2013) as well but was not simulated by Peano et al. (2017). This discrepancy might be related to the coarse spatial resolution (5 km) of the Peano et al. (2017) model (especially for the deep and narrow trough in Jakobshavn) or processes upstream of the glacier might have counterbalanced the glacier retreat, which we could not simulate with a 1-D flow line model. The difference to Nick et al. (2013) can be explained by their different treatment of calving processes (in their model freshwater depth in the crevasses was linked to runoff) or submarine melting (Nick et al., 2013, did not account for the influence of changing subglacial discharge). Also, Nick et al. (2013) used the surface elevation and velocity profile from the centerline. For Helheim glacier and Kangerlussuaq glacier they took the width of the whole catchment area whereas at Jakobshavn Isbrae the width was constrained to the width of the trough and a constant lateral flux was added to gain the high grounding line flux of Jakobshavn Isbrae. By contrast, we used a flux-weighted average of the whole glacier catchment area to represent each individual glacier.

We also investigated how different forcing factors influence the simulated future SLR. For the ensemble of the 12 glaciers, SLR was over 3-fold larger when the changes in subglacial discharge and ocean temperature were added to changes in SMB. This underlines the critical role of submarine melting in future GrIS contribution to SLR. Moreover, we found significantly larger SLR when the subglacial discharge is allowed to increase in the scenarios. In fact, the amount of SLR attributed to subglacial discharge is similar to the SLR attributed to an increased ocean temperature. Thus, for future projections, both factors affecting submarine melt rate – subglacial discharge and ocean temperature – need to be taken into account. Also, we show that even the almost negligible (compared to SLR) SMB forcing results in a considerable contribution to SLR due to the dynamical response of the glaciers. This response, however is strongly controlled by the underlying bed topography of each individual glacier (e.g., Rink Isbrae and Store Glacier). It should also be noted that our 1-D flow line model is based on a crevasse depth calving law and thus does not account for undercut calving or buoyancy-driven calving (Benn et al., 2017), which in turn is strongly influenced by submarine melting. This mechanism might act as a further amplifier of glacial mass loss that is not accounted for in our results.

Our experiments also reveal large uncertainty ranges, primarily attributed to the different combinations of the two model parameters that determine submarine melting and calving fluxes. Nonetheless, the simulated melt / calving ratios lie within the uncertainty range of observations, and reducing the uncertainties with more precise observational data would probably improve future simulations. On the other hand, our results are not significantly affected by the choice of CTD or reanalysis data when defining the initial ocean temperature and salinity profiles. This suggests that accu-

rate process-based models and observational constraints on submarine melt and calving are more important when making projections about the future response of Greenland outlet glaciers to climate change. Additional uncertainty related to dynamic parameters and topography data (bedrock, width) are not included in this study.

Overall, we obtain a total Greenland glacier SLR contribution of approximately 50 mm when assuming a linear relationship between the glacier's present-day grounding line discharge and their contribution to future SLR. Our estimate for SLR is lower than in Nick et al. (2013) (65–183 mm) partly due to the fact that we also took into consideration smaller marine-terminating glaciers. As Goelzer et al. (2013) argue, these glaciers probably become land-terminating faster than glaciers with a large grounding line discharge and have less mass influenced by ice–ocean interaction. Therefore our upscaling method for the strong climate change scenario should not be used past the year 2100. Furthermore, we demonstrate the sensitivity of the upscaling method to the choice and number of glaciers: by using only four glaciers as in Nick et al. (2013), the different choice of glaciers leads to uncertainty ranges of 15–80 mm SLR.

Our simulations considered a constant catchment area for each glacier and did not account for potential changes in lateral inflow from the ice sheet interior. Such increased mass inflow could result in a smaller grounding line retreat, but an increased inflow would also result in a broadening of the catchment area, as Goelzer et al. (2013) indicate, which could increase ice sheet mass loss further upstream. The full impact can only be assessed with experiments in which outlet glaciers and the parent ice sheet are fully coupled. Additionally, the 1-D flow line model treats lateral processes in a simplified manner, so that more complex bedrock geometries (e.g., branching of glaciers, individual sills, unsymmetrical valley forms) are poorly represented in these estimates. For a first approximation, though, we treat the SLR of 5 cm as additional to that simulated with coarse-resolution GrIS ice sheet models since the cumulative SMB forcing (without glacier response) over the glaciers' area is negligible. Some inconsistency arises from the fact that the database used to initialize the glaciers at the year 2000 is actually based on the measurements made in 2008/2009, but the total contribution of GrIS to global seal level rise during the first 8 years of the 21st century was only about 3 mm and glaciers contributed not more than half of that. Thus this inconsistency has only a minor effect on our moderate approximation of 50 mm.

By adding the 5 cm contribution of outlet glaciers to the 8.8 cm (midrange scenario) simulated by Calov et al. (2018) for the year 2100 using an ice sheet model under the same climate scenario, we arrive at a total GrIS contribution of 138 mm (103–168 mm from lower sample size range).

This implies that the dynamical response of Greenland's outlet glaciers to climate change can increase GrIS contribution to SLR in 2100 by over 50 %.

*Data availability.* For each glacier, the data on glacier geometry, as well as the forcing data (temperature-salinity-profiles, surface mass balance and subglacial discharge under RCP8.5), are openly accessible (https://doi.org/10.5281/zenodo.3365934, Beckmann and Perette, 2019a). The glacier–plume model is still under development by MP and thus is not published yet. The version that was used for this publication is online under a private repository (https://github.com/jojobeck/glacier-plume-model.git, Perrette and Beckmann, 2019) and may be accessed by contacting Johanna Beckmann and Mahé Perrette. The plume model, as a stand-alone subroutine in FORTRAN, is freely accessible at https://github.com/jojobeck/fjordmelt.git, (Beckmann and Perette, 2019b).

*Supplement.* The supplement related to this article is available online at: https://doi.org/10.5194/tc-13-1-2019-supplement.

*Author contributions.* JB designed the study together with AG. MP wrote the glacier model in FORTRAN and provided 1-D topography data for the 12 glaciers of this study. JB coupled the numerical plume model to the glacier model, and implemented the surface-correction method. Together with SB, RC and MW, JB created the projected subglacial discharge and surface mass balance data set for each glacier. JB carried out the experiments, created the figures and wrote the paper, supported by all co-authors.

*Competing interests.* The authors declare that they have no conflict of interest.

*Acknowledgements.* This work was funded by Leibniz-Gemeinschaft, WGL Pakt für Forschung SAW-2014-PIK-1. Matteo Willeit and Reinhard Calov acknowledge support by the BMBF-funded project PalMod.

*Financial support.* This research has been supported by the Leibniz-Gemeinschaft (WGL Pakt für Forschung (grant no. SAW-2014-PIK-1)).

The article processing charges for this open-access publication were covered by the Potsdam Institute for Climate Impact Research (PIK).

*Review statement.* This paper was edited by G. Hilmar Gudmundsson and reviewed by three anonymous referees.

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
