# Peer review of "Modeling the response of Greenland outlet glaciers to global warming using a coupled flowline-plume model"

_The Cryosphere, 2018_

## Referee Comment (RC1) · Anonymous Referee #1 · 10 Jun 2018

**1 Summary statement**

The paper by J. Beckmann et al. analyses the response of 12 glaciers in Greenland to future warming over the next 100 years using a 1D flowline and plume models. The forcing is based on atmosphere and ocean model results from CMPI5, and used in the flowline and plume models. The manuscript is usually clear and well written, and the figures are appropriate. However it is surprising to still rely on flowline models when buttressing and lateral effects can play such a big role in the evolution of glaciers terminating in narrow fjords. The comparison to previous modeling results (especially regional models other than Nick et al. [2013] is rather limited and the implications of the

assumptions made in the models are not discussed. Also, the reasons to use ocean temperatures at 400 m depth are not clear: the depth of the water properties used in the plume model should depend on the fjord and ice front properties (depth of the fjord, presence of sills, depth at the grounding line, ...) and not be identical for all the glaciers. I therefore think that the numerous simplifications made in this paper are not well argued or justified, and the implications on the model results are not assessed and presented in this manuscript.

2 Major comments

As mentioned by the authors, scaling sea level rise at the scale of Greenland using results from only a few glaciers is highly speculative. However, this is still what is done in this paper without assessing the uncertainty of such a scaling. It would be important to quantify the uncertainty in the scaling by comparing results obtained with a subset of models. Also, do you think that 12 glaciers are representative of Greenland? They are many kind of different glaciers, with or without marine terminating fronts, with or without ice shelves, with completely different geometries and fjord conditions; some glaciers are mostly impacted by changes in ice front position, or subglacial hydrology (with different types of subglacial hydrology regimes, ...)? So is it reasonable to use such a small sampling of glaciers and consider that their behavior is representative of the âĹij200 glaciers of the Greenland ice sheet?

I don't understand why the water conditions at 400 m depth are used in the plume model for all the glaciers. All the fjords and glaciers have very different geometry conditions (with sills, ...) that should be taken into account in order to have the right conditions for the plume model.

It looks like the authors are trying to set-up the initial conditions so that the glaciers are in steady state. Many glaciers experienced large changes over the past coupled of decades and they are therefore not in steady-state. I think it is more important to have initial conditions close present state than close to a steady-state, especially as initial

conditions impact the system for a very long time (much longer than the simulation time in this paper). Furthermore, it is difficult to add the present trend to the simulated changes as glaciers are not exactly linear systems.

I don't understand why only one atmosphere model is used for the future forcing while several ocean models are used. There is also no clear distinction between the spread in results caused by the different climate scenarios and the different initial states, and their relative importance for the different glaciers. Is it more important to improve the external forcings (and which one) or to improve the initial conditions to reduce the uncertainty in future glacier's evolution?

It seems like the authors read a coupled of references [Nick et al., 2013; Goelzer et al., 2013], and keep using them all over the manuscript. They are also many regional models (that are not 1D) that should be used to compare the results of this study.

Finally, I must say that I am a bit tired of seeing studies based on flowline models in 2018. I agree that such studies are still very useful to investigate new processes for example, but they should not be used to do future projections of ice sheets, given the importance of buttressing, lateral effects, complex topography ... when 2D regional models can be run at high resolution and provide more accurate results.

3 Line by line comments

p.1 l.6: crudely → simplistic (models use simple parameterizations because the processes remain unknown as mentioned at the beginning of the sentence)

p.1 l.10: Is the regional climate model used only for the SMB or also for other properties?

p.1 l.12 (and l.14 and l.15): use present tense instead of past in the abstract: used → used.

p.1 l.22: the scaling is quite speculative. What happens if you do the scaling with a smaller set of glaciers? What is the uncertainty in this scaling?

p.1 l.19 and l.23: If I understand correctly, the numbers given here do not include the current trend in mass loss from the Greenland ice sheet (13.8 cm in the conclusions). This is rather confusing and provides numbers smaller than expected.

p.2 l.4: "Two processes are largely responsible": What are the other processes that account for mass loss to a lesser extent?

p.2 l.6: "marine-terminating" → "marine terminating"

p.2 l.8: It is not just warming of the ocean, but also changed in the circulation.

p.2 l.9: I doubt that the lower contribution in Fettweis et al. [2013] is 0 cm, it should be 50 mm ($9 \pm 4$ cm). This seems rather contradictory with the actual contribution.

p.2 l.13: Adding references to papers that detail the limitations of modeling of the Greenland ice sheet would be appropriate (e.g. Goelzer et al. [2017]; Khan et al. [2014]).

p.2 l.18-20: I think this is disregarding all the efforts made to improve continental scale models, as some models now have a resolution of about 1 km in marine terminating glaciers [Goelzer et al., 2018]. This is also a bit oversimplifying the problem: the limitations of numerical models are not just resolution, there is also limited observations, external forcings not appropriate, ... So this part of the introduction has to be more balanced.

p.2 l.21-25: Following along the same line, I think jumping from continental scale 3D models to 1D flowline models is a bit reductive, as they are many things in between. Several regional models with 2D or 3D models are starting to show interesting results [Muresan et al., 2016; Bondzio et al., 2017]. Some studies even included representation of ocean with a plume model [Vallot et al., 2018]. So I think the introduction should be improved and not just reduced to Goelzer et al. [2013] and Nick et al. [2013].

p.2 l.32: "that that"

p.3 l.1: You just mentioned that the approach from Nick et al. [2013] is not appropriate, but you follow the same one, just with slightly more glaciers. I am not sure I understand the logic here.

p.3 l.28: "with 3D ice sheet model" → "with 3D ice sheet models"

p.3 l.30: "we used instead" → "we use instead"

p.3 l.31: I agree that continental scale Greenland models are not the best tool to study these processes, but why not use 2D basin models that would at least include lateral deformations and buttressing is important to correctly capture the behavior of narrow outlet glaciers terminating in fjords.

p.4 Eq.2: Can you explain the choice made to incorporate the lateral stress?

p.4 l.9: Where does the basal sliding coefficient come from?

p.4 Eq.5: How different is this from simply applying water pressure at the front?

p.4 l.21: So is there a point exactly at the grounding line position? This should be better explained. Also, how is treated the stretching of the grid, in particular the variables assigned to the new grid points?

p.4 l.27: How about the ice front? Does it evolve with time? And following what criteria? You need to describe the subgrid-scale treatment of the ice front.

p.5 l.2: A quick explanation of the plume model in a few sentences should be added.

p.5 l.8: How is a vertical profile of melt applied to a 1D model, in which there is basically no vertical dimension? So what values is used for the melt (maximum, average, ...)?

p.5 l.12: What happens above the plume? Zero melt?

p.5 l.17: I don't understand "added to the vertical mass balance term B". Is the melt applied to retreat the ice front? Or just to thin the ice close to the ice front? This melt should cause ice front retreat.

p.6 l.1: "Also, did we include" → "We also included"

p.6 l.9 "BedmACHINEev2" → "BedMarchine v2". Also there is new version [Morlighem et al., 2017] that compiled all existing bathymetry data around the Greenland. p.6 l.15: "in the ice sheet" → "in a previous ice sheet"

p.6 l.24: Why not use the mask in Calov et al. (2018)? Combining difference sources for the different datasets might lead to some inconsistencies between the datasets.

p.6 l.26: Explain that the change in "basal melt" refers to ice shelf basal melt and not grounded ice basal melt. I was initially confused given that the previous paragraph talks about subglacial hydrology.

p.7 Eq.9: To be honest I don't like this flux correction in the SMB. The problem of inconsistent datasets and initialization procedures is a real problem that we are facing as a community, and that deserved better treatment than a simple flux correction. This is calibrated for the initial state, but as the glacier evolves with time it is most likely not to be valid anymore. How does this correction impact the results?

p.8 l.19: I am confused about this comparison at different depths? Why not use temperature profiles over the entire depth? Also how did you choose these depths? Do they correspond to the depth of warm or cold water? Or the changes in the thermocline? What is the rational for this choice?

p.8 l.23: This is also the case for Jakobshavn (figure 4).

p.8 l.26-32: I have the impression (and this is not very clear in the manuscript) that you don't use the sill depth in the fjords to determine the water properties in front of the glacier. The sills block the warm water at depth, which can significantly impact the water properties. This should be included for the plume model. Why not use that instead of an arbitrary depth of 400 m? Accurately including the fjord properties in important to separate the response due to the trend in climate changes from the impact of local conditions of the glaciers and the fjords.

p.8 l.29: "larger" → "deeper"

p.9 l.10: Again here, why used the temperature at 400 m depth and not the temperature at the grounding line depth? I think the value used should be designed to best represent the conditions in each and every fjord instead of using a generic value systematically applied to all the fjords.

p.9 l.16-21: It would be great to see the values of the different results, and especially how the different runs agree with the observations. More details on the choice of runs selected should also be added.

p.9 l.22-26: This paragraph is not clear.

p.9 l.27: scaling of what? How is that done?

p.10 l.6: What is 3.3?

p.10 l.14-18: I think this could be easily simplifies in saying that you use the volume above flotation.

p.10 l.21: Mention that is the present-day simulated state.

p.10 l.21: It is not clear what you mean by calving ratio.

p.10 l.23: The grounding line position is not clear on the figure, the ice front position is. Also most of these glaciers do not have any floating tongue, so it would be better to use the term ice front in this case.

p.10 l.21-30: Do you actually want the glaciers to be stable or to be representative of the present-day conditions? Because many of these glaciers are losing mass and retreating today, so how much should a spin-up with present-day conditions lead to stable conditions?

p.11 l.2: I thought that most of these glaciers did not have floating termini anymore!

p.11 l.6 "by Enderlin .."

p.11 l.17: The numbers you provide do not include the present day changes? This is quite surprising and ends up presenting very low sea level change numbers that are not in good agreement with today's observations. It also questions the initialization procedure of the model, how much can we separate the present state and future changes given that the initial conditions have a lasting effect on the results.

p.11 l.26: "excluding" → "separating"

p.11 l.27: This is not very clear, try to better separate the numbers for SMB only, elevation feedback, climate change trend, ocean, ... as is done in figure 11.

p.11 l.30: "substantially" → "substantial"

p.12 l.13: The potential SLR and grounding line retreat are actually not listed in the tables.

p.12 l.15: "uncertainties" → "spread"

p.12 l.18: There is only one model used to generate SMB, so where is the spread coming from? It is not clear if is caused only by the different initial conditions used or if there is something else. Also, why is there only one model used to generate SMB and several for the ocean?

p.13 l.1: "1D line plume model" → "1D plume model" (same in other places in the manuscript). Also "Jenkins (2011)" → "(Jenkins, 2011)"

p.13 l.12: How does that compare to other 2D or 3D models of Jakobshavn [e.g., Muresan et al., 2016; Bondzio et al., 2017]?

p.13 l.19-20: remove

p.13 l.30-35: Use present tense instead of past tense

p.14 l.5: What are the numbers for the entire Greenland if you only take the same glaciers as Nick et al. [2013]? How are these numbers impacted by the choice of

glacier? So, if you only include a subset of the 10 glaciers used in this study, how does the sea level contribution of Greenland vary? It would be interesting to compute some kind of uncertainty associated with this method.

p.14 l.7: "our our"

Fig.2: Is there a white dot in the fjord? It's not very clear. I don't understand the choice for the use of CTD profiles. Why not use all (or a combination of the different) profiles? "depth of 400 m" → "depth of at least 400 m". "od" → "of"

Fig.3: The temperature from the reanalysis data at 700 m depth is quite off compared to the CTD. What are the implications for the plume model and the glacier evolution?

Fig.4: same as Fig.3

Fig.6: It would be better to label all the dots (they are only 12). Again here, why use the depth-averaged temperature and not the temperature that most impact the plume model?

Fig.8: Why present the results from only one ocean model and not from all of them? What is the implication of large discrepancy at 700 m depth between the model and the CTD measurement?

Fig.9: Is the observed bedrock directly taken from the BedMachine dataset along the centerline or is it representative of the entire glacier (of its entire width)? How many stable states are used for each glacier? I could not find this information in the manuscript. And as mentioned above, do you really want the initial configuration to be stable or to represent the current state of the glacier? I am not sure "transparent lines" is the appropriate term.

Fig.10: "median-range3": repeat the superscript meaning here. Fig.11: "vom" → "from"

Fig.12: Try to use the same order as for Fig.11 for the lines. Fig.13: Would be better to repeat the entire caption.

Fig.14: What is "ocean temperature trend 1"?

Tab.2: I thought that most glaciers in Greenland did not had floating termini any more, so why are there relatively large ratios of melting?

Tab.4: It is not clear what the sum of grounding line retreat represent. It is a rather unusual metric.

References

Bondzio, J., M. Morlighem, H. Seroussi, T. Kleiner, M. Ruckamp, J. Mouginot, T. Moon, E. Larour, and A. Humbert, The mechanisms behind Jakobshavn Isbræ's acceleration and mass loss: A 3-D thermomechanical model study, Geophys. Res. Lett., 44, doi:10.1002/2017GL073309, 2017.

Fettweis, X., B. Franco, M. Tedesco, J. H. van Angelen, J. T. M. Lenaerts, M. R. van den Broeke, and H. Gall Ìąee, Estimating the Greenland ice sheet surface mass balance contribution to future sea level rise using the regional atmospheric climate model MAR, Cryosphere, 7(2), 469–489, doi:10.5194/tc-7-469-2013, 2013.

Goelzer, H., P. Huybrechts, J. J. Fu ÌĹrst, F. M. Nick, M. L. Andersen, T. L. Ed- wards, X. Fettweis, A. J. Payne, and S. Shannon, Sensitivity of Greenland ice sheet projections to model formulations, J. Glaciol., 59(216), 733–749, doi:10.3189/ 2013JoG12J182, 2013.

Goelzer, H., A. Robinson, H. Seroussi, and R. S. W. van de Wal, Recent Progress in Greenland Ice Sheet Modelling, Curr. Clim. Change Rep., doi:10.1007/ s40641-017-0073-y, 2017.

Goelzer, H., et al., Design and results of the ice sheet model initialisation experiments initMIP-Greenland: an ISMIP6 intercomparison, The Cryosphere, 12 (4), 1433–1460, doi:10.5194/tc-12-1433-2018, 2018.

Khan, S. A., et al., Sustained mass loss of the northeast Greenland ice sheet triggered

by regional warming, Nat. Clim. Change, 4(4), 292–299, doi:10.1038/NCLIMATE2161, 2014.

Morlighem, M., et al., Bedmachine v3: Complete bed topography and ocean bathymetry mapping of greenland from multi-beam echo sounding combined with mass conservation, Geophys. Res. Lett., 44(21), 11,051–11,061, doi:10.1002/2017GL074954, 2017GL074954, 2017.

Muresan, I. S., et al., Modelled glacier dynamics over the last quarter of a century at Jakobshavn IsbraeôŘřĂ , Cryosphere, 10, 597–611, doi:10.5194/tc-10-597-2016, 2016.

Nick, F. M., A. Vieli, M. L. Andersen, I. Joughin, A. Payne, T. L. Edwards, F. Pattyn, and R. S. W. van de Wal, Future sea-level rise from Greenland's main outlet glaciers in a warming climate, Nature, 497 (7448), 235–238, 2013.

Vallot, D., et al., Effects of undercutting and sliding on calving: a global approach applied to Kronebreen, Svalbard, Cryosphere, 12, 609–625, doi:10.5194/tc-12-609-2018, 2018.
* * *

---

## Referee Comment (RC2) · Anonymous Referee #2 · 14 Jun 2018

PAPER SUMMARY

This paper investigates, by means of numerical modelling, the evolution of 12 outlet Greenland glaciers in the next century (2100). The employed numerical models are a 1D flowline glacier model and 1D (ocean) plume model, they are coupled together.

Two aspects represent important limitations of this work: the use of a 1D glacier model for confined glaciers and the methodology followed in forcing and using the 1D coupled plume model. Some of the assumptions of this work are not properly addressed or discussed, as well as some of the consequences on the obtained results.

This paper is clearly written, with the exception of some paragraphs that may lead to

some confusion about the experimental setup (e.g. It is not clear if you actually run SICOPOLIS or not. Including a "methods section" may ease the reading).

MAIN COMMENTS

On the plume model:

I think that using the coupled 1D plume model is a great improvement. However some experimental choices limit the validity of this improvement.

At page 5 – line 2 is written that "since the plume model in some cases underestimate... we also scale the simulated melt rate profile by a factor Beta...".

I have some comments on this: the relation between the plume forcings (temperature, salinity, shelf/tongue slope, subglacial discharge, ...) and melt rate is given by robust physical equations (Jenkins, 2011; Beckmann et al. 2018). I believe that tuning the obtained melt rates with a multiplying factor waste all the efforts made in using (and coupling) the plume model. What is the need of this sophisticated model if then the computed melt rates are scaled to observed melt rates? Then why not using a simple depth dependent parameterization (e.g. Martin et al., 2011)?

You tuned the computed plume melt rates on present day oberved melt rates. How can you assume that this "present day" scaling will still be valid in 50/100 years? This choice is crucial in terms of providing a robust basal forcing for the glaciers evolution. I think that this assumption should be discussed.

Given the inherent large uncertainties in forcing conditions (both in CTD and in re-analysis, page 8 line 3) what about forcing the plume model with a range of plausible temperature and salinity (from CTD and/or reanalysis) and with a range of subglacial discharges instead of tuning the computed melt rate?

It is not clear why you decide to use reanalysis data at 200, 400 and 700 meters of depth instead of using continous vertical profiles. Moreover, for future simulations you say: "...closest 400m-depth-point neighbor...". Is this motivated by line 29 to 31 at page

8? I understand this choice but I believe that you shold explain this better, clearly motivating also at page 9.

On the glacier model:

I get why you decide to use a 1D flowline model: however I think that the limitations related to this approach (neglect of processes at the lateral boundaries and of buttressing, which play a crucial role in the evolution of ice masses) are not properly tackled and are mostly addressed by saying that 1D models are the only one available for this kind of study.

This is probably right if you want to model 12 (or more) glaciers at the time, but for single glacier the last few years have seen important improvements in modelling alternatives that have produced results for some glaciers that are also modelled in this work (Chaulet et al., 2012; Seddik et al., 2012; Muresan et al., 2016; Peano et al., 2017; Goelzer et al., 2017). I think that the discussion about 1D model limitations should be expanded.

SPECIFIC COMMENTS

Page 1 – line 15: "factor analysis". With factor analysis it is usally meant a statistical method like the Empirical Orthogonal Functions (EOFs), in your work you just exclude (one at the time) the different forcings, I would not strictly define this procedure as a factor analysis.

Page 2 – line 5: instead of "global" I would use "atmospheric"

Page 2 – line 4 to 8: I found this paragraph ok, but I would rearrange it a little bit putting the described processes in the same order you are introducing them.

Page 2 – line 6: "marine terminating" instead of "marine- terminating"

Page 2 – line 16: "In order to..." this should be a new paragraph

Page 2 – line 32: "that" is repeated two times
Page 2 – line 35: "Since we are.." this should be a new paragraph

Page 3 – line 1: I would say that the main (and only) improvement consists in using the coupled plume model. I consider the fact of studying more glaciers just as an "extension" of Nick et al. 2013 work. Moreover, from the scaling perspective, are we sure that the considered glaciers are really representative of all the Greenland glaciers? especially given their variety in terms of glaciers and of confining fjords geometries/conditions.

Page 3 – line 4: ok, but submarine melt rate depends also on the geometrical features of the tongue (shape, slope,...)

Page 3 – line 9 to 11: Maybe you can think about shortly describing how the scaling works.

Page 5 – line 1 to 5: I would expand the plume paragraph since it is the real innovative part of this study. Maybe a short introduction of the basic physics and equations. Otherwise is not clear what do you mean with the E entraiment parameter unless looking at Beckmann et al. (2018) (or already knowing what you are talking about).

Page 5 – line 17: "to the vertical mass balance term B", add the equation number

Page 5 – line 18 to 20: I imagine that when the plume detaches the melt rate is set to zero but this is not written explicitly. Is this the case?

Page 5 – line 21: this part confused me. "...off-line using the ice sheet model" which one? This is the first time that you mention the use of an ice sheet model. Later it appears that it is SICOPOLIS.(see comment to page 6 – line 15 to 25)

Page 6 – line 1: "did we" "we did". Could you explain better in what this upscaling consists and how it works?

Page 6 – line 2: it would add more clarity defining what is meant with "melting to calving ratio"

Page 6 – line 12: just a detail: I would number the figures in the order of appeareance in the manuscript

Page 6 – line 15 to 25: From here it looks you actually run the ice sheet model, is this correct? (look comment to page 10 – line 6).I suggest to introduce explicitly the fact that you have run SICOPOLIS.

Page 6 – line 23,24: "...is assigned to the closest glacier within a maximum of 50 km". This is an important approximation since is related to the plume forcing, however is not properly discussed, expecially in terms of uncertainty in the obtained results.

Page 6 – line 27,28: "neglect the effect of grounding line retreat".As above, this represents another important assumption but it is not properly discussed.

Page 8 – line 12: "...presence of sills in the fjord...in the vicinity of the glacier front." I would explain why is that after this line, instead than explaining it later for the continental shelf (at page 8 – line 24 to 30).

Page 9 – line 16: could you provide a table with the prescribed submarine melt rate and the range of values for the dynamic parameters? (maybe in the supplementary)

Page 9 – line 25: with "...only factors.." do you mean that since temperature and salinity are "held constant" (thus not changing) their contribution in impacting melt rates is constant in comparison to the impacts due to a varying grounding line depth and tongue shape/slope? I suggest to reformulate this paragraph

Page 9 – line 29 "...is close to equilibrium state.." what do you mean with equilibrium? Later you speak about stable state. Do you mean steady? I would argue that currently Greenland glaciers are definitely not in a steady condition.

Page 10 – line 5: "...each glacier 3.4..." something is missing between glacier and 3.4

Page 10 – line 6: "...glacier individually 3.3..." something is missing between individually and 3.3

Page 10 – line 6: Here it is not clear if you took the data from Calov et al. 2018 or if you actually run the model

Page 10 – line 22,24: this part about the interplay between melting, calving and bedrock is interesting. I would add few more details.

Page 11 – line 6: a space is missing before "Enderlin"

Page 11 – line 15: "model versions" do you mean the the spin-up ensemble?

Page 11 – line 16: why not changing also the subglacial discharge? It is such an important forcing for the plume and comes from several approximations (fixed grounding line and closest neighboring approach).

Page 11 – line 17: at page 10 (line 8 to 10) is said that also the unforced model drift is calculated. Then this drift is removed by subtracting it from calculated values. This implies that a linear behaviour for glaciers is assumed. I think that this should be properly discussed.

Page 11 – line 25-27: as above, this implies linearity but glaciers are definitely not linear systems.This issue is just slightly addressed at page 12 – line 4. Page 12 – line 21,22: you attribute the source of uncertainty to Beta, this comes from the fact that Beta is responsible for the imposed melt rate (through the tuning procedure). However Beta is just a model parameter, I think that avoiding the use of Beta (as suggested in the main comments) could also improve this part of the work, it will allow you to relate uncertainties to physical quantities.

Page 13 – line 18 to 20: something is wrong here, an entire sentence is repeated.

Page 13 – line 33: same as above. Your results are not affected by CTD/reanalysys temperature and salinity because the Beta tuning incorporates all the uncertainties.

Page 13 – line 35: "...observational constraints on submarine melt..." as explained in the main comments I think that we should rely on melting formulation as less as

possible dependent from a tuning on observations, especially for future projections.

Page 14 – line 4: "and" repeated two times

Page 14 – line 7: "our" repeated two times

Figure 3(a): I think that using white dots is a bit unfortunate, also the red star is not very visible.

Figure 11: "from" instead of "vom"

REFERENCES

Beckmann, Johanna, Mahé Perrette, and Andrey Ganopolski. "Simple models for the simulation of submarine melt for a Greenland glacial system model."ÂăThe CryosphereÂă12.1 (2018): 301.

Gillet-Chaulet, F., Gagliardini, O., Seddik, H., Nodet, M., Durand, G., Ritz, C., Zwinger, T., Greve, R., and Vaughan, D. G.: Greenland ice sheet contribution to sea-level rise from a new-generation ice-sheet model, The Cryosphere, 6, 1561-1576, https://doi.org/10.5194/tc-6-1561-2012, 2012.

Goelzer, H., A. Robinson, H. Seroussi, and R. S. W. van de Wal, Recent Progress in Greenland Ice Sheet Modelling, Curr. Clim. Change Rep., 2017.

Jenkins, Adrian. "Convection-driven melting near the grounding lines of ice shelves and tidewater glaciers."ÂăJournal of Physical OceanographyÂă41.12 (2011): 2279-2294.

Winkelmann, R., et al. "The Potsdam parallel ice sheet model (PISM-PIK)-Part 1: Model description."ÂăThe CryosphereÂă5.3 (2011)

Muresan, I. S., Khan, S. A., Aschwanden, A., Khroulev, C., Van Dam, T., Bamber, J., van den Broeke, M. R., Wouters, B., Kuipers Munneke, P., and Kjær, K. H.: Modelled glacier dynamics over the last quarter of a century at Jakobshavn Isbræ, The Cryosphere, 10, 597-611, https://doi.org/10.5194/tc-10-597-2016, 2016.

Peano, D., Colleoni, F., Quiquet, A., & Masina, S. (2017). Ice flux evolution in fast flowing areas of the Greenland ice sheet over the 20th and 21st centuries. Âă Journal of Glaciology, Âă63(239), 499-513. doi:10.1017/jog.2017.12

Seddik, Hakime, et al. "Simulations of the Greenland ice sheet 100 years into the future with the full Stokes model Elmer/Ice." Âă Journal of Glaciology Âă58.209 (2012): 427-440.
* * *

---

## Author Comment (AC1) · 6 Sep 2018

**Response to Reviewer 1**

We thank the reviewer of the constructive reviews and suggestions. The comments by the reviewer are in indented blocks and italic fonts.

**Major comments**

> *As mentioned by the authors, scaling sea level rise at the scale of Greenland using results from only a few glaciers is highly speculative. However, this is still what is done in this paper without assessing the uncertainty of such a scaling. It would be important to quantify the uncertainty in the scaling by comparing results obtained with a subset of models. Also, do you think that 12 glaciers are representative of Greenland? They are many kind of different glaciers, with or without marine terminating fronts, with or without ice shelves, with completely different geometries and fjord conditions; some glaciers are mostly impacted by changes in ice front position, or subglacial hydrology(with different types of subglacial hydrology regimes, ...)? So is it reasonable to use such a small sampling of glaciers and consider that their behavior is representative of the 200 glaciers of the Greenland ice sheet?*

We agree with the reviewer that the accurate estimate of the contribution of Greenland outlet glaciers to future sea level rise will only be possible when all 200+ glaciers will be accurately modeled. The fig.14 in our paper, first of all presents the test of the "scaling up technique" used by Nick et al. (2013) who estimated the total contribution of outlet Greenland glaciers by using results of simulations made only for four major Greenland glaciers. Note that Nick et al. (2013) did not tested whether a correlation between present-day discharge and future sea level rise even exists. Here we used 12 glaciers which differ significantly by discharge and location. We do not claim that they properly represent all Greenland glaciers but, still, this is an obvious step forward compare to the previous study. We found (Fig. 14) that some correlation between present-day discharge and the future contribution to sea level rise does exist . Since we consider only a small subset of Greenland glaciers and only some sources of uncertainties, we do not think that the detailed uncertainties analysis of the relationship between discharge and sea level rise would be very helpful. However, we will demonstrate how the uncertainty ranges by the choice of glaciers in the supplementary part.

> *I don't understand why the water conditions at 400 m depth are used in the plume model for all the glaciers. All the fjords and glaciers have very different geometry conditions (with sills, ...) that should be taken into account in order to have the right conditions for the plume model. It looks like the authors are trying to set-up the initial conditions so that the glaciers are in steady state.*

This is a misunderstanding and we will improve this part of the manuscript to make more clear how we constructed the T-S profile and why we use "the water conditions at 400 m depth". Firstly, it is important to note that for all 12 glaciers used in this study, there are at least some CTD profiles from the adjacent fjords and, in spite of some problems with CTD profiles (discussed in the manuscript), the CTD profiles are always our first choice to force the plume model. However, CTD profiles are not yet available for all 200+ Greenland glaciers. This is is why (in addition to CTD profile) we tested whether T-S profiles in Greenland fjords can be constructed using the results of the ocean reanalysis project. Since the reanalysis data are only available for the open ocean, we used the nearest to the fjord's mouth reanalysis grid cells with the depth 200, 400 and 700 m (these are the top three vertical levels in the reanalysis data set). By comparing reanalysis data with the corresponding CTD data (Fig. 4 and 5) we found that at depth 200 and 400 m reanalysis and CTD data are in reasonable agreement while for the 700 m depth they are completely off, which is explained by the fact the 700m depth-points are always located outside the continental shelf .Therefore they are not appropriate to produce vertical temperature profile in the fjords . Therefore instead of interpolated between 400 and 700 m values, we choose to prescribed below 400 m as constant temperature equal to temperature at the depth 400 m in the reanalysis data.

We then compare submarine melt computed using CTD profiles with those have been computed using temperature and salinity profiles from 200 and 400 m depths in the reanalysis data. We found that results are in reasonable agreement. Therefore we recommend as a temporal option (before better data will be available) to use the nearest gridcells with depth 200 and 400 m to construct T and S profile from the reanalysis data for the fjords for which CTD data are not yet available.

> *Many glaciers experienced large changes over the past couple of decades and they are therefore not in steady-state. I think it is more important to have initial conditions close present state than close to a steady-state, especially as initial conditions impact the system for a very long time (much longer than the simulation time in this paper). Furthermore, it is difficult to add the present trend to the simulated changes as glaciers are not exactly linear systems.*

The meaning of the "present day" should be properly defined – otherwise it causes confusion. We did not assume that the glacier are in the equilibrium state at present, i.e. in the year 2018. In our paper under "present day" we mean the years 2000 which we use as the starting time for all our forced simulations. The choice of the year 2000 is motivated by the fact that the mass loss of GrIS during the last decade of 20the century was rather small (ca. 0.1 mm/yr) compare to that has been observed in the 21$^{st}$ century. This justify our assumption about quasiequilibrium state of Greenland glaciers at the beginning of experiments. Some inconsistency arises from the fact that the database (BedmACHInev2) we used to initialized the glaciers at the year 2000 are actually based on the measurements made in 2008/2009.However even the total contribution of GrIS to global seal level rise during the first 8 years of the 21$^{st}$ century was only about 3 mm and glaciers contributed not more than half of that. This is of course a very small number compared too our total estimate of 50 mm of glaciers contribution during the entire 21$^{st}$ century. For this reason we see no need in adding "present day" trend to the results we obtained. To the contrary, we extracted from the results a very small trend diagnosed in the control (unforced) run.

> *I don't understand why only one atmosphere model is used for the future forcing while several ocean models are used.*

It is well-known that for the entire GrIS contribution to sea level rise, climate change scenarios (both in term of GHGs concentration and the model output) are the major source of uncerstainties. In this this study we decided to concentrate on the new issue, namely glacier-ocean interaction. Therefore, we used output of only one regional climate model for a single climate change scenario (RCP8.5) and concentrated on the uncertainties related to parameterizations of submarine melting and calving. We agree with the reviewer that applying another atmospheric model would introduce additional uncertainties in sea level rise via different smb and subglacial discharge. We would like to address this in future work.

> *There is also no clear distinction between the spread in results caused by the different climate scenarios and the different initial states, and their relative importance for the different glaciers. Is it more important to improve the external forcings (and which one) or to improve the initial conditions to reduce the uncertainty in future glacier's evolution?*

We do not agree with the reviewer here. In Figure 15 we demonstrate the spread of results for each single forcing scenario and therefore attribute the spread to the different initial states, thus beta, and fwd. For clarity we will state in the section future results: "We attribute the major source of uncertainty to the different combinations of the model parameters fwd and beta. "

> *It seems like the authors read a coupled of references [Nick et al., 2013; Goelzer et al., 2013], and keep using them all over the manuscript. They are also many regional models (that are not 1D) that should be used to compare the results of this study.*

Citing of Nick and Goelzer is natural since we used a similar approach and compare our results with these two studies. Of course, we are aware about regional Greenland ice sheet modeling and, although in most cases it is difficult to compare directly our results with regional modeling, , we will include a comparison to other regional models for a broader discussion.

> *Finally, I must say that I am a bit tired of seeing studies based on flowline models in 2018. I agree that such studies are still very useful to investigate new processes for example, but they should not be used to do future projections of ice sheets, given the importance of buttressing, lateral effects, complex topography ... when 2D regional models can be run at high resolution and provide more accurate results.*

We respect the reviewer's opinion on the issue which models should be used for future projections of ice sheets. However, the focus of our study is not on future projection of the Greenland Ice Sheet but rather on the response of outlet marine-terminated glaciers to climate change and, primarily, on the analysis of uncertainties related to two poorly constrained processes: submarine melt and calving. We appreciate the importance of buttressing, lateral effects and complex topography which unavoidably are treated in a rather simplistic way in the 1-D model. However, we doubt whether at present 2D models can really provide "more accurate results" since accurate modeling of marine terminated glaciers would require as input accurate knowledge of present and future (i) fjord bathymetry, (ii) temporal variability of the 3D fields of temperature, salinity and velocity in the fjord, and (iii) the spatial-temporal  distribution of subglacial discharge of melt water into the fjords. Even at present all these characteristics are not accurately known for Greenland glaciers  and in most cases they are not known at all. At the same time we agree that the paper will benefit from the discussion of model limitations and future perspectives.

**Line by line comments**

> *p.1 l.6: crudely → simplistic (models use simple parameterizations because the processes remain unknown as mentioned at the beginning of the sentence)*

Agreed, we replace "crudely" with "simplistically"

> *p.1 l.10: Is the regional climate model used only for the SMB or also for other properties?*

The other property is surface runoff as mentioned in the sentence." ...forcing the model with changes in surface mass balance and surface runoff...,"

> *p.1 l.12 (and l.14 and l.15): use present tense instead of past in the abstract: used →*
>
> *used.*

Agree, we will adapted the tense.

> *p.1 l.22: the scaling is quite speculative. What happens if you do the scaling with a smaller set of glaciers? What is the uncertainty in this scaling?*

We thank the reviewer for this interesting suggestion. We will derive such an uncertainty as proposed.

> *p.1 l.19 and l.23: If I understand correctly, the numbers given here do not include the current trend in mass loss from the Greenland ice sheet (13.8 cm in the conclusions). This is rather confusing and provides numbers smaller than expected.*

The numbers given in the abstract (14 mm for twelve glaciers and 50 mm obtained by scaling up for all glaciers) can be considered as complimentary to the numbers order of 100 mm computed in the coarse-resolution GrIS models (e.g. Calov et al., 2018) because in the latter the mass loss of the GrIS is mostly controlled by changes of SMB, while mass loss of outlet glaciers is primarily controlled by increased submarine melt. We will make this point very clear in the revised manuscript.

> *p.2 l.4: "Two processes are largely responsible": What are the other processes that*

*account for mass loss to a lesser extent?*

We will add the percentage in brackets: (60 %) surface melting and (40 %) dynamical processes.

*p.2 l.6: "marine-terminating" → "marine terminating"*

Agreed.

*p.2 l.8: It is not just warming of the ocean, but also changed in the circulation.*

We agree, the circulation changes led to the warming of the ocean. The sentence will now read: "...which can in turn be attributed to a warming of the subpolar North Atlantic ocean, induced by circulation changes, and increased subglacial discharge"

*p.2 l.9: I doubt that the lower contribution in Fettweis et al. [2013] is 0 cm, it should be 50 mm (9 ± 4 cm). This seems rather contradictory with the actual contribution.*

We thank the reviewer for spotting this mistake. 0 cm was cited in Fettweis et al. 2013 from other studies. We changed the number to 50 mm.

*p.2 l.13: Adding references to papers that detail the limitations of modeling of the Greenland ice sheet would be appropriate (e.g. Goelzer et al. [2017]; Khan et al. [2014]).*

We thank the reviewer for this suggestion. We will cite the above-mentioned literature.

*p.2 l.18-20: I think this is disregarding all the efforts made to improve continental scale models, as some models now have a resolution of about 1 km in marine terminating glaciers [Goelzer et al., 2018]. This is also a bit oversimplifying the problem: the limitations of numerical models are not just resolution, there is also limited observations,external forcings not appropriate, ... So this part of the introduction has to be more balanced.*

We fully agree with this point and will modify the introduction accordingly.

*p.2 l.21-25: Following along the same line, I think jumping from continental scale 3D models to 1D flowline models is a bit reductive, as they are many things in between. Several regional models with 2D or 3D models are starting to show interesting results [Muresan et al., 2016; Bondzio et al., 2017]. Some studies even included representation of ocean with a plume model [Vallot et al., 2018]. So I think the introduction should be improved and not just reduced to Goelzer et al. [2013] and Nick et al. [2013].*

We thank the reviewer for pointing out this different studies. We will mention these and several other  applications of 3D models to study response on  regional and  shorter time scalesAs far as a very interesting paper by Vallot et al. (2018) nicely illustrates that high-resolution and physically based modeling of glacier-ocean interaction is already possible but absolutely impractical for the study of glacier response to global warming. While Vallot et al. (2018) studied only one melt season with the glacier model., they were able to run plume model only for 10 minutes and only for a small fraction of the ice front. Therefore we cannot see an alternative to highly simplified parameterization of the glacier-ocean interaction when the centennial time scale response is concerned.

*p.2 l.32: "that that"*

Will be deleted.

*p.3 l.1: You just mentioned that the approach from Nick et al. [2013] is not appropriate, but you follow the same one, just with slightly more glaciers. I am not sure I understand the logic here.*

We did not state that the approach from Nick et al "is not appropriate". This study has obvious limitations, such as using of only four largest glaciers to project the the entire 200+ glaciers contribution to sea level rise as well as a very simplistic parameterization for the submarine melt. We wrote that "we followed an approach similar to Nick et al. (2013) but with several notable improvements" and our major improvement is using of more glaciers and more physically based parameterization for submarine melt.

*p.3 l.28: "with 3D ice sheet model" → "with 3D ice sheet models"*

Agreed, will be changed in the revised version.

*p.3 l.30: "we used instead" → "we use instead"*

Agreed, will be changed in the revised version.

*p.3 l.31: I agree that continental scale Greenland models are not the best tool to study these processes, but why not use 2D basin models that would at least include lateral deformations and buttressing is important to correctly capture the behavior of narrow outlet glaciers terminating in fjords.*

We thank the reviewer for his suggestion but we want to point out that 1D flowline models include lateral deformation and buttressing in a simplistic manner. We will however mention the limitation of the 1D model in the discussion.

*p.4 Eq.2: Can you explain the choice made to incorporate the lateral stress?*

The lateral stress term is necessary because the glaciers we considered for this study, like most Greenland marine-terminating glaciers, narrow-down toward their terminus (width of the order of 5 km, besides Petermann), with velocity of the order of 1000 meters per year (10 000 m/a for Jakobshavn Isbrae, according to present observations), making it impossible to neglect lateral drag. The stress term was derived e.g. by Veen and Whillans (1996), and used by various authors since (e.g. Nick et al., 2013; Enderlin et al., 2013; Schoof et al, 2017).

*p.4 l.9: Where does the basal sliding coefficient come from?*

The basal sliding coefficient (equation 2) was determined, along with other uncertain parameters, from calibration to present-day state. We will give a list ob basal sliding coefficient in the SI.

*p.4 Eq.5: How different is this from simply applying water pressure at the front?*

It is not and we thank the reviewer for spotting the lack of explanation here. We will introduce the description of equation 5 with "while at the calving front, the difference of the hydrostatic pressure between ice and sea water is balanced by the longitudinal stress gradient".

*p.4 l.21: So is there a point exactly at the grounding line position? This should be better explained. Also, how is treated the stretching of the grid, in particular the variables assigned to the new grid points?*

Yes, there is. Grid stretching is performed so that there is always a cell edge at the interpolated grounding line position. The new calving front position is determined so that the total glacier volume is not modified by interpolation. For every new point in the interior, model variables are interpolated from previous grid. The first grid point at the ice divide remains unchanged. If ice grid points on the new grid lie outside the ice domain on the previous grid, as it is typically the case for the last cell before the calving front, ice thickness from the last grid cell is extended.
We will clarify this and give a deeper explanation of the stretching grid in the next version of the paper.

*p.4 l.27: How about the ice front? Does it evolve with time? And following what criteria? You need to describe the subgrid-scale treatment of the ice front.*

We agree with the reviewer and will describe the subgrid-scale and ice-front treatment in the revised version of the paper.

> *p.5 l.2: A quick explanation of the plume model in a few sentences should be added.*

Agreed, for completeness we will add and describe now the equations of plume model.

> *p.5 l.8: How is a vertical profile of melt applied to a 1D model, in which there is basically no vertical dimension? So what values is used for the melt (maximum, average, ...)?*

The cumulative melt rate is calculated as a volume flux and added to the mass balance term. We describe in more detail the treatment of the cases floating tongue and tidewater glacier.

> *p.5 l.12: What happens above the plume? Zero melt?*

We thank the reviewer for spotting this lack of information.
Above the plume, so if the plume ceases, we set the melt rate to a minimum background melt which is given by the last melt value of the ceasing plume. We will add the important information in the revised version of the paper.

> *p.5 l.17: I don't understand "added to the vertical mass balance term B". Is the melt applied to retreat the ice front? Or just to thin the ice close to the ice front? This melt should cause ice front retreat.*

We thank the reviewer for mentioning this unclarity. The melt is applied to thin the front, and does not cause retreat. Thinning subsequently leads to calving.
We will explicitly describe the ice front treatment of the glacier model in the 2nd version of the paper.

> *p.6 l.1: "Also, did we include" → "We also included"*

*Will be chnaged to the reviewers suggestion in the revised version of the paper.*

> *p.6 l.9 "BedmACHINEev2" → "BedMarchine v2". Also there is new version [Morlighem et al., 2017] that compiled all existing bathymetry data around the Greenland. p.6*

We are aware of this new data set which we used when constructed vertical temperature profiles from the reanalysis data. However we derived glacier geometries when this dataset was not yet available and we had no time to repeat this work with the new dataset. We will use it in our future work.

> *l.15:"in the ice sheet" → "in a previous ice sheet"*

the sentence will be change to: "...running the ice sheet model SICOPOLIS by Calov et al. 2018"

> *p.6 l.24: Why not use the mask in Calov et al. (2018)? Combining difference sources for the different datasets might lead to some inconsistencies between the datasets.*

We used the same ice mask. For clarity we would write "defined by the ice mask from SICOPOLIS in Calov et al. 2018"

> *p.6 l.26: Explain that the change in "basal melt" refers to ice shelf basal melt and not grounded ice basal melt. I was initially confused given that the previous paragraph talks about subglacial hydrology.*

Basal melt here is the melt under the grounded ice sheet, that does as well contribute with the surface runoff to the subglacial discharge. The sentence will be changed to: "In our future

scenarios when simulating subglacial discharge we include changes in surface runoff, basal melt (of the grounded ice sheet)..."

> *p.7 Eq.9: To be honest I don't like this flux correction in the SMB. The problem of inconsistent datasets and initialization procedures is a real problem that we are facing as a community, and that deserved better treatment than a simple flux correction. This is calibrated for the initial state, but as the glacier evolves with time it is most likely not to be valid anymore. How does this correction impact the results?*

The need for using of flux correction or similar methods originate from imperfectness of climate and ice sheet models and there is no reason to like it. Eventually, when ice sheet models will be improved, the flux correction will be abandoned as it happened already in the climate modeling community.  However, at present, it is not possible to simulate accurately present-day elevation and spatial extend of GrIS using the SMB obtained from regional climate models. This is why we believe that using of flux correction is  superior  compared to using of a completely unrealistic initial state of GrIS simulated with the realistic SMB, especially, for the purpose of modeling GrIS response to climate change on centennial time scale. The dependence of simulate sea level contribution on the used corrected flux does exist, however, we found it to be not very strong for most of glaciers, by performing experiments with different relaxation times. Such weak dependence can be explained by the fact that for the outlet glaciers (unlike the rest of GrIS), changes in SMB plays only a secondary role in glaciers retreat compare to changes in submarine melt and calving.

> *p.8 l.19: I am confused about this comparison at different depths? Why not use temperature profiles over the entire depth? Also how did you choose these depths? Do they correspond to the depth of warm or cold water? Or the changes in the thermocline? What is the rational for this choice?*

As we explained  above (and we will make it more clear in the revised manuscript) we use the reanalysis data as the fallback option for the fjords for which there are no CTDs available. The reanalysis data are available at the regular grid and at the vertical level 5, 30, 50, 100,200, 400, 700, ???, etc  m. Since most of submarine melting occures below 100 m and typical depth of Greenland fjords is up to 700m  m, we restricted our comparison of (continuous) CTD profiles with the reanalysis data at these three available depths – 200, 400 and 700 m. The main conclusion we made is that when constructing vertical temperature profile using reanalysis data it is better (better agreement with CTD) when we fixed temperature below 400 m rather than interpolate between 400 and 700 m. We will add a corresponding figure to make this part of discussion more clear.

> *p.8 l.23: This is also the case for Jakobshavn (figure 4).*

Yes we agree that also Jakobshavn shows the same feature but this is shown in Figure 4. The actual location of the reanalysis point is only shown in Fig. 6 exemplary for Store Glacier. We will therefore write now:

"Figure 4 and 5 compare the temperature at these depths from reanalysis data with available CTD profiles measured over past several decades exemplary for Jakobshavn-Isbrae and Store Glacier. Since Greenland is surrounded by the continental  shelf with typical depths of 200-400 meters, most of the 700-meter depth points in reanalysis data are located outside the fjords in the deeper ocean, far away from the glacier mouth as shown in Fig. 6 exemplary for Store Glacier"

> *p.8 l.26-32: I have the impression (and this is not very clear in the manuscript) that you don't use the sill depth in the fjords to determine the water properties in front of the glacier. The sills block the warm water at depth, which can significantly impact the water properties. This should be included for the plume model. Why not use that instead of an arbitrary depth of 400 m? Accurately including the fjord properties in important to separate the response due to the trend in climate changes from the impact of local conditions of the glaciers and the fjords.*

Again, we thank the reviewer for this suggestion. We will demonstrate that for temperature profiles derived from reanalysis data, changes according to a (shallow) sill depth do not improve the temperature profile in comparison to terminus-close CTD measurements.

> *p.8 l.29: "larger" → "deeper"*

*Agreed, will be changed accordingly.*

> *p.9 l.10: Again here, why used the temperature at 400 m depth and not the temperature at the grounding line depth? I think the value used should be designed to best represent the conditions in each and every fjord instead of using a generic value systematically applied to all the fjords.*

*Here we only derive a trend at the 400m depth point, from CMIP 5 models, since the continental shelf only allows water masses to pass from 0 to 400 m depth and the deep bottom water controls submarine melting. We add this trend to the total temperature profile (measured and reanalysis) which includes the temperature at the grounding line depth. We will clarify this in the revised version.*

> *p.9 l.16-21: It would be great to see the values of the different results, and especially how the different runs agree with the observations. More details on the choice of runs selected should also be added.*

We demonstrate or results of the spin-up experiments of the present-day tuning in Figure 9. For completeness we now list the values of the 4 dynamcial parameters, beta and fwd range in the SI.

> *p.9 l.22-26: This paragraph is not clear.*

Agreed. We would rewrite the paragraph to:
"Once the four dynamic parameters and the relaxation time scale are set in our pre-calibration, we switch to the coupled glacier-plume model for the spin-up experiment.
In the spin-up experiments the submarine melt rate is now derived by the plume equations which require subglacial discharge and a temperature and salinity profile as input-data. We used monthly subglacial discharge for the year 2000. Vertical temperature and salinity profiles in these experiments were taken from reanalysis data, averaged over the time interval 1990-2010 or from recent CTD data, and held constant over time. Nonetheless, in the spin-up experiments the submarine melt rate isn't necessarily constant since changes in the grounding line depth and shape of a floating tongue (if present) affect the plume equations"

> *p.9 l.27: scaling of what? How is that done?*

That was explained in 2.2 but we will rewrite in brackets :" ( factor in a range from 0.3 to 3 that multiplies the simulated melt rate profile)"

> *p.10 l.6: What is 3.3?*

We forgot the word "section".

> *p.10 l.14-18: I think this could be easily simplifies in saying that you use the volume above flotation.*

Agreed. We will delete the lengthy explanation with equations and added the sentence. "The contributing ice volume V_SLR is determined by the lost ice volume above flotation from each glacier"

> *p.10 l.21: Mention that is the present-day simulated state.*

*We will.*

> *p.10 l.21: It is not clear what you mean by calving ratio.*

*We will explain more carefully in the revised version.*

> *p.10 l.23: The grounding line position is not clear on the figure, the ice front position is. Also most of these glaciers do not have any floating tongue, so it would be better to use the term ice front in this case.*

We will add a close up view of the grounding line position in the SI. Glaciers named as tidewater glaciers as e.g. Helheim still evolve small tongues mostly before the melt season. We will added the sentence:

"Note that we allow for small floating termini, since many tidewater glaciers still evolve them on a seasonal scale and in nature they are also mostly undercut and do not have a pure vertical cliff"

> *p.10 l.21-30: Do you actually want the glaciers to be stable or to be representative of the present-day conditions? Because many of these glaciers are losing mass and retreating today, so how much should a spin-up with present-day conditions lead to stable conditions?*

See our response to the 3$^{rd}$ major comment

> *p.11 l.2: I thought that most of these glaciers did not have floating termini anymore!*

We now address this issue by explaining why we allow for glacier tongues to evolve in our glacier model. (see answer to two comments above)

> *p.11 l.6 "by Enderlin .."*

Will be adapted.

> *p.11 l.17: The numbers you provide do not include the present day changes? This is quite surprising and ends up presenting very low sea level change numbers that are not in good agreement with today's observations. It also questions the initialization procedure of the model, how much can we separate the present state and future changes given that the initial conditions have a lasting effect on the results.*

We are not certain which numbers are meant here by the reviewer and how these numbers can be in agreement (or disagreement) with observations. In our paper we give only the contribution to SLR for the period 2000 – 2100. Our median estimate for the all Greenland glaciers based on upscaling is 50 mm, which is within the previous estimates for the same value. We argue that this number is complimentary to the SLR contribution simulated by a global GrIS model which does not account for ice sheet-ocean interaction (e.g. Calov et al., 2018). The sum of these two separate contributions (see our Discussion) gives ca. 140 mm, which is well within the range of existing estimates (e.g. IPCC, 2013; Fürst et al., 2015; ). At the same time the recent estimates for the total GrIS contribution to SLR around the year 2000 is about 0.2-0.4 mm/a of which only half is attributed to the enhanced solid discharge (Enderlin et al., 2014). These numbers are not negligible but still significantly smaller than the average SLR which we simulated for the entire 21th century. Therefore the assumption we made that glaciers at 2000 were in quasi-equilibrium cannot have significant effect on our estimates for the SLR.

> *p.11 l.26: "excluding" → "separating"*

Will adapt accordingly.

> *p.11 l.27: This is not very clear, try to better separate the numbers for SMB only, elevation feedback, climate change trend, ocean, ... as is done in figure 11.*

*We will clarify the statements.*

> *p.11 l.30: "substantially" → "substantial"*

Will adapt accordingly.

> *p.12 l.13: The potential SLR and grounding line retreat are actually not listed in the tables.*

*We disagree, since they are listed in Table 3 and 4.*

> *p.12 l.15: "uncertainties" → "spread"*

Will adapt accordingly.

> *p.12 l.18: There is only one model used to generate SMB, so where is the spread*

*coming from? It is not clear if is caused only by the different initial conditions used or if here is something else. Also, why is there only one model used to generate SMB and several for the ocean?*

The spread is actually coming from the different initial condition caused by the freshwater depth and beta. We will insert now this explanation " Since there is only one SMB forcing the spread originates from the different initial states cause by the different fwd and beta combination."

*p.13 l.1: "1D line plume model" → "1D plume model" (same in other places in the manuscript). Also "Jenkins (2011)" → "(Jenkins, 2011)"*

Will adapt accordingly throughout the manuscript.

*p.13 l.12: How does that compare to other 2D or 3D models of Jakobshavn [e.g., Muresan et al., 2016; Bondzio et al., 2017]?*

We will compare and discuss our resultsto the mentioned literature .

*p.13 l.19-20: remove*

Will adapt accordingly.

*p.13 l.30-35: Use present tense instead of past tense*

Will adapt accordingly.

*p.14 l.5: What are the numbers for the entire Greenland if you only take the same glaciers as Nick et al. [2013]? How are these numbers impacted by the choice of glacier? So, if you only include a subset of the 10 glaciers used in this study, how does the sea level contribution of Greenland vary? It would be interesting to compute some kind of uncertainty associated with this method.*

We thank the reviewer for this interesting suggestion and will derive an uncertainty estimation.

*p.14 l.7: "our our"*

Will be deleted.

*Fig.2: Is there a white dot in the fjord? It's not very clear. I don't understand the choice or the use of CTD profiles. Why not use all (or a combination of the different) profiles?*

Since the plume equation require the temperature of the ambient water that entrains into the plume, we chose the closest,deep avaible  CTD measurement (closest to the glacier terminus) not CTDs far away. We will improve the Fig. 2.

*"depth of 400 m" → "depth of at least 400 m". "od" → "of"*

Will adapt accordingly.

*Fig.4: same as Fig.3*

Yes, therefore we never use temperature profile from reanalysis data at 700m depth, since the are located outside the continental shelf.

*Fig.6: It would be better to label all the dots (they are only 12). Again here, why use the depth-averaged temperature and not the temperature that most impact the plume model?*

This was done to get a overview of how far the profiles of CTD and reanalysis data are actually of. Nonetheless, we will plot now all the CTD an reanalysis data temperature profile for each glacier in the SI .

*Fig.8: Why present the results from only one ocean model and not from all of them?*

Results of only one model  and only for one location is shown in Fig. 8 just for illustration. The total range of temperature trends derived from different models and for different locations are given in Table 1.

*What is the implication of large discrepancy at 700 m depth between the model and the CTD measurement?*

The likely reason for this discrepancy (actually 1°C error is not large for the GCMs) is that the nearest model grid point with the depth 700 m is located far from the CTD location. This is why, similarly to constructions of the vertical temperature profile from the reanalysis data, where the lowest depth we used was 400 m, to construct ocean warming scenarios we also disregarded levels below 400 m and instead prescribed temperature trend simulated by CMIP5 models at the depth 400 m. Note, that for climate change scenarios we did not use absolute values but only the anomalies simulated by CMIP5 models. These temperature trends for different locations and models are listed in table 1 (SI).

> *Fig.9: Is the observed bedrock directly taken from the BedMachine dataset along thecenterline or is it representative of the entire glacier (of its entire width)? How many sta-*
> *ble states are used for each glacier? I could not find this information in the manuscript.*
> *And as mentioned above, do you really want the initial configuration to be stable or to represent the current state of the glacier? I am not sure "transparent lines" is the appropriate term.*

We will clarify the source from our bedrock. The number of stable states is depicted in the SI, for each stable state one dot is shown. We will ad the number of stable states to our parameter list.

> *Fig.10: "median-range3": repeat the superscript meaning here. Fig.11: "vom" → "from"*

Will adapt accordingly.

> *Fig.12: Try to use the same order as for Fig.11 for the lines. Fig.13: Would be better to repeat the entire caption.*

Will adapt accordingly.

> *Fig.14: What is "ocean temperature trend 1"?*

Listed in Table 1. We will correct in the revised version.

> *Tab.2: I thought that most glaciers in Greenland did not had floating termini any more,so why are there relatively large ratios of melting?*

They do, within the season the can evolve short termini that are after the melt season mostly calved. We will give some literature references for this.

> *Tab.4: It is not clear what the sum of grounding line retreat represent. It is a rather unusual metric.*

We agree with the reviewer and will change the entry in the table to the average grounding line retreat.

---

## Author Comment (AC2) · 6 Sep 2018

**Response to reviewer 2**

We thank the reviewer for his constructive comments. The comments by the reviewer are in indented blocks and italic fonts.

> *This paper investigates, by means of numerical modelling, the evolution of 12 outlet Greenland glaciers in the next century (2100). The employed numerical models are a 1D flowline glacier model and 1D (ocean) plume model, they are coupled together. Two aspects represent important limitations of this work: the use of a 1D glacier model for confined glaciers and the methodology followed in forcing and using the 1D coupled plume model. Some of the assumptions of this work are not properly addressed or discussed, as well as some of the consequences on the obtained results. This paper is clearly written, with the exception of some paragraphs that may lead to some confusion about the experimental setup (e.g. It is not clear if you actually run SICOPOLIS or not. Including a "methods section" may ease the reading).*

We run SICOPOLIS and details to this can be found in our earlier study Calov et al 2018. All coauthors in this current paper contributed also to Calov et al. 2018. For this manuscript, only the output data on subglacial discharge from Calov et al 2018. were used to force the coupled glacier plume model.

**Main comments**

> *On the plume model:*
> *I think that using the coupled 1D plume model is a great improvement. However some experimental choices limit the validity of this improvement.*
> *At page 5 – line 2 is written that "since the plume model in some cases underestimate...*
> *we also scale the simulated melt rate profile by a factor Beta...".*
> *I have some comments on this: the relation between the plume forcings (temperature,salinity, shelf/tongue slope, subglacial discharge, . . .) and melt rate is given by robust physical equations (Jenkins, 2011; Beckmann et al. 2018). I believe that tuning the obtained melt rates with a multiplying factor waste all the efforts made in using (and coupling) the plume model. What is the need of this sophisticated model if then the computed melt rates are scaled to observed melt rates? Then why not using a simple depth dependent parameterization (e.g. Martin et al., 2011)?*

Indeed, Jenkin's model of turbulent plume is based on the first principles and therefore it is expected it provides robust qualitative relationship between submarine melt ,ocean temperature and the slope of glacier front. Whether this model is also quantitatively correct for each Greenland fjord is another issue. The real world is very different from the assumptions behind the linear plume model since during summer season significant amount of melt water is delivered into the fjord through a number subglacial channels. . At present, there is no way to simulate realistically the large ensemble of different plumes, as well as many other processes (tidal circulation in the fjord, undercutting, etc) which may also contribute to submarine melt. To describe this complex reality we proposed to use the Jenkin's linear plume model but with additional correction by parameter beta. Obviously there is no prove that this parameter will stay constant for the next 100 years but still we believe that our approach represent an important improvement  compare to a much simple parameterization (we assume that the reviewer means here the parameterization by Beckmann & Goosse, 2003) since we explicitly account for the dependence of submarine melt on subglacial discharge which is very important factor for the global warming simulations.

> *You tuned the computed plume melt rates on present day observed melt rates. How can you assume that this "present day" scaling will still be valid in 50/100 years? This choice is crucial in terms of providing a robust basal forcing for the glaciers evolution. I think that this assumption should be discussed.*

As we explained above, there is no reason to expect that a very simple Jenkin's linear plume model can accurately described complex reality of Greenland fjords even at present and nd there is reason to expect that correction parameter beta will remain constant over 50 or 100 years. The reviewer is absolutely right (see Fig. 15): the choice of melt and calving parameters is the source of the largest uncertainties in glaciers contribution to future SLR and one of the aim of our paper is to report this problem. How to fix this problem is beyond the scope of this paper.

> *Given the inherent large uncertainties in forcing conditions (both in CTD and in re-analysis, page 8 line 3) what about forcing the plume model with a range of plausible temperature and salinity (from CTD and/or reanalysis) and with a range of subglacial discharges instead of tuning the computed melt rate?*

Obviously, uncertainties in temperature profiles and subglacial discharge also contribute to the SLR uncertainties but very unlikely they contribute to the discrepancy between melt rate simulated by Jenkin's model an real one. Indeed, typical uncertainties in water temperature of $1^\circ$C will result in 20% uncertainties in melt rate. The uncertainties of 50% in subglacial discharge results only in 15% uncertainties in melt rate (due to cubic root dependence). At the same time, as we show in Beckmann et al (2018), melt rate simulated by linear plume model can deviate from observed one by factor 2-3.

> *It is not clear why you decide to use reanalysis data at 200, 400 and 700 meters of depth instead of using continous vertical profiles. Moreover, for future simulations you say: "...closest 400m-depth-point neighbor...". Is this motivated by line 29 to 31 at page C28? I understand this choice but I believe that you shold explain this better, clearly motivating also at page 9.*

The first reviewer has a similar question which is addressed in our response. Obviously, this part of our paper was not clear enough and will improve it in the revised version.

> *On the glacier model:*
> *I get why you decide to use a 1D flowline model: however I think that the limitations related to this approach (neglect of processes at the lateral boundaries and of buttressing, which play a crucial role in the evolution of ice masses) are not properly tackled and are mostly addressed by saying that 1D models are the only one available for this kind of study. This is probably right if you want to model 12 (or more) glaciers at the time, but for single glacier the last few years have seen important improvements in modelling alternatives that have produced results for some glaciers that are also modelled in this work (Chaulet et al., 2012; Seddik et al., 2012; Muresan et al., 2016; Peano et al.,2017; Goelzer et al., 2017). I think that the discussion about 1D model limitations should be expanded.*

We agree with the reviewer and will discuss more in depth the limitation of a 1D glacier model. Also will we introduce more work from other authors on 3d models on glaciers.

**Specific comments**

> *Page 1 – line 15: "factor analysis". With factor analysis it is usally meant a statistical method like the Empirical Orthogonal Functions (EOFs), in your work you just exclude*
> *(one at the time) the different forcings, I would not strictly define this procedure as a factor analysis.*

We will change "factor-analysis" to " sensitivity analysis of the forcing-factors".

*Page 2 – line 5: instead of "global" I would use "atmospheric"*

We will adapt accordingly.

*Page 2 – line 4 to 8: I found this paragraph ok, but I would rearrange it a little bit putting the described processes in the same order you are introducing them.*

We will improve this paragraph.

*Page 2 – line 6: "marine terminating" instead of "marine- terminating"*

We will adapt accordingly.

*Page 2 – line 16: "In order to..." this should be a new paragraph*

We will adapt accordingly.

*Page 2 – line 32: "that" is repeated two times*
We will delete the second 'that'.

*Page 2 – line 35: "Since we are.." this should be a new paragraph*

*We will insert a new paragraph.*

*Page 3 – line 1: I would say that the main (and only) improvement consists in using the coupled plume model. I consider the fact of studying more glaciers just as an "extension" of Nick et al. 2013 work.*
*Moreover, from the scaling perspective, are we sure that the considered glaciers are really representative of all the Greenland*
*glaciers? especially given their variety in terms of glaciers and of confining fjords geometries/conditions.*

Agreed, we will change the part to:, "...we followed an approach similar to (Nick et al. 2013) but with one notable improvement.: For calculations of the vertically distributed submarine melt, we used a turbulent plume parameterization following (Jenkins et al. 2011)."
We considered 12 glaciers as in improvement compared to Nick et al. and selected them since they represent different ice flow regimes and different environmental conditions. Nonetheless a greater sample size could decrease always uncertainty, which is a factor we will add in the discussion.

*Page 3 – line 4: ok, but submarine melt rate depends also on the geometrical features*
*of the tongue (shape, slope,…)*

Agreed, we will change the sentence to. "According to this parameterization, the submarine melt rate depends not only on ambient water temperature in fjords but also on seasonally varying subglacial discharge, shape and angle of the glacier tongue."

*Page 3 – line 9 to 11: Maybe you can think about shortly describing how the scaling works.*

Agreed we will give a short explanation of the scaling here.

*Page 5 – line 1 to 5: I would expand the plume paragraph since it is the real innovative part of this study. Maybe a short introduction of the basic physics and*

> *equations. Otherwise is not clear what do you mean with the E entraiment parameter unless looking at Beckmann et al. (2018) (or already knowing what you are talking about).*

We agree with the reviewer and extended now the whole paragraph and add the equation for the plume model.

> *Page 5 – line 17: "to the vertical mass balance term B", add the equation number*

We will add the equation number.

> *Page 5 – line 18 to 20: I imagine that when the plume detaches the melt rate is set to zero but this is not written explicitly. Is this the case?*

We thank the reviewer for spotting the lack of information.
The plume never detaches from the glacier in the model, it only ceases by slowing down the velocity to zero. When this happens, the melt rate is set to a minimum melt rate to ensure background melting. We will describe this in the revised version as the following: ". If the plume already ceases before reaching the calving front x_cf, we numerically introduce a minimal background melting determined by the last melt rate value before the plume ceased".

> *Page 5 – line 21: this part confused me. "...off-line using the ice sheet model" which*
> *one? This is the first time that you mention the use of an ice sheet model. Later it appears that it is SICOPOLIS.(see comment to page 6 – line 15 to 25)*

Yes, we used SICOPOLIS output data which is described detailed in Carlov et al. 2018. We will change the sentence to:
"Subglacial discharge Q for each glacier was provided off-line by the data output from simulations of the ice sheet model SICOPOLIS (Calov et al. 2018) with explicit treatment of basal hydrology (Section 3.3), then applied to the line plume in distributed form q= Q(W)$^{-1}$."

> *Page 6 – line 1: "did we" "we did". Could you explain better in what this upscaling consists and how it works?*

Agreed, we will describe better the upscaling.

> *Page 6 – line 2: it would add more clarity defining what is meant with "melting to calving ratio"*

Ageed, we will insert" (Grounding mass flux lost by submarine melting divided by mass loss of calving)"

> *Page 6 – line 12: just a detail: I would number the figures in the order of appeareance in the manuscript.*

Agreed,we will adapt accordingly.

> *Page 6 – line 15 to 25: From here it looks you actually run the ice sheet model, is this correct? (look comment to page 10 – line 6).I suggest to introduce explicitly the fact that you have run SICOPOLIS.*

Yes, we run Sicopolis earlier  and details can be found in Calov at el 2018. However, after the SICOPOLIS simulation we used the subglacial output data to force the coupled glacier plume model off-line. The sentence will be changed to:
 "The former two sources are computed directly from the ice sheet model SICOPOLIS by (Calov et al. 2018)."

> *Page 6 – line 23,24: "...is assigned to the closest glacier within a maximum of 50 km". This is an important approximation since is related to the plume forcing,*

*however is not properly discussed, expecially in terms of uncertainty in the obtained results.*

We will discuss this more in the revised version

*Page 6 – line 27,28: "neglect the effect of grounding line retreat".As above, this represents another important assumption but it is not properly discussed.*

True. We will add :
"For neighboring glaciers with a competing catchment area, a strong grounding line retreat may strongly affect the distribution of the subglacial discharge between those glaciers (water piracy, Lindbäck et al 2015). This affect is not included in this study."

*Page 8 – line 12: "...presence of sills in the fjord...in the vicinity of the glacier front." I would explain why is that after this line, instead than explaining it later for the continental shelf (at page 8 – line 24 to 30).*

We will adapt accordingly.

*Page 9 – line 16: could you provide a table with the prescribed submarine melt rate and the range of values for the dynamic parameters? (maybe in the supplementary)*

Agreed, we will provide such a table in the SI.

*Page 9 – line 25: with "...only factors.." do you mean that since temperature and salinity are "held constant" (thus not changing) their contribution in impacting melt rates is constant in comparison to the impacts due to a varying grounding line depth and tongue shape/slope? I suggest to reformulate this paragraph*

Yes, we just wanted to point out that although the temperature-salinity profile is held constant the melt rate isn't necessarily constant due to the glacier's changing geometry. We will now write "Nonetheless, in the spin-up experiments, the submarine melt rate isn't necessarily constant since changes in the grounding line depth and shape of a floating tongue (if any) affect the plume equations. "

*Page 9 – line 29 "...is close to equilibrium state.." what do you mean with equilibrium?*
*Later you speak about stable state. Do you mean steady? I would argue that currently Greenland glaciers are definitely not in a steady condition.*

The same issue was addressed by reviewer 1 ( major comments) and we defined our definition on the present-state more clearly. This will be added in the revised manuscript.

*Page 10 – line 5: "...each glacier 3.4..." something is missing between glacier and 3.4*

We will delete 3.4.

*Page 10 – line 6: "...glacier individually 3.3..." something is missing between individually and 3.3*

We thank the reviewer for spotting the mistake we will insert the word "section".

*Page 10 – line 6: Here it is not clear if you took the data from Calov et al. 2018 or if you actually run the model*

This paper and Calov et al. (2018) are closely related. They originate from the same project and are written essentially by the same group of authors. Calov et al. (2018) describes the model of Greenland glacier system, experimental setup and results of several climate change experiments. In Calov et al. (section 5) we also described how we computed time-dependent subglacial discharge for individual Greenland glaciers using SICOPOLIS and MAR output, and the basal hydrology model HYDRO. In the current work, we used this time-dependent discharge as the forcing for modeling of 12 selected glaciers. We will clarify this issue in the revised manuscript.

> *Page 10 – line 22,24: this part about the interplay between melting, calving and bedrock is interesting. I would add few more details.*

Agreed, we will add a broader discussion in the revised version.

> *Page 11 – line 6: a space is missing before "Enderlin"*

Will insert space.

> *Page 11 – line 15: "model versions" do you mean the the spin-up ensemble?*

Yes, we will now write.:"After obtaining the present-day state, we then ran the spin-up ensemble with all valid beta/fwd combination ..."

> *Page 11 – line 16: why not changing also the subglacial discharge? It is such an important forcing for the plume and comes from several approximations (fixed grounding line and closest neighboring approach).*

We fully agree with the reviewer that in this study we explore only a fraction of uncertainty sources. In particular subglacial discharge as well as SMB also depends on the choice of the regional climate model and the global climate models which has been used to provide boundary conditions for the regional model. However, we believe that Fig. 15 already provides a very important inside into the major source of uncertainties in simulated glaciers contribution to SLR. Namely, it shows that the uncertainties in the choice of model parameters is likely to be the largest source of the SLR uncertainty. Thus to considerably narrow down these uncertainties, the glacier model parameters have to be better constrained.

> *Page 11 – line 17: at page 10 (line 8 to 10) is said that also the unforced model drift is calculated. Then this drift is removed by subtracting it from calculated values. This implies that a linear behaviour for glaciers is assumed. I think that this should be properly discussed.*

Of course it is known that glaciers response to climate change is nonlinear and we do not assume such linearity. Our modeling approach is based on the assumption that glaciers were in equilibrium at the year 2000. However, to ensure that all glaciers are in the perfect equilibrium with the 2000 year forcing would be required to perform infinitive number of infinitively long spin-up experiments which is not possible even with fast model. This is why we apply as additional constrain, namely, we excluded all model realizations with positive (mass gain) trend and require that the simulated negative trend is significantly smaller than simulated SLR response to climate change scenario. Still, we have to tolerate non-negligible drift in the control runs - otherwise we will be left with to few accepted model realizations. This is why we decided to exclude such drift from the forced run which we believe is still better to bot do it. We will discuss this issue in more details in the revised manuscript.

> *Page 11 – line 25-27: as above, this implies linearity but glaciers are definitely not*

*linear systems. This issue is just slightly addressed at page 12 – line 4. Page 12 – line*

We did not assume that glaciers are linear systems. As we explained in the response to the first reviewer, the drift is rather small since we only accepted such model versions in which drift is smaller than simulated SLR in the forced experiments. Of course, zero drift in unforced experiment would be preferable, but this cannot be achieved with the finite computational resources. Therefore we are left with two to options: (i) to leave forced experiments as they are or (i) to exclude unforced drift from the forced experiments. Both options are imperfect but we prefer the second one.

*21,22: you attribute the source of uncertainty to Beta, this comes from the fact that Beta is responsible for the imposed melt rate (through the tuning procedure). However Beta is just a model parameter, I think that avoiding the use of Beta (as suggested in he main comments) could also improve this part of the work, it will allow you to relate uncertainties to physical quantities.*

As we explained above, there is no physical reasons why the linear plume model should produce correct results with beta=1. See also Beckmann et al. (2018).

*Page 13 – line 18 to 20: something is wrong here, an entire sentence is repeated.*

The repetition will be deleted.

*Page 13 – line 33: same as above. Your results are not affected by CTD/reanalysys temperature and salinity because the Beta tuning incorporates all the uncertainties.*

We agree with the reviewer that colder temperatures in the reanalysis data set in some (but not all cases) can be balanced by a higher beta. We will provide the new table showing which beta values were used for CTD and reanalysis data set.

*Page 13 – line 35: "...observational constraints on submarine melt..." as explained in the main comments I think that we should rely on melting formulation as less as possible dependent from a tuning on observations, especially for future projections.*

We agree that this would be a nice idea but not at the present level of ice sheet-ocean interaction. As we showed in Beckmann et al 2018, Jenkin's linear plume model does not produce observed submarine melt and therefore should be corrected. Even with this correction believe that our approach is more physically based and therefore more trustworthy than those used in previous studies.

*Page 14 – line 4: "and" repeated two times*

We will delete.
*Page 14 – line 7: "our" repeated two times*

We will delete.

*Figure 3(a): I think that using white dots is a bit unfortunate, also the red star is not very visible.*

Agreed, we will improve Figure 3 for more visible CTD location.

*Figure 11: "from" instead of "vom"*
We will adapt accordingly.

---

## Author Response (AR1)

We thank the reviewers for the constructive reviews and suggestions. The comments by the reviewers are in indented blocks and italic fonts. Our response follows each comment and changes in the manuscript are in guotation marks.

**Response to Reviewer 1**

**Major comments**

As mentioned by the authors, scaling sea level rise at the scale of Greenland using results from only a few glaciers is highly speculative. However, this is still what is done in this paper without assessing the uncertainty of such a scaling. It would be important to quantify the uncertainty in the scaling by comparing results obtained with a subset of models. Also, do you think that 12 glaciers are representative of Greenland? They are many kind of different glaciers, with or without marine terminating fronts, with or without ice shelves, with completely different geometries and fjord conditions; some glaciers are mostly impacted by changes in ice front position, or subglacial hydrology(with different types of subglacial hydrology regimes, ...)? So is it reasonable to use such a small sampling of glaciers and consider that their behavior is representative of the 200 glaciers of the Greenland ice sheet?

We agree with the reviewer that the accurate estimate of the contribution of Greenland outlet glaciers to future sea level rise will only be possible when all 200+ glaciers will be accurately modeled. The Fig.14 in our paper, first of all presents the test of the "scaling up technique" used by Nick et al. (2013) who estimated the total contribution of outlet Greenland glaciers by using results of simulations made only for four major Greenland glaciers. Note that Nick et al. (2013) did not tested whether a correlation between present-day discharge and future sea level rise even exists. Here we used 12 glaciers which differ significantly by discharge and location. We do not claim that they properly represent all Greenland glaciers but, still, this is an obvious step forward compare to the previous study. We found (Fig. 14) that some correlation between present-day discharge and the future contribution to sea level rise does exist . Since we consider only a small subset of Greenland glaciers and only some sources of uncertainties, we do not think that the detailed uncertainties analysis of the relationship between discharge and sea level rise would be very helpful. However, we demonstrated how the uncertainty ranges by the choice of glaciers in the supplementary part Figure S7. Determining SLR from these 4 different glaciers leads a SLR range of (10.3-16.8 cm), which we ist no mentioned in our discussion.

I don't understand why the water conditions at 400 m depth are used in the plume model for all the glaciers. All the fjords and glaciers have very different geometry conditions (with sills, ...) that should be taken into account in order to have the right conditions for the plume model. It looks like the authors are trying to set-up the initial conditions so that the glaciers are in steady state.

This is a misunderstanding and we no improved this part of the manuscript (Section 3.4) to make more clear how we constructed the T-S profile and why we use "the water conditions at 400 m depth". Firstly, it is important to note that for all 12 glaciers used in this study, there are at least some CTD profiles from the adjacent fjords and, in spite of some problems with CTD profiles (discussed in the manuscript), the CTD profiles are always our first choice to force the plume model. However, CTD profiles are not yet available for all 200+ Greenland glaciers. This is is why (in addition to CTD profile) we tested whether T-S profiles in Greenland fjords can be constructed using the results of the ocean reanalysis project. Since the reanalysis data are only available for the open ocean, we used the nearest to the fjord's mouth reanalysis grid cells with the depth 200, 400 and 700 m (these are the top three vertical levels in the reanalysis data set). By comparing reanalysis and CTD data are in reasonable agreement while for the 700 m depth they are completely off, which is explained by the fact the 700m depth-points are always located outside the continental shelf .Therefore they are not appropriate to produce vertical temperature profile in the fjords . Therefore instead of interpolated between 400 and 700 m values, we choose to

prescribed below 400 m as constant temperature equal to temperature at the depth 400 m in the reanalysis data. We then compare submarine melt computed using CTD profiles with those have been computed using temperature and salinity profiles from 200 and 400 m depths in the reanalysis data. We found that results are in reasonable agreement. Therefore we recommend as a temporal option (before better data will be available) to use the nearest gridcells with depth 200 and 400 m to construct T and S profile from the reanalysis data for the fjords for which CTD data are not yet available. Moreover, we now show -in the new version of the manuscript- that if we construct temperature files from reanalysis data and consider sill (Fig. S2, supplementary information), the discrepancy grows bigger between CTD measurement close to the glacier front and to the reconstructed TS profile (Fig, S3 supplementary Information) four our selected glaciers.

Many glaciers experienced large changes over the past couple of decades and they are therefore not in steady-state. I think it is more important to have initial conditions close present state than close to a steady-state, especially as initial conditions impact the system for a very long time (much longer than the simulation time in this paper). Furthermore, it is difficult to add the present trend to the simulated changes as glaciers are not exactly linear systems.

The meaning of the "present day" should be properly defined – otherwise it causes confusion. We did not assume that the glacier are in the equilibrium state at present, i.e. in the year 2018. In our paper under "present day" we mean the years 2000 which we use as the starting time for all our forced simulations. The choice of the year 2000 is motivated by the fact that the mass loss of GrIS during the last decade of 20the century was rather small (ca. 0.1 mm/yr) compare to that has been observed in the 21st century. This justify our assumption about guasieguilibrium state of Greenland glaciers at the beginning of experiments. Some inconsistency arises from the fact that the database (BedmACHInev2) we used to initialized the glaciers at the year 2000 are actually based on the measurements made in 2008/2009. However even the total contribution of GrIS to global seal level rise during the first 8 years of the 21st century was only about 3 mm and glaciers contributed not more than half of that. This is of course a very small number compared too our total estimate of 50 mm of glaciers contribution during the entire 21st century. For this reason we see no need in adding "present day" trend to the results we obtained. To the contrary, we extracted from the results a very small trend diagnosed in the control (unforced) run. We now state in section 4.1. (page 12): "We chose the year 2000 as the guasi-equilibrium initial state for "future" climate change simulations since the mass loss of GrIS during the last decade of 20the century was rather small (ca. 0.1 mm/yr in sea level equivalent) compare to that has been observed in the 21st century ((Vaughan et al., 2013)."

**I don't understand why only one atmosphere model is used for the future forcing while several ocean models are used.**

It is well-known that for the entire GrIS contribution to sea level rise, climate change scenarios (both in term of GHGs concentration and the model output) are the major source of uncertainties. In this this study we decided to concentrate on the new issue, namely glacier-ocean interaction. Therefore, we used output of only one regional climate model for a single climate change scenario (RCP8.5) and concentrated on the uncertainties related to parameterizations of submarine melting and calving. We agree with the reviewer that applying another atmospheric model would introduce additional uncertainties in sea level rise via different smb and subglacial discharge. We would like to address this in future work.

There is also no clear distinction between the spread in results caused by the different climate scenarios and the different initial states, and their relative importance for the different glaciers. Is it more important to improve the external forcings (and which one) or to improve the initial conditions to reduce the uncertainty in future glacier's evolution?

We do not agree with the reviewer here. In Figure 15 we demonstrate the spread of results for each single forcing scenario and therefore attribute the spread to the different initial states, thus

beta, and fwd. For clarity we state now in the section future results: "We attribute the major source of uncertainty to the different combinations of the model parameters fwd and beta. "

It seems like the authors read a coupled of references [Nick et al., 2013; Goelzer et al., 2013], and keep using them all over the manuscript. They are also many regional models (that are not 1D) that should be used to compare the results of this study.

Citing of Nick and Goelzer is natural since we used a similar approach and compare our results with these two studies. Of course, we are aware about regional Greenland ice sheet modeling and, although in most cases it is difficult to compare directly our results with regional modeling, , we included a comparison to other regional models for a broader discussion in the introduction aswell as in the discussion part.

Finally, I must say that I am a bit tired of seeing studies based on flowline models in 2018. I agree that such studies are still very useful to investigate new processes for example, but they should not be used to do future projections of ice sheets, given the importance of buttressing, lateral effects, complex topography ... when 2D regional models can be run at high resolution and provide more accurate results.

We respect the reviewer's opinion on the issue which models should be used for future projections of ice sheets. However, the focus of our study is not on future projection of the Greenland Ice Sheet but rather on the response of outlet marine-terminated glaciers to climate change and, primarily, on the analysis of uncertainties related to two poorly constrained processes: submarine melt and calving. We appreciate the importance of buttressing, lateral effects and complex topography which unavoidably are treated in a rather simplistic way in the 1-D model. However, we doubt whether at present 2D models can really provide "more accurate results" since accurate modeling of marine terminated glaciers would require as input accurate knowledge of present and future (i) fjord bathymetry, (ii) temporal variability of the 3D fields of temperature, salinity and velocity in the fjord, and (iii) the spatial-temporal distribution of subglacial discharge of melt water into the fjords. Even at present all these characteristics are not accurately known for Greenland glacier and in most cases they are not known at all. At the same time we agree that the paper now benefits from the discussion of model limitations and future perspectives.

**Line by line comments**

*p.1 l.6: crudely* → *simplistic (models use simple parameterizations because the processes remain unknown as mentioned at the beginning of the sentence)* Agreed, we replaced "crudely" with "simplistically"

*p.1 l.10: Is the regional climate model used only for the SMB or also for other properties?*

The other property is surface runoff as mentioned in the sentence." ...forcing the model with changes in surface mass balance and surface runoff...,"

p.1 l.12 (and l.14 and l.15): use present tense instead of past in the abstract: used  $\rightarrow$  used.

Agreed, we adapted the tense.

p.1 I.22: the scaling is quite speculative. What happens if you do the scaling with a smaller set of glaciers? What is the uncertainty in this scaling?

We thank the reviewer for this interesting suggestion. As mentioned above, we derived now such an uncertainty as proposed by taking 4 (same number as Nick et al) different glaciers and determine a minimum and maximum SLR number (Fig. S7). The range derived from this method I mentioned now in the discussion part of the manuscript.

p.1 I.19 and I.23: If I understand correctly, the numbers given here do not include the current trend in mass loss from the Greenland ice sheet (13.8 cm in the conclusions). This is rather confusing and provides numbers smaller than expected.

The numbers given in the abstract (14 mm for twelve glaciers and 50 mm obtained by scaling up for all glaciers) can be considered as complimentary to the numbers order of 100 mm computed in the coarse-resolution GrIS models (e.g. Calov et al., 2018) because in the latter the mass loss of the GrIS is mostly controlled by changes of SMB, while mass loss of outlet glaciers is primarily controlled by increased submarine melt. We now write: "... we estimate the mid-range contribution of all Greenland glaciers to 21st-century sea level rise to be approximately 50mm. This number adds to SLR derived from a stand-alone, coarse resolution ice sheet model and thus increases SLR by over 50 \%."

**p.2 I.4: "Two processes are largely responsible": What are the other processes that account for mass loss to a lesser extent?**

We added the percentage in brackets: (60 %) surface melting and (40 %) dynamical processes.

**p.2 I.6: "marine-terminating" $\rightarrow$ "marine terminating"**

Agreed.

*p.2 I.8: It is not just warming of the ocean, but also changed in the circulation.* We agree, the circulation changes led to the warming of the ocean. The sentence now reads: "...which can in turn be attributed to a warming of the subpolar North Atlantic ocean, induced by circulation changes, and increased subglacial discharge"

p.2 I.9: I doubt that the lower contribution in Fettweis et al. [2013] is 0 cm, it should be 50 mm (9 ± 4 cm). This seems rather contradictory with the actual contribution.
 We thank the reviewer for spotting this mistake. 0 cm was cited in Fettweis et al. 2013 from other studies. We changed the number to 50 mm.

p.2 I.13: Adding references to papers that detail the limitations of modeling of the Greenland ice sheet would be appropriate (e.g. Goelzer et al. [2017]; Khan et al. [2014]).

We thank the reviewer for this suggestion. We cited now: "The contribution of the second process remains highly uncertain because processes related to the response of marine terminated Greenland glaciers are still not properly represented in the contemporary GrIS models (Straneo and Heimbach, 2013; Khan et al., 2014; Goelzer et al., 2017)."the above-mentioned literature.

p.2 I.18-20: I think this is disregarding all the efforts made to improve continental scale models, as some models now have a resolution of about 1 km in marine terminating glaciers [Goelzer et al., 2018]. This is also a bit oversimplifying the problem: the limitations of numerical models are not just resolution, there is also limited observations, external forcings not appropriate, ... So this part of the introduction has to be more balanced.

Yes, as cited above we now write:"The contribution of the second process remains highly uncertain because processes related to the response of marine terminated Greenland glaciers are still not properly represented in the contemporary GrIS models (Straneo and Heimbach, 2013; Khan et al., 2014; Goelzer et al., 2017).

p.2 I.21-25: Following along the same line, I think jumping from continental scale 3D models to 1D flowline models is a bit reductive, as they are many things in between. Several regional models with 2D or 3D models are starting to show interesting results [Muresan et al., 2016; Bondzio et al., 2017]. Some studies even included representation of ocean with a plume model [Vallot et al., 2018]. So I think the introduction should be improved and not just reduced to Goelzer et al. [2013] and Nick et al. [2013]. We thank the reviewer for pointing out this different studies. We mention now these and several other applications of 3D models to study response on regional and shorter time scales. As far as a very interesting paper by Vallot et al. (2018) nicely illustrates that high-resolution and physically based modeling of glacier-ocean interaction is already possible but absolutely impractical for the study of glacier response to global warming. While Vallot et al. (2018) studied only one melt season with the glacier model., they were able to run plume model only for 10 minutes and only for a small fraction of the ice front. Therefore we cannot see an alternative to highly simplified parameterization of the glacier-ocean interaction when the centennial time scale response is concerned. However, we now mention the other studies in the introduction (p.2-3):"For regional settings, 3D models with a simple ocean melting parameterization were applied to study the historical (last 20 -30 years) retreat of Jakobshaven Isbrae (Muresan and Khan,2016; Bondzio et al., 2017). A more advanced treatment of submarine melt rate was done by Vallot et al.(2018). They coupled a plume model based on the Navier-Stokes equations with a full-Stokes ice sheet model. With this off-line coupling, glacier dynamics for one melt season were simulated for Kronebreen Glacier in Svalbard.

p.2 I.32: "that that"

Deleted.

p.3 I.1: You just mentioned that the approach from Nick et al. [2013] is not appropriate, but you follow the same one, just with slightly more glaciers. I am not sure I understand the logic here.

We did not state that the approach from Nick et al "is not appropriate". This study has obvious limitations, such as using of only four largest glaciers to project the the entire 200+ glaciers contribution to sea level rise as well as a very simplistic parameterization for the submarine melt. We wrote that "we followed an approach similar to Nick et al. (2013) but with several notable improvements" and our major improvement is using of more glaciers and more physically based parameterization for submarine melt.

*p.3 l.28: "with 3D ice sheet model"*  $\rightarrow$  *"with 3D ice sheet models"* Agreed, changed in the revised version.

*p.3 l.30: "we used instead"*  $\rightarrow$  *"we use instead"* Agreed, changed in the revised version.

p.3 I.31: I agree that continental scale Greenland models are not the best tool to study these processes, but why not use 2D basin models that would at least include lateral deformations and buttressing is important to correctly capture the behavior of narrow outlet glaciers terminating in fjords.

We thank the reviewer for his suggestion but we want to point out that 1D flowline models include lateral deformation and buttressing in a simplistic manner. We , however mention the limitation of the 1D model in the discussion: "Additionally, the 1D flowline model treat lateral processes in a simplified manner, so that more complex bedrock geometries (e.g. branching of glaciers, individual sills, unsymmetrical valley forms) are poorly represented in these estimations."

**p.4 Eq.2: Can you explain the choice made to incorporate the lateral stress?**

The lateral stress term is necessary because the glaciers we considered for this study, like most Greenland marine-terminating glaciers, narrow-down toward their terminus (width of the order of 5 km, besides Petermann), with velocity of the order of 1000 meters per year (10 000 m/a for Jakobshavn Isbrae, according to present observations), making it impossible to neglect lateral drag. The stress term was derived e.g. by Veen and Whillans (1996), and used by various authors since (e.g. Nick et al., 2013; Enderlin et al., 2013; Schoof et al, 2017).We now write in the manuscript: "The lateral stress term likewise used by e.g. Nick

et al. (2013); Enderlin and Howat (2013); Schoof et al. (2017), and originally derived by Van Der Veen and Whillans (1996),

is necessary to account for lateral resistance in fast-flowing, laterally-confined glaciers typical for Greenland."

**p.4 I.9: Where does the basal sliding coefficient come from?**

The basal sliding coefficient (equation 2) was determined, along with other uncertain parameters, from calibration to present-day state. We now give a list ob basal sliding coefficient and other dynamical parameters in the supporting information (table S2).

**p.4 Eq.5: How different is this from simply applying water pressure at the front?**

It is not and we thank the reviewer for spotting the lack of explanation here. We now introduce the description of equation 5 with "while at the calving front x cf balancing the longitudinal stress with the hydrostatic sea water pressure and incorporating the flow law of ice yields longitudinal stretching".

**p.4 I.21: So is there a point exactly at the grounding line position? This should be better explained. Also, how is treated the stretching of the grid, in particular the variables assigned to the new grid points?**

Yes, there is. We now write: "Grid stretching is performed so that there is always a cell edge at the interpolated grounding line position. The new calving front position is determined so that the total glacier volume is not modified by interpolation. For every new point in the interior, model variables are interpolated from previous grid. The first grid point at the ice divide remains unchanged. If ice grid points on the new grid lie outside the ice domain on the previous grid, as it is typically the case for the last cell before the calving front, ice thickness from the last grid cell is extended.

**p.4 I.27: How about the ice front? Does it evolve with time? And following what criteria? You need to describe the subgrid-scale treatment of the ice front.**

We agree with the reviewer and now describe the subgrid-scale and ice-front treatment in section 2.3 (Coupling between glacier and plume model) as follows:

"If there is no floating tongue, submarine melting is applied to the last grounded cell, otherwise it is applied starting from the first floating cell.

Thus the submarine melt rate reduces the thickness of the glacier cell. A reduced thickness at the first floating cell or last grounded cell leads to grounding line retreat since the grounding line position is determined by interpolation of the ice thickness above flotation at each time step. Thinning the last floating cell leads to calving front retreat by either melting the total cell or by calving, which increases with thinning. "

**p.5 I.2: A quick explanation of the plume model in a few sentences should be added.**

Agreed, for completeness we added and described now the equations of plume model.

p.5 l.8: How is a vertical profile of melt applied to a 1D model, in which there is basically no vertical dimension? So what values is used for the melt (maximum, average, ...)?

The cumulative melt rate is calculated as a volume flux and added to the mass balance term. The integral (cumulative melt rate) is partitioned over various glacier

cells (or only one cell ) in the case of a tidewater glacier. This total submarine melt rate, in a cell by cell basis, is substitute as the submarine melt rate M per units of length for each glacier cell. If

there is no floating tongue, submarine melting is applied to the last grounded cell, otherwise it is applied starting from the

first floating cell. We describe now in more detail with the accompanying equations the treatment of submarine melt rate (page 7 until line 20).

**p.5 I.12: What happens above the plume? Zero melt?**

We thank the reviewer for spotting this lack of information.

Above the plume, so if the plume ceases, we set the melt rate to a minimum background melt which is given by the last melt value of the ceasing plume. We added the important information in the revised version of the paper: "If the plume already ceases before reaching the calving front x cf, we numerically introduce a minimal background melting determined by the last melt rate value before the plume ceased."

p.5 I.17: I don't understand "added to the vertical mass balance term B". Is the melt applied to retreat the ice front? Or just to thin the ice close to the ice front? This melt should cause ice front retreat.

We thank the reviewer for mentioning this unclarity. The melt is applied to thin the front, and does cause retreat if the cell is totally melted. Also thinning subsequently leads to calving, which then cause the ice front to retreat as well.

We explicitly describe the ice front treatment of the glacier model in the revised version of the paper.:"Thus the submarine melt rate reduces the thickness of the glacier cell. A reduced thickness at the first floating cell or last grounded cell leads to grounding line retreat since the grounding line position is determined by interpolation of the ice thickness above flotation at each time step. Thinning the last floating cell leads to calving front retreat by either melting the total cell or by calving, which increases with thinning."

p.6 l.1: "Also, did we include"  $\rightarrow$  "We also included"

Adapted.

p.6 I.9 "BedmACHINEev2"  $\rightarrow$  "BedMarchine v2". Also there is new version [Morlighem et al., 2017] that compiled all existing bathymetry data around the Greenland. p.6

We are aware of this new data set which we used when constructed vertical temperature profiles from the reanalysis data. However we derived glacier geometries when this dataset was not yet available and we had no time to repeat this work with the new dataset. We will use it in our future work.

*I.15:"in the ice sheet"*  $\rightarrow$  *"in a previous ice sheet"* The sentence is changed to: "The former two sources are computed directly by the ice sheet model SICOPOLIS (Calov et al., 2018)."

p.6 I.24: Why not use the mask in Calov et al. (2018)? Combining difference sources for the different datasets might lead to some inconsistencies between the datasets.

We used the same ice mask as it is the model out put from Calob et al. (2018). For clarity we now write: "The entire basal water flux (runoff, basal melt, and water from the temperate layer) is routed by the hydraulic potential using a multi-flow direction flux routing algorithm, as described in (Calov et al., 2018). All water transfer is assumed to be instantaneous. Water that passes through the boundary of prescribed SICOPOLIS ice mask is assigned to the closest glacier within a maximum distance of 50 km.

p.6 I.26: Explain that the change in "basal melt" refers to ice shelf basal melt and not grounded ice basal melt. I was initially confused given that the previous paragraph talks about subglacial hydrology.

Basal melt here is the melt under the grounded ice sheet, that does as well contribute with the surface runoff to the subglacial discharge. For clarity we now write: "Subglacial discharge represents the sum of basal melt ( melt under the grounded ice sheet), water drainage from the temperate layer and surface runoff".

p.7 Eq.9: To be honest I don't like this flux correction in the SMB. The problem of inconsistent datasets and initialization procedures is a real problem that we are facing as a community, and that deserved better treatment than a simple flux correction. This is calibrated for the initial state, but as the glacier evolves with time it is most likely not to be valid anymore. How does this correction impact the results?

The need for using of flux correction or similar methods originate from imperfectness of climate and ice sheet models and there is no reason to like it. Eventually, when ice sheet models will be improved, the flux correction will be abandoned as it happened already in the climate modeling community. However, at present, it is not possible to simulate accurately present-day elevation and spatial extend of GrIS using the SMB obtained from regional climate models. This is why we believe that using of flux correction is superior compared to using of a completely unrealistic initial state of GrIS simulated with the realistic SMB, especially, for the purpose of modeling GrIS response to climate change on centennial time scale. The dependence of simulate sea level contribution on the used corrected flux does exist, however, we found it to be not very strong for most of glaciers, by performing experiments with different relaxation times. Such weak dependence can be explained by the fact that for the outlet glaciers (unlike the rest of GrIS), changes in SMB plays only a secondary role in glaciers retreat compare to changes in submarine melt and calving.

p.8 I.19: I am confused about this comparison at different depths? Why not use temperature profiles over the entire depth? Also how did you choose these depths? Do they correspond to the depth of warm or cold water? Or the changes in the thermocline? What is the rational for this choice?

As we explained above (and we make it now more clear in the revised manuscript) we used the reanalysis data as the fallback option for the fjords for which there are no CTDs available. The reanalysis data are available at the regular grid and at the vertical level 5, 30, 50, 100,200, 400, 700, etc ...3000 m. Since most of submarine melting occurs below 100 m and typical depth of Greenland fjords is up to 700m, we restricted our comparison of (continuous) CTD profiles with the reanalysis data at these three available depths – 200, 400 and 700 m. The main conclusion we made is that when constructing vertical temperature profile using reanalysis data it is better (better agreement with CTD) when we fixed temperature below 400 m rather than interpolate between 400 and 700 m. We added now a corresponding figure in the SI (Fig. S3) to make this part of discussion more clear.

**p.8 l.23: This is also the case for Jakobshavn (figure 4).**

Yes we agree that also Jakobshavn shows the same feature but this is shown in Figure 4. The actual location of the reanalysis point is only shown in Fig. 6 exemplary for Store Glacier. We therefore write now:

"Figure 4 and 5 compare the temperature at these depths from reanalysis data with available CTD profiles measured over past several decades for Jakobshavn-Isbrae and Store Glacier. Since Greenland is surrounded by the continental shelf with typical depths of 200–400 meters, most of the 700-meter depth grid-cells in the reanalysis data are located outside the shelves, far away from the glacier mouth as shown in Fig. 6 on the example of Store Glacier."

p.8 I.26-32: I have the impression (and this is not very clear in the manuscript) that

you don't use the sill depth in the fjords to determine the water properties in front of the glacier. The sills block the warm water at depth, which can significantly impact the water properties. This should be included for the plume model. Why not use that instead of an arbitrary depth of 400 m? Accurately including the fjord properties in important to separate the response due to the trend in climate changes from the impact of local conditions of the glaciers and the fjords.

Again, we thank the reviewer for this suggestion. We demonstrate now (Fig. S3) that for temperature profiles derived from reanalysis data, changes according to a (shallow) sill depth do not improve the temperature profile in comparison to terminus-close CTD measurements. We therefore write: "Similar to the continental shelf, 'blocking' shallow sills in a fjords modify the water masses near the grounding line of a glacier. However, considering of the sill depth (Fig. S2, supporting information) when reconstructing the T-S profiles from the reanalysis data only leads to an even stronger temperature bias (dashed line Fig. S3, supporting information). Therefore, we always use the reanalysis data from 400m depth to construct T-S profiles irrespectively of the sill's depth".

 $p.8 \ l.29$ : "larger"  $\rightarrow$  "deeper" Agreed, changed accordingly.

p.9 I.10: Again here, why used the temperature at 400 m depth and not the temperature at the grounding line depth? I think the value used should be designed to best represent the conditions in each and every fjord instead of using a generic value systematically applied to all the fjords.

Here we only derive a trend at the 400m depth point, from CMIP 5 models, since the continental shelf only allows water masses to pass from 0 to 400 m depth and the deep bottom water controls submarine melting. We add this trend to the total temperature profile (measured and reanalysis) which includes the temperature at the grounding line depth. We clarify now::"For future simulations, we prescribed simple scenarios for the ocean temperature anomalies based on temperature trends simulated by several CMIP5 models (GFDL-ESM2G, MPI-ESM-LR, and HadGEM2-CC). The trend is added to the T-S profiles (both CTD and reanalysis) for the future simulations. To determine this temperature trend we use the closest to the fjord model grid-cell with the depth larger than 400m for each CMIP5 model. The temperature trends were approximated by linear regression as illustrated in Fig. 8. The Figure shows as well, the big discrepancy between the model temperatures and CTD measurement at 700m depth which was the motivation to use 400 m depth only." this in the revised version.

**p.9 I.16-21: It would be great to see the values of the different results, and especially how the different runs agree with the observations. More details on the choice of runs selected should also be added.**

We demonstrate or results of the spin-up experiments of the present-day tuning in Figure 9. For completeness we now list the values of the 4 dynamical parameters, beta and fwd range and number of simulations in the SI Table S2 and S3.

**p.9 I.22-26: This paragraph is not clear.**

Agreed. We would rewrite the paragraph to:

"Once the four dynamic parameters and the relaxation time scale are set in our precalibration, we performed a set of spin-up experiment with the coupled glacier-plume model for each glacier. In the spin-up experiments the submarine melt rate is now simulated interactively by the plume model which requires subglacial discharge and temperature and salinity profiles as inputdata. We used monthly subglacial discharge for the year 2000. Vertical temperature and salinity profiles in these experiments were taken from the reanalysis data, averaged over the time interval 1990–2010 or from recent CTD data, and were held constant in time (Fig. S3, supporting information). Nonetheless, in the spin-up experiments the submarine melt rate is not constant since changes in the grounding line depth and shape of a floating tongue (if exist) affect the submarine melt. We chose the year 2000 as the quasi-equilibrium initial state for "future" climate change simulations since the mass loss of GrIS during the last decade of 20the century was rather small (ca. 0.1

mm/yr in sea level equivalent) compare to that has been observed in the 21st century ((Vaughan et al., 2013)."

**p.9 I.27: scaling of what? How is that done?**

That was explained in 2.2 but we rewrite in brackets :" (factor in a range from 0.3 to 3 that multiplies the simulated melt rate profile)"

*p.10 l.6: What is 3.3?* We forgot the word "section". Now inserted.

**p.10 I.14-18: I think this could be easily simplifies in saying that you use the volume above flotation.**

Agreed. We deleted the lengthy explanation with equations and added the sentence. "The contributing ice volume V\_SLR is determined by the lost ice volume above flotation from each glacier"

p.10 I.21: Mention that is the present-day simulated state.

Done.

*p.10 l.21: It is not clear what you mean by calving ratio.* We explain after the first occurrence: "(grounding line mass flux lost by submarine melting divided by mass loss of calving)"

p.10 I.23: The grounding line position is not clear on the figure, the ice front position is. Also most of these glaciers do not have any floating tongue, so it would be better to use the term ice front in this case.

We added a close-up view of the grounding line position in the SI (Fig. S5). Glaciers named as tidewater glaciers as e.g. Helheim still evolve small tongues mostly before the melt season. We added the sentence:

"Note that we allow for small floating termini, since many tidewater glaciers still evolve them on a seasonal scale and glacier fronts are also mostly undercut and thus missing a pure vertical cliff without any floating terminus (Bevan et al., 2012; Straneo et al., 2016; Rignot et al., 2015).

p.10 I.21-30: Do you actually want the glaciers to be stable or to be representative of the present-day conditions? Because many of these glaciers are losing mass and retreating today, so how much should a spin-up with present-day conditions lead to stable conditions?

See our response to the 3rd major comment

*p.11 I.2: I thought that most of these glaciers did not have floating termini anymore!* We now address this issue by explaining why we allow for glacier tongues to evolve in our glacier model. (see answer o two comments above)

p.11 I.6 "by Enderlin .."

Adapted.

p.11 I.17: The numbers you provide do not include the present day changes? This is quite surprising and ends up presenting very low sea level change numbers that are not in good agreement with today's observations. It also questions the initialization procedure of the model, how much can we separate the present state and future changes given that the initial conditions have a lasting effect on the results.

We are not certain which numbers are meant here by the reviewer and how these numbers can be in agreement (or disagreement) with observations. In our paper we give only the contribution to SLR for the period 2000 – 2100. Our median estimate for the all Greenland glaciers based on upscaling is 50 mm, which is within the previous estimates for the same value. We argue that this number is complimentary to the SLR contribution simulated by a global GrIS model which does not

account for ice sheet-ocean interaction (e.g. Calov et al., 2018). The sum of these two separate contributions (see our Discussion) gives ca. 140 mm, which is well within the range of existing estimates (e.g. IPCC, 2013; Fürst et al., 2015; ). At the same time the recent estimates for the total GrIS contribution to SLR around the year 2000 is about 0.2-0.4 mm/a of which only half is attributed to the enhanced solid discharge (Enderlin et al., 2014). These numbers are not negligible but still significantly smaller than the average SLR which we simulated for the entire 21th century. Therefore the assumption we made that glaciers at 2000 were in quasi-equilibrium cannot have significant effect on our estimates for the SLR.

p.11 *l.26: "excluding"*  $\rightarrow$  *"separating"* Adapted accordingly.

p.11 I.27: This is not very clear, try to better separate the numbers for SMB only, elevation feedback, climate change trend, ocean, ... as is done in figure 11.

We write now:

"When forced by comprehensive climate change scenarios (changes in SMB with the surface elevation feedback, ocean temperature T and subglacial discharge Q) the median estimate for SLR contribution from all 12 glaciers is about 17 mm at the year 2100. To quantify the role of the individual forcing factors, we perform additional set of simulation with the model versions corresponding to the median SLR response by applying different forcing factors separately. We found that from the 17 mm over 70 % of SLR is caused by increased submarine melting due to the ocean warming T and increased subglacial discharge Q (Fig. 11 b). We found that both factors, T and Q, contributed an approximately equally to SLR. The reaming 30 % are attributed to the glacier's response to changes in SMB (Fig. 11 b, orange curve)."

*p.11 I.30: "substantially"*  $\rightarrow$  *"substantial"* Adapted accordingly.

p.12 I.13: The potential SLR and grounding line retreat are actually not listed in the tables.

We disagree, since they are listed in Table 3 and 4.

p.12 l.15: "uncertainties"  $\rightarrow$  "spread" Adapted accordingly.

p.12 I.18: There is only one model used to generate SMB, so where is the spread coming from? It is not clear if is caused only by the different initial conditions used or if here is something else. Also, why is there only one model used to generate SMB and several for the ocean?

The spread is actually coming from the different initial condition caused by the freshwater depth and beta. We inserted now this explanation "Since there is only one SMB forcing the spread originates from the different initial states cause by the different fwd and beta combination."

p.13 I.1: "1D line plume model" → "1D plume model" (same in other places in the manuscript). Also "Jenkins (2011)" → "(Jenkins, 2011)"
 Adapted accordingly throughout the manuscript.

p.13 l.12: How does that compare to other 2D or 3D models of Jakobshavn [e.g., Muresan et al., 2016; Bondzio et al., 2017]?

We compare to other 3d simulations in our discussion:

"Simulated SLR contributions for the year 2100 compare well to values from Nick et al. (2013) for Jakobshavn Isbrae. The conservative estimations of Jakobshavn Isbrae contribution to SLR obtained with the 3D model of Bondzio et al. (2017) also lie within our uncertainty range. For the Kangerlussuaq Glacier our estimates for SLR contribution exceed estimation of Nick et al. by 2 mm, while for the Helheim Glacier our SLR estimations are below the estimation of Nick et al. (2013). In our simulations all glaciers experience a grounding line retreat which is found as well by Nick et al. (2013) but was not simulated by Peano et al. (2017). This discrepancy might be related

to the coarse spatial resolution (5 km) of Peano et al. (2017) model (especially for the deep and narrow trough in Jakbobshavn ) or processes upstream of the glacier might have counterbalanced the glacier retreat, which we could not simulate with a 1D flowline model."

p.13 I.19-20: remove

Adapted accordingly.

p.13 I.30-35: Use present tense instead of past tense.

Adapted accordingly.

p.14 I.5: What are the numbers for the entire Greenland if you only take the same glaciers as Nick et al. [2013]? How are these numbers impacted by the choice of glacier? So, if you only include a subset of the 10 glaciers used in this study, how does the sea level contribution of Greenland vary? It would be interesting to compute some kind of uncertainty associated with this method.

We thank the reviewer for this interesting suggestion and as mention above, we derived an uncertainty estimation by choosing 4 different glaciers in the SI.

p.14 I.7: "our our"

Deleted.

Fig.2: Is there a white dot in the fjord? It's not very clear. I don't understand the choice or the use of CTD profiles. Why not use all (or a combination of the different) profiles?

Since the plume equation require the temperature of the ambient water that entrains into the plume, we chose the closest, (and deep as the grounding line) available CTD measurement (closest to the glacier terminus) not CTDs far away. Fig. 2. was improved

"depth of 400 m"  $\rightarrow$  "depth of at least 400 m". "od"  $\rightarrow$  "of" Adapted accordingly.

Fig.4: same as Fig.3

Yes, therefore we never use temperature profile from reanalysis data at 700m depth, since the are located outside the continental shelf.

*Fig.6: It would be better to label all the dots (they are only 12). Again here, why use the depth-averaged temperature and not the temperature that most impact the plume model?*

This was done to get a overview of how far the profiles of CTD and reanalysis data are actually of. We miss-wrote, since we actually show the temperature at the grounding line depth, which is the one that most impacts the plume model. We changed the axis-titles in Figure 6 and labeled all the dots, as suggested by the reviewer. Nevertheless, for transparency, we now show all the CTD an reanalysis data temperature profile for each glacier in the SI (Fig. S3).

*Fig.8: Why present the results from only one ocean model and not from all of them?*

Results of only one model and only for one location is shown in Fig. 8 just for illustration. The total range of temperature trends derived from different models and for different locations are given in Table S1, SI.

What is the implication of large discrepancy at 700 m depth between the model and the CTD measurement?

The likely reason for this discrepancy (actually 1°C error is not large for the GCMs) is that the nearest model grid point with the depth 700 m is located far from the CTD location. This is why, similarly to constructions of the vertical temperature profile from the reanalysis data, where the lowest depth we used was 400 m, to construct ocean warming scenarios we also disregarded

levels below 400 m and instead prescribed temperature trend simulated by CMIP5 models at the depth 400 m. Note, that for climate change scenarios we did not use absolute values but only the anomalies simulated by CMIP5 models. These temperature trends for different locations and models are listed in table S1 (SI).

Fig.9: Is the observed bedrock directly taken from the BedMachine dataset along the centerline or is it representative of the entire glacier (of its entire width)? How many stable states are used for each glacier? I could not find this information in the manuscript.

And as mentioned above, do you really want the initial configuration to be stable or to represent the current state of the glacier? I am not sure "transparent lines" is the appropriate term.

We clarified the source from our bedrock. In part 3.2:

"We use the BedmACHInev2 data for bedrock topography (Morlighem et al., 2014). Fjord bathymetry was extended manually by considering available data (Mortensen et al., 2013; Schaffer et al., 2016; Dowdeswell et al., 2010; Syvitski et al., 1996; Rignot et al., 2016)." The number of stable states is listed in the table S1 in the SI.

Fig.10: "median-range3": repeat the superscript meaning here. Fig.11: "vom"  $\rightarrow$  "from"

Done.

Fig.12: Try to use the same order as for Fig.11 for the lines. Fig.13: Would be better to repeat the entire caption.

Done.

*Fig.14: What is "ocean temperature trend 1"?F* Listed in *Table* 1. Corrected in the revised version.

Tab.2: I thought that most glaciers in Greenland did not had floating termini any more, so why are there relatively large ratios of melting?

They do, within the season the can evolve short termini that are after the melt season mostly calved. We write:

"Note that we allow for small floating termini, since many tidewater glaciers still evolve them on a seasonal scale and glacier fronts are also mostly undercut and thus missing a pure vertical cliff without any floating terminus (Bevan et al., 2012; Straneo et al., 2016; Rignot et al., 2015)."

Tab.4: It is not clear what the sum of grounding line retreat represent. It is a rather unusual metric.

We agree with the reviewer and changed the entry in the table to the average grounding line retreat.

**Response to reviewer 2**

This paper investigates, by means of numerical modelling, the evolution of 12 outlet Greenland glaciers in the next century (2100). The employed numerical models are a 1D flowline glacier model and 1D (ocean) plume model, they are coupled together. Two aspects represent important limitations of this work: the use of a 1D glacier model for confined glaciers and the methodology followed in forcing and using the 1D coupled plume model. Some of the assumptions of this work are not properly addressed or discussed, as well as some of the consequences on the obtained results. This paper is clearly written, with the exception of some paragraphs that may lead to some confusion about the experimental setup (e.g. It is not clear if you actually run SICOPOLIS or not. Including a "methods section" may ease the reading).

We run SICOPOLIS and details to this can be found in our earlier study Calov et al 2018. All coauthors in this current paper contributed also to Calov et al. 2018. For this manuscript, only the output data on subglacial discharge from Calov et al 2018. were used to force the coupled glacier plume model.

**Main comments**

On the plume model:

I think that using the coupled 1D plume model is a great improvement. However some experimental choices limit the validity of this improvement. At page 5 – line 2 is written that "since the plume model in some cases underestimate...

we also scale the simulated melt rate profile by a factor Beta...". I have some comments on this: the relation between the plume forcings (temperature,salinity, shelf/tongue slope, subglacial discharge, . . .) and melt rate is given by robust physical equations (Jenkins, 2011; Beckmann et al. 2018). I believe that tuning the obtained melt rates with a multiplying factor waste all the efforts made in using (and coupling) the plume model. What is the need of this sophisticated model if then the computed melt rates are scaled to observed melt rates? Then why not using a simple depth dependent parameterization (e.g. Martin et al., 2011)?

Indeed, Jenkin's model of turbulent plume is based on the first principles and therefore it is expected it provides robust qualitative relationship between submarine melt ,ocean temperature and the slope of glacier front. Whether this model is also quantitatively correct for each Greenland fjord is another issue. The real world is very different from the assumptions behind the linear plume model since during summer season significant amount of melt water is delivered into the fjord through a number subglacial channels. At present, there is no way to simulate realistically the large ensemble of different plumes, as well as many other processes (tidal circulation in the fjord, undercutting, etc) which may also contribute to submarine melt. To describe this complex reality we proposed to use the Jenkin's linear plume model but with additional correction by parameter beta. Obviously there is no prove that this parameter will stay constant for the next 100 years but still we believe that our approach represent an important improvement compare to a much simple parameterization (we assume that the reviewer means here the parameterization by Beckmann & Goosse, 2003) since we explicitly account for the dependence of submarine melt on subglacial discharge which is very important factor for the global warming simulations.

You tuned the computed plume melt rates on present day observed melt rates. How can you assume that this "present day" scaling will still be valid in 50/100 years? This choice is crucial in terms of providing a robust basal forcing for the glaciers evolution. I think that this assumption should be discussed.

As we explained above, there is no reason to expect that a very simple Jenkin's linear plume model can accurately described complex reality of Greenland fjords even at present and thus there is reason to expect that correction parameter beta will remain constant over 50 or 100 years. The reviewer is absolutely right (see Fig. 15): the choice of melt and calving parameters is the source of the largest uncertainties in glaciers contribution to future SLR and one of the aim of our paper is to report this problem. How to fix this problem is beyond the scope of this paper.

Given the inherent large uncertainties in forcing conditions (both in CTD and in reanalysis, page 8 line 3) what about forcing the plume model with a range of plausible temperature and salinity (from CTD and/or reanalysis) and with a range of subglacial discharges instead of tuning the computed melt rate?

Obviously, uncertainties in temperature profiles and subglacial discharge also contribute to the SLR uncertainties but very unlikely they contribute to the discrepancy between melt rate simulated by Jenkin's model an real one. Indeed, typical uncertainties in water temperature of 1°C will result

in 20% uncertainties in melt rate. The uncertainties of 50% in subglacial discharge results only in 15% uncertainties in melt rate (due to cubic root dependence). At the same time, as we show in Beckmann et al (2018), melt rate simulated by linear plume model can deviate from observed one by factor 2-3.

It is not clear why you decide to use reanalysis data at 200, 400 and 700 meters of depth instead of using continous vertical profiles. Moreover, for future simulations you say: "...closest 400m-depth-point neighbor...". Is this motivated by line 29 to 31 at page C28? I understand this choice but I believe that you shold explain this better, clearly motivating also at page 9.

The first reviewer has a similar question which is addressed in our response. Note that we always use continuous profiles. Obviously, this part of our paper was not clear enough and imopoved it in the revised version.

**On the glacier model:**

I get why you decide to use a 1D flowline model: however I think that the limitations related to this approach (neglect of processes at the lateral boundaries and of buttressing, which play a crucial role in the evolution of ice masses) are not properly tackled and are mostly addressed by saying that 1D models are the only one available for this kind of study. This is probably right if you want to model 12 (or more) glaciers at the time, but for single glacier the last few years have seen important improvements in modelling alternatives that have produced results for some glaciers that are also modelled in this work (Chaulet et al., 2012; Seddik et al., 2012; Muresan et al., 2016; Peano et al., 2017; Goelzer et al., 2017). I think that the discussion about 1D model limitations should be expanded.

We agree with the reviewer and discussed more in depth the limitation of a 1D glacier model in the discussion part:

"Additionally, the 1D flowline model treat lateral processes in a simplified manner, so that more complex bedrock geometries (e.g. branching of glaciers, individual sills, unsymmetrical valley forms) are poorly represented in these estimations."

Also did introduce more work from other authors on 3d models on glaciers in the introduction part: "Peano et al. (2017) investigated the 5 biggest ice streams and outlet glaciers in Greenland with a 3D ice-sheet model on a resolution of 5 km. Seddik et al. (2012) and Gillet-Chaulet et al. (2012) included improved model physics by using a full-Stokes approach and refined resolution over fast flow regions with adaptive mesh techniques. Their setup however, did not yet allow to simulate glacier retreat. Most of the ice-sheet simulations also do not describe the interaction between glaciers and the ocean explicitly, but in some cases, for instance in Fürst et al. (2015), ocean melting is parameterized indirectly by increasing the basal sliding factor as ocean temperature increases. For the RCP scenario 8.5, they calculated a SLR between 155 and 166 mm at the year 2100 for the entire ice sheet atmospheric and oceanic forcing. For regional settings on 3D models with a simple ocean melting parameterization were applied to study the historical (last 20 - 30 vears) retreat of Jakobshaven Isbrae (Muresan and Khan, 2016; Bondzio et al., 2017). A more advanced treatment of submarine melt rate was done by Vallot et al. (2018). They coupled a plume model based on the Navier-Stokes equations with a full-Stokes ice sheet model. With this off-line coupling, glacier dynamics for one melt season were simulated for Kronebreen Glacier in Svalbard.'

**Specific comments**

Page 1 – line 15: "factor analysis". With factor analysis it is usally meant a statistical method like the Empirical Orthogonal Functions (EOFs), in your work you just exclude (one at the time) the different forcings, I would not strictly define this procedure as a factor analysis.

We changed "factor-analysis" to " sensitivity analysis of the forcing-factors".

Page 2 – line 5: instead of "global" I would use "atmospheric" Adapted accordingly.

Page 2 – line 4 to 8: I found this paragraph ok, but I would rearrange it a little bit putting the described processes in the same order you are introducing them. We now describe the processes in the order we introduce them.

Page 2 – line 6: "marine terminating" instead of "marine- terminating" Adapted accordingly.

Page 2 – line 16: "In order to..." this should be a new paragraph Adapted accordingly.

Page 2 – line 32: "that" is repeated two times Deleted the second 'that'.

Page 2 – line 35: "Since we are.." this should be a new paragraph Insert a new paragraph.

Page 3 – line 1: I would say that the main (and only) improvement consists in using the coupled plume model. I consider the fact of studying more glaciers just as an "extension" of Nick et al. 2013 work.

Moreover, from the scaling perspective, are we sure that the considered glaciers are really representative of all the Greenland

glaciers? especially given their variety in terms of glaciers and of confining fjords geometries/conditions.

Agreed, we changed the part to:, "...we followed an approach similar to Nick et al. (2013) but for different glacier-types and with one notable improvement.: For calculations of the vertically distributed submarine melt, we use a turbulent plume parameterization following Jenkins (2011)."

We considered 12 glaciers as in improvement compared to Nick et al. and selected them since they represent different ice flow regimes and different environmental conditions. We mention now that a sufficient sample size is crucial for the scaling method: "These resulting regression line is however not statistically significant. This underlines the importance of choosing a sufficiently large sample size."

Page 3 – line 4: ok, but submarine melt rate depends also on the geometrical features of the tongue (shape, slope,...)

Agreed, we changed the sentence to. "According to this parameterization, the submarine melt rate depends not only on ambient water temperature in fjords but also on seasonally varying subglacial discharge, shape and angle of the glacier tongue."

Page 3 – line 9 to 11: Maybe you can think about shortly describing how the scaling works.

Agreed we added: "In particular we derived a proportional factor between present-day grounding line discharge and future SLR using results of simulations for all twelve glaciers."

Page 5 – line 1 to 5: I would expand the plume paragraph since it is the real innovative part of this study. Maybe a short introduction of the basic physics and equations. Otherwise is not clear what do you mean with the E entrainment parameter unless looking at Beckmann et al. (2018) (or already knowing what you are talking about).

We agree with the reviewer and extended now the whole paragraph and add the equation for the plume model.

Page 5 – line 17: "to the vertical mass balance term B", add the equation number We added the equation number.

Page 5 – line 18 to 20: I imagine that when the plume detaches the melt rate is set to zero but this is not written explicitly. Is this the case?

We thank the reviewer for spotting the lack of information.

The plume never detaches from the glacier in the model, it only ceases by slowing down the velocity to zero. When this happens, the melt rate is set to a minimum melt rate to ensure background melting. We describe this in the revised version as the following: ". If the plume already ceases before reaching the calving front  $x_cf$ , we numerically introduce a minimal background melting determined by the last melt rate value before the plume ceased".

Page 5 – line 21: this part confused me. "...off-line using the ice sheet model" which one? This is the first time that you mention the use of an ice sheet model. Later it appears that it is SICOPOLIS.(see comment to page 6 – line 15 to 25)

Yes, we used SICOPOLIS output data which is described detailed in Carlov et al. 2018. We now write:

"We prescribe the subglacial discharge for each glacier simulated off-line with a monthly time step from the output of the ice sheet model SICOPOLIS."

Page 6 – line 1: "did we" "we did". Could you explain better in what this upscaling consists and how it works?

We added now we describe the upscaling method now firstly rough in the introduction part: "In particular we derived a proportional factor between present-day grounding line discharge and future SLR using results of simulations for all twelve glaciers."

Page 6 – line 2: it would add more clarity defining what is meant with "melting to calving ratio"

Agreed, we inserted: "(grounding line mass flux lost by submarine melting divided by mass loss of calving)"

Page 6 – line 12: just a detail: I would number the figures in the order of appeareance in the manuscript.

Agreed, adapted accordingly.

Page 6 – line 15 to 25: From here it looks you actually run the ice sheet model, is this correct? (look comment to page 10 – line 6). I suggest to introduce explicitly the fact that you have run SICOPOLIS.

Yes, we run Sicopolis earlier and details can be found in Calov at el 2018. However, after the SICOPOLIS simulation we used the subglacial output data to force the coupled glacier plume model off-line. The sentence is changed to:

"The former two sources are computed directly from the ice sheet model SICOPOLIS by (Calov et al. 2018)."

Page 6 – line 23,24: "...is assigned to the closest glacier within a maximum of 50 km". This is an important approximation since is related to the plume forcing, however is not properly discussed, expecially in terms of uncertainty in the obtained results.

We now discuss: "This maximum distance is necessary in areas where only few named glacier positions are available (mostly in the South of Greenland) and the distance between glaciers is large. For most of the coastline, especially in the area of our selected glaciers, this distance has no

effect on the results. We did not separately study the uncertainty in subglacial discharge related to this approach, but rather accounted for this uncertainty implicitly through the uncertainty of the scaling coefficient  $\beta$  for the submarine melt rate (see chapter 4.1)."

Page 6 – line 27,28: "neglect the effect of grounding line retreat". As above, this represents another important assumption but it is not properly discussed.

True. We added :

"For neighboring glaciers with a competing catchment area, a strong ice sheet retreat may strongly affect the distribution of the subglacial discharge between those glaciers (Lindbäck et al., 2015). This effect is not included in this study."

Page 8 – line 12: "...presence of sills in the fjord...in the vicinity of the glacier front." I would explain why is that after this line, instead than explaining it later for the continental shelf (at page 8 – line 24 to 30).

We rearranged the whole sub-chapter on temperature and salinity profiles.

Page 9 – line 16: could you provide a table with the prescribed submarine melt rate and the range of values for the dynamic parameters? (maybe in the supplementary)

Agreed, we provided it int the table S1 in the SI.

Page 9 – line 25: with "...only factors.." do you mean that since temperature and salinity are "held constant" (thus not changing) their contribution in impacting melt rates is constant in comparison to the impacts due to a varying grounding line depth and tongue shape/slope? I suggest to reformulate this paragraph

Yes, we just wanted to point out that although the temperature-salinity profile is held constant the melt rate isn't necessarily constant due to the glacier's changing geometry. We now write "Nonetheless, in the spin-up experiments, the submarine melt rate isn't necessarily constant since changes in the grounding line depth and shape of a floating tongue (if exists) affect the plume equations. "

Page 9 – line 29 "...is close to equilibrium state.." what do you mean with equilibrium? Later you speak about stable state. Do you mean steady? I would argue that currently Greenland glaciers are definitely not in a steady condition.

The same issue was addressed by reviewer 1 (major comments) and we defined our definition on the present-state (2010) more clearly.

Page 10 – line 5: "...each glacier 3.4..." something is missing between glacier and 3.4

Delete 3.4. (typo).

Page 10 – line 6: "...glacier individually 3.3..." something is missing between individually and 3.3

We thank the reviewer for spotting the mistake we insert the word "section".

Page 10 – line 6: Here it is not clear if you took the data from Calov et al. 2018 or if you actually run the model

This paper and Calov et al. (2018) are closely related. They originate from the same project and are written essentially by the same group of authors. Calov et al. (2018) describes the model of Greenland glacier system, experimental setup and results of several climate change experiments.

In Calov et al. (section 5) we also described how we computed time-dependent subglacial discharge for individual Greenland glaciers using SICOPOLIS and MAR output, and the basal hydrology model HYDRO. In the current work, we used this time-dependent discharge as the forcing for modeling of 12 selected glaciers. We clarified this issue in the revised manuscript.

**Page 10 – line 22,24: this part about the interplay between melting, calving and bedrock is interesting. I would add few more details.**

**We now broaden the discussion and write:**

"We found that for some glaciers, the grounding line demonstrates a high sensitivity to the melting/calving ratio, while others are primarily controlled by their bedrock topography and have relatively small variations in their grounding line position over the whole melting/calving ratio range. The Gade and Upernavik North glaciers are examples of the latter case (Fig. S6, supporting information). In general, we observed higher velocities at the glacier terminus when higher calving rates were applied. Thus, if a glacier is not strongly buttressed by a sill or lateral resistance, different values of velocity at the glacier terminus due to different d w strongly affect the equilibrium grounding line position. Such behavior points on the crucial role of the bedrock topography for glacier dynamics."

Page 11 – line 6: a space is missing before "Enderlin"

Insert space.

Page 11 – line 15: "model versions" do you mean the the spin-up ensemble? Yes, we now write.:"After obtaining the present-day state, we then ran the model ensemble with all valid beta/fwd combination ..."

Page 11 – line 16: why not changing also the subglacial discharge? It is such an important forcing for the plume and comes from several approximations (fixed grounding line and closest neighboring approach).

We fully agree with the reviewer that in this study we explore only a fraction of uncertainty sources. In particular subglacial discharge as well as SMB also depends on the choice of the regional climate model and the global climate models which has been used to provide boundary conditions for the regional model. However, we believe that Fig. 15 already provides a very important inside into the major source of uncertainties in simulated glaciers contribution to SLR. Namely, it shows that the uncertainties in the choice of model parameters is likely to be the largest source of the SLR uncertainty. Thus to considerably narrow down these uncertainties, the glacier model parameters have to be better constrained.

Page 11 – line 17: at page 10 (line 8 to 10) is said that also the unforced model drift is calculated. Then this drift is removed by subtracting it from calculated values. This implies that a linear behaviour for glaciers is assumed. I think that this should be properly discussed.

Of course it is known that glaciers response to climate change is nonlinear and we do not assume such linearity. Our modeling approach is based on the assumption that glaciers were in equilibrium at the year 2000. However, to ensure that all glaciers are in the perfect equilibrium with the 2000 year forcing would be required to perform infinitive number of infinitively long spin-up experiments which is not possible even with fast model. This is why we apply as additional constrain, namely, we excluded all model realizations with positive (mass gain) trend and require that the simulated negative trend is significantly smaller than simulated SLR response to climate change scenario. Still, we have to tolerate non-negligible drift in the control runs - otherwise we will be left with to few accepted model realizations. This is why we decided to exclude such drift from the forced run which we believe is still better to do it. We included now:

"All results shown here have a small model drift subtracted from the calculated values, to ensure that the simulated SLR is a response to the climate change."

linear systems. This issue is just slightly addressed at page 12 – line 4.

We did not assume that glaciers are linear systems. As we explained in the response to the first reviewer, the drift is rather small since we only accepted such model versions in which drift is smaller than simulated SLR in the forced experiments. Of course, zero drift in unforced experiment would be preferable, but this cannot be achieved with the finite computational resources. Therefore we are left with two to options: (i) to leave forced experiments as they are or (i) to exclude unforced drift from the forced experiments. Both options are imperfect but we prefer the second one.

21,22: you attribute the source of uncertainty to Beta, this comes from the fact that Beta is responsible for the imposed melt rate (through the tuning procedure). However Beta is just a model parameter, I think that avoiding the use of Beta (as suggested in he main comments) could also improve this part of the work, it will allow you to relate uncertainties to physical quantities.

As we explained above, there is no physical reasons why the linear plume model should produce correct results with beta=1. See also Beckmann et al. (2018).

Page 13 – line 18 to 20: something is wrong here, an entire sentence is repeated. The repetition was deleted.

Page 13 – line 33: same as above. Your results are not affected by CTD/reanalysys temperature and salinity because the Beta tuning incorporates all the uncertainties.

We agree with the reviewer that colder temperatures in the reanalysis data set in some (but not all cases) can be balanced by a higher beta. We provided the new table showing which beta values were used for CTD and reanalysis data set in the supporting information (Tab. S3)

Page 13 – line 35: "...observational constraints on submarine melt..." as explained in the main comments I think that we should rely on melting formulation as less as possible dependent from a tuning on observations, especially for future projections.
We agree that this would be a nice idea but not at the present level of ice sheet-ocean interaction. As we showed in Beckmann et al 2018, Jenkin's linear plume model does not produce observed submarine melt and therefore should be corrected. Even with this correction believe that our approach is more physically based and therefore more trustworthy than those used in previous studies.

Page 14 – line 4: "and" repeated two times

Deleted.

Page 14 – line 7: "our" repeated two times

Deleted.

Figure 3(a): I think that using white dots is a bit unfortunate, also the red star is not very visible.

Agreed, we improved Figure 3 for more visible CTD location.

*Figure 11: "from" instead of "vom"* Adapted accordingly.

[revised manuscript text omitted]

---

## Author Response (AR2)

We thank the reviewers for the constructive reviews and suggestions. The comments by the reviewers are in indented blocks and italic fonts. Our response follows each comment and changes in the manuscript are – if shortly- written here in quotation marks.

**Reviewer 1**

> This paper forces a 1d coupled glacier-plume model with future climate from the CMIP5 climate models to project the future behavior of 12 of Greenland's tidewater glaciers. The topic of sea level rise and ice-ocean interactions in Greenland is of great contemporary interest, and the use of a plume model is certainly an improvement (subject to concerns about tuning and true representation of coupling between melting and calving) on previous similar studies such as Nick et al. 2013.
>
> Overall I think it is easy to criticize papers doing future projections because there are a great many factors to consider and decisions to be made, and many compromises are necessary. I have tried to write my review with this in mind, and I commend the authors for bringing many datasets and models together and for addressing the many challenges inherent in projecting future sea level from Greenland. Contrary to the previous reviewers I did not find the paper to be clearly written; perhaps it is simply the nature of describing a complex model with many inputs, but I found several passages to be rather hard to follow (notably the ocean forcing description), which will also come across in my review.

We understand that the complexity of Model inputs can be confusing and we therefore now introduced Figure 2, which shows an overview and the interaction of all the different model components. A short summary on the the data input introduces the subsection in section 3.

> I do have some major concerns, some of which might be misunderstandings arising from a lack of clarity, but these should be addressed or the writing improved. For example, I am concerned there may be some important errors with the surface mass balance. On the use of a flowline model, I agree with the previous reviewers that this is a serious limitation and the field is beginning to move beyond flowline models. On balance however, I do not think this should preclude publication provided this is clearly acknowledged, the paper focuses more on the main qualitative points of interest and the major comments below can be addressed.

As we will explain in the major comment section, we do not see any "important errors" in our calculations of the surface mass balance. We are aware that some workers do "move beyond flowline models" but this doesn't make flowline models irrelevant. The modeling of such a complex and nonlinear system as the Greenland glacier system, it is natural to use different modeling approaches (spectrum of models) which are useful to address different aspects of the problem.

> For me, the two strongest points this paper makes are (i) that when upscaling results from a handful of glaciers to the whole ice sheet we should be careful about which and how many glaciers we sample, and (ii) quantifying the importance of dynamics-SMB coupling in future projections, though I found the discussion and figures on this point to be highly confusing. In general I felt the paper could benefit from focusing on/emphasizing/clarifying these points.
> I have split my comments in major and minor comments. I have not noted every spelling and grammatical error as these are rather numerous and should be easy to spot.

**Major comments**

> 1. Representation of melting and the coupling of melting to calving
> A plume model is used to represent submarine melting of the calving front. When a floating tongue is present, this is naturally applied as a thinning of the tongue. When the glacier has no floating portion the submarine melting is applied as a thinning of the last grounded cell, which is less natural (it is essentially treating the submarine melting as a surface mass balance). Calving is represented via the commonly used crevasse depth criterion. My major comment is: how do we know that this treatment of combined melting/calving actually captures the effect of submarine melting on calving? How does it relate to emerging understanding of the coupling between melting and calving through undercutting (e.g. Benn et al. 2017, J. Glac.)?
> For calving fronts without a floating portion, how well does the application of submarine melting as a 'vertical' surface mass balance term represent the 'truer' process of melting incising horizontally into the calving front?

*At the moment it feels as though the authors have taken their approach because it is the one which works for their model rather than on the basis of representation of processes. Reaching the conclusions of this paper without discussing at all whether the model actually captures ice-ocean processes seems remiss. I see that this is very briefly mentioned in the discussion and conclusions sections but this is too little and too late at the moment.*

a) **Treatment of submarine melt in the model**. It only seems "natural" to treat lateral submarine melt as the flux in the equation for ice thickness (Eq. 1). In fact, as any gridded ice sheet model (including 3-D models), our flowline model has only one characteristics for ice distribution – ice thickness and therefore submarine melt can only appear in the equation for ice thickness. However, in the case of an absence of floating tongue, changes in the thickness of the last grid cell is immediately recalculate to the change of the position of grounding line. As a result, our grounding line respond to changes in submarine melt in continuous and realistic manner but, of course, neglects undercutting (see the next section).

b) **Undercutting**. We are aware about recent studies on the effects of submarine melting on calving processes, but these studies are still limited by several types of calving styles and the high resolution models used in ( Benn et al. 2017, J. Glac.) cannot be applied to realistic conditions. Undercutting can amplify calving but can also stabilize glaciers. Benn et al. (2017, J. Glac) presume that there is a rather "small effect of under-cutting on glacier stability " in Greenland. Undercutting is not included in our simulations because it cannot be explicitly described in 1-D model and appropriate parameterization for this processes does not exist yet. However, the large uncertainties of the calving processes are included implicitly in our parameter β . Of course, undercutting is not the only process which is still poorly understood and not included in our simulations. However we strongly disagree with the reviewer's statement that we chose our approach "*because it is the one which works for their model rather than on the basis of representation of processes*". We used a rather standard modeling approach which has been used in numerous previous studies but add one improvement – explicit treatment of submarine melt throug h turbulent plume parameterization. Our result clearly show that this is an important improvement.

*2. Tuning of model*
*In order to initialize the model to obtain glaciers close to their present day state, the authors first vary 4 key parameters with a fixed (non-plume) melt rate and find a combination of these parameters which puts the glacier close to its present day state. They then turn on the plume model and tune 2 key parameters to maintain the glacier close to present day. The resulting set of parameters is used for the future simulations.Overall, I have two main issues. First, do you know that the values of the first 4 parameters are the only values which would work? It is not at all obvious that there should be 1 unique set of parameters which work. This is important because a different set of parameters might lead to a different evolution in the future.*

The four parameters determining glaciers dynamics have been selected by minimizing errors in accurately measured ice thickness and discharge through the grounding line. Of course, the selection of a single model performance metric to some extent is subjective, but as soon as this metric is selected, the values of four parameters are uniquely defined unless completely different combinations of these four parameters produce precisely the same value of the performance metric which is extremely unlikely. Thus the answer to the the first reviewer's question is " other combinations of these parameters also will work but not as good as the set we used. Since our aim was to investigate the effect of global warming on the future glacier dynamics and mass loss, we chose to only vary two parameters - *β* and *fwd* – which control submarine melt and calving. Already variations of these two parameterrs lead to large uncertainty ranges. Introducing even more uncertain parameters would would reduce the readability of the paper.

*Second, I am uncomfortable with how much the parameters vary from glacier to glacier (by two orders of magnitude in some cases); one would hope that if the model has a good representation of the physics then the parameters should take reasonably close values between glaciers.*

In fact, most of the parameters that determine the glacier flow are in the same order of magnitude for different glaciers. The only notable exception is the high width-scaling parameter $W_s$ (enhances lateral stress ) for Store glacier and comes from the fact that Store glacier is a fast-flowing glacier situated to the top of a sill (Fig. S9), that makes this glacier unstable unless there are high lateral stresses acting to stabilize it. The great variance of the sliding factor $A_s$ between different glaciers is not surprising and has been shown recently by Stearns and van der Veen (2018, Science)

*Once more this leads me to question how robust the future projections are. For example, it might be that due to uncertainties in other aspects of the model (e.g. bed topography), you have to tune up*

*the melt rate a lot to match the glacier in the present day (high value of beta). But then presumably you are hard-wiring that glacier to be highly sensitive to ocean warming and prejudicing its future evolution. I am not convinced that this is physical rather than just an artifact of the initialization and missing model physics or poor input data.*

There are always uncertainties in model physics and input data that will influence the numerical experiments. However, we do not hardwire our glacier model to be more sensitive to submarine melting, since we always vary β from 0.3 to 3. (see Table S3).

*3. Clarity of description – notably ocean forcing*
*I found certain aspects of the description of the model and its inputs rather confusing - in general I think the paper would be much improved if the description of the model and inputs could be clarified and simplified.*

*The most confusing part for me at the moment is the fjord temperature and salinity profiles. If I understand correctly, you use either the CTD profiles or the reanalysis data for the spin-up, and then you add the CMIP5 trends on top of the spin-up period to do the future projections. If this is the case it needs to be made clearer. It also wasn't clear to me when the CTD data was used and when the reanalysis data was used. There is a long discussion of how the CTD data and reanalysis data differ, but ultimately this discussion comes to nothing because you use both anyway. This is one example of how this paper could be a lot more readable – perhaps move the detailed discussion comparing CTD and reanalysis to the supplement, allowing you to focus on describing what actually goes into the model. It would be great if you could provide an equivalent to equation 17 for the fjord temperature – this would really help the reader understand what is being done.*

We agree with the reviewer and moved the part on how the TS profiles were constructed to the supporting information. Equation 19 now describes the future ocean temperature forcing. Furthermore we now inserted a new figure (Fig. 2) that gives an overview off all the input data used for the future scenarios. A short summary of all the input data is given in section 3.:

"To simulate the response of the glacier-plume model to future global warming we considered the potential changes of surface mass balance (SMB) and submarine melting. To this end, for each glacier, we derived data sets for three forcing factors  from the year 2000 till  2100: spatially distributed SMB (section 3.3), subglacial discharge (section 3.3) and fjord water temperatures (section 3.4).
For changes in SMB we used anomalies from the simulation with the regional climate model MAR, forced by global GCM MIROC for the RCP8.5 scenario. In our previous study (Calov et al., 2018) we used the same SMB changes toforce the 3D ice-sheet model SICOPOLIS . Now we use results of this simulation to compute the subglacial discharge for each glacier from simulated surface runoff . Changes in ocean temperature were included by applying a linear warming trend, derived from several different CMIP5 models. On every time step the three forcing factors where provided as data-input and forced the glacier-plume model (Fig. 2). While for each glacier the future evolution of the subglacial discharge and ocean temperature where firmly prescribed in the data-sets, the SMB-input was interactively corrected for the surface elevation feedback and thus considered the glacier surface height on each time step. The upcoming subsections describe the choice of glaciers, how the geometry for the 1D model was derived and how the corresponding forcing factors were determined and applied."

*4. Use of CMIP models*
*The authors use 1 CMIP model for the surface mass balance (or in fact a regional climate model forced by the CMIP model) and 3 CMIP models for the ocean forcing. This disparity has been commented on by the other reviews and I am not convinced by the authors' response. As the authors themselves state in their response to previous reviews, "climate change scenarios (both in terms of GHGs concentration and model output) are the major source of uncertainties." This makes it sound like you might have reached different conclusions if you had used different CMIP models – can you be sure that your conclusions are independent of the CMIP models or that the CMIP models you have used are in some way representative of others?*

Indeed we used a single climate change scenario simulated with the regional climate model MAR forced by output of the MIROC5 model for the RCP8.5 concentration scenario, the same as we used in Calov et al. (2018). The choice of MAR-MIROC5 scenario is justified by the fact that it is medium in terms of Greenland SMB change compare to the results for other GCMs (see fig.3 in Calov et al., 2018). Of course, it would be useful to perform similar study with other climate change scenarios but

it is important to realized that generating of each scenario is extremely computationally and time-demanding procedure involving several authors of the paper. The first stage of this procedure was acquiring regional model output, interpolated it to SICOPOLIS ice sheet grid and calculating several parameters needed for elevation correction of SMB, surface air temperature and surface runoff for each year and for each model grid cell. At the second stage, the SICOPOLIS model has been run for 100 years forced by the SMB anomalies. The output of this experiment - annually averaged elevation corrected surface runoff, basal melt rate and elevation - were used during the third stage using to force the basal hydrology model HYDRO to calculate monthly subglacial discharge for each year and for each glacier. Finally, SMB and elevation corrected coefficients from the regular ice sheet model grid were interpolated to the central lines of each glacier and, together with simulated subglacial discharge, used to drive the glacier-plume model. Obviously, for each new scenario, all these stages have to be repeated. Although we performed simulations only with one climate scenario (but for a range of ocean warming scenarios), we see no reason why we would arrive to a different conclusion if we would use another GCMs. Of course, the numbers for SLR are scenario-dependent but the main conclusion of our paper that changes in submarine melt due to ocean warming and increased subglacial discharge are the dominant factor determining the contribution of Greenald glaciers to SLR and that this contribution is comparable with mass losses from the rest of GrIS are robust.

*5. Upscaling of SLR*
*I think the discussion on scaling up of sea level rise from a handful of glaciers to the whole ice sheet is very interesting and important. I wonder if this could be emphasized more in the paper by bringing supplementary figure 7 into the main paper and expanding the discussion? For a direct comparison to Nick et al. 2013, could you do the linear regression with the same 4 glaciers as in their paper?*

We agree and moved Figure S7 into the manuscript (Now, Figure 13). We also added discussion of the sensitivity of the upscaling method.
Unfortunately, the selection of glaciers from Nick et al. 2013 includes Petermann glacier, which we did not chose in our selection since we showed in Beckmann et al. (2018a) that simulated with the plume model submarine melt for Peterman glacier did not show good agreement with the obserservation, since the Coriolis force is not considered in the plume model and for very long floating tongue of Petermann glacier this is serious omission

*6. Surface mass balance*
*I am a little surprised about how small the surface mass balance contribution is without dynamics (Fig. 11, brown). According to Fettweis et al 2013, MAR forced by MIROC5 results in SLR of 9.2 cm by 2100 due to surface mass balance alone. I appreciate this is for the whole ice sheet, but your 12 glaciers probably cover ~5% of the ice sheet area and therefore I would expect a rough SMB-only SLR of 0.05*92 mm = 4.5 mm from your glaciers which is much larger than your brown shading. Why is this? Possibly I am getting confused about your separation in Fig. 11 – what is the difference between the orange and brown shading? Could you clarify this in the text as well? For example, you say "that the SMB-forcing alone derived from MAR (without the glacier's response) has an almost negligible effect on SLR (Fig. 11 b, brown curve)" – how can this be the case when MAR projects 9.2 cm of SLR for the whole ice sheet when forced by MIROC5 (Fettweis et al. 2013)?*

We doubled checked our numbers and come to the same result as before. The 9cm from Fettweis come from the whole ice sheet. Our glacier cathment area (see fig. 3) however is not evenly partitioned between accumulation zone and ablation zone with the accumulation zone absolutely dominates (Imagine a triangle, where only the tip belongs to the ablation zone and the rest of the triangle belongs to the accumulation zone). Thus a net surface mass loss is rather small since it is only controlled by the (much smaller) ablation area which is less than 5 %. If we want to compare with Fettweis here, we could probably compare the sum of the width of all the 12 glacier termini (55 km) to the length of the Greenland coast line(44 000 km). This gives us a fraction of 0.1%. which in turn would correspondd only 0.1mm SLR.

*Page 15, lines 16-18, Fig. 15 and Fig S1: Similarly, I find it hard to believe that the SMB contribution to sea level is negative for some glaciers under an RCP 8.5 scenario (e.g. Rink – Fig. 15). Looking at SMB anomalies by 2100 in MAR forced by MIROC5 (Fettweis et al. 2013, Fig. 5, bottom left panel), it certainly doesn't look like any glacier would have an increasingly positive surface mass balance and it doesn't look like there is any reason for Rink to be very different than Store (which is nearby), as is implied by Fig. 15. Can you check these numbers?*

Rink glacier has no negative contribution to SLR, it is about 0 in our experiment. We added a zero-line in the plot now and corrected in the text. It is true, that Rink and Store glacier a similar  SMB forcing (see Figure S1, equivalent to less than 0.05 mm SLR) but Fig. 15 here shows the dynamic response of the glacier to the SMB forcing and it is very different for these two glaciers. Therefore it is important where the forcing acts (close to termini ) and how e.g. the underlying bedrock forms the dynamic response of the glacier: As seen in the new Figure 6, Store glacier is located on the tip of a sill and a small negative SMB forcing at the glacier termini is sufficient to push Store glacier on the steep retrograde bed which leads its strong retreat whereas Rink glacier is rather stable. We thank the reviewer for this question, since it shows the importance of the dynamics response and we therefore put this example into the text.

> *A second comment: you say in the introduction that you neglect the effect of ice sheet boundary retreat on subglacial discharge. Do you also neglect the effect of ice sheet boundary retreat on SLR from surface mass balance? In other words, are you still summing up the surface mass balance contribution to sea level from areas where the ice sheet has retreated (e.g. the 30 km over which Jakobshavn is projected to retreat). If so, presumably you might be substantially overestimating the contribution to sea level from SMB?*

We think our sentences were maybe a bit confusing in this part about the subglacial discharge.
The phrase 'neglecting ice sheet boundary retreat' was unfortunate and referred to a pure technicality on the allocation procedure of the subglacial discharge for each glacier.
In our future scenarios when simulating subglacial discharge we accounted for changes in surface runoff, basal melt, and ice sheet retreat since it was determined by simulations with Sicopolis (Calov et al 2018). At the beginning of the simulations we determined the boundary gird cells  of the present-day ice mask that belong to each fjord and glacier. Thus the discharge out of this ice-mask cells (to which the subglacial discharge is routed) determines the discharge into the fjord. This 'routing end-points' for each glacier were held constant over future simulations . Thus the present-day ice mask was used only for the routing and allocation of the subglacial discharge.

 All our experiment of the glacier, show the dynamic response, where glacier retreat (and ice sheet retreat )is of course considered. Since this phrasing of 'neglecting the ice sheet boundary effect' lead to so much confusion we deleted it aiming for more clarity in the paper.

> *In general I was quite confused by how you are splitting up the different components of sea level - could you make this very clear (e.g. particularly the difference between the brown and orange shadings in Fig. 11)?*

The height of the brown area represents the "static SMB effect" which is computed as e cumulative integral of SMB anomaly  from MAR over a constant cathement area and constant (present-day ) elevation of all 12 glaciers . The height of theof yellow area shows the additional SLR contribution from responding glacier dynamics, namely changes in velocity caused by changes in glaciers elevation caused by SMB changes (but without effect of elevation changes on SMB). At last the height of the orange area represents an additional effect of elevation correction on SMB. Thus comparison of brown, yellow and orange areas clearly show that the main effect of SMB on glaciers mas loss occurs indirectly, through the changes in glaciers dynamics (velocities) which is not a trivial result. The red area represent adding effect of temperature change and blue area –adding effect of subglacial discharge change. Note, that this is not classical factor analysis where the effect of different factors are investigated separately. Here we add factors sequentially to illustrate the importance of all three factors – SMB, ocean temperature and subglacial discharge.
To clarify rewrote the whole part:

"When forced by comprehensive climate change scenarios (changes in SMB with the surface elevation feedback, ocean temperature T and subglacial discharge Q) the median estimate for SLR contribution from all 12 glaciers is about 18 mm (17.9 mm) at the year 2100 (Fig. 8 a, and Fig. 8 b, blue curve). To quantify the role of the individual forcing factors, we performed an additional set of simulations with the same model versions corresponding to the median SLR response (18mm) but applying the three different forcing factors in sequence. With the same model version we rerun for each glacier the experiment omitting changes in subglacial discharge (denoted "SMB + T" in Fig. 8 b, pink curve) and omitting changes in subglacial discharge and ocean temperatures (denoted "SMB" in Fig. 8 b, sum of the brown, orange and yellow areas). The total effect of SMB change on SLR is decomposed into "static" (brown), "dynamic" (yellow) and effect of elevation correction (orange). "Static" effect was computed as the cumulative integral of SMB anomalies over the fixed present-day catchment and elevation of individual glaciers. As Fig. 8b shows, this component is close to zero which is explained by the geometry of the glaciers' catchment area, where the ablation area is

much smaller than accumulation area. For some glaciers, the cumulative SMB over the glacier's catchment even increased towards the end of 21 century due to increased precipitation over accumulation area (Fig S1, supporting information). Thus most of SLR due to SMB change alone occur through the dynamical processes – thinning and acceleration of glaciers, which in turn affects the calving rate.  The surface elevation feedback (Fig. 8 b, yellow curve). has only a minor effect on the glaciers response to SMB change which is not the case for the entire GrIS (Calov et al., 2018) where this effect is important.

As explained above, we attribute 30 % of the 18mm SLR to the response to changes in SMB alone. The remaining 70 % of SLR is thus caused by the response toocean warming  and increased subglacial discharge  (Fig. 8 b, blue and pink area together). We found that both factors, ocean warming and increased subglacial discharge, are of comparable importance for SLR (by comparing the blue and pink curve in Fig. 8 b These estimates are valid only for the cumulative SLR of all 12 glaciers. Each individual glacier may respond differently to the individual forcing factors. For instance, Kong-Oscar Glacier (Fig. 9) is slightly gaining mass with the SMB forcing alone and shows a retreat by 10 km and contribution 1 mm to SLR due to ocean warming. When the increase in subglacial discharge is added to the ocean warming, the glacier retreats another 10 km and contributes additionally 2 mm to SLR.

**Minor comments**

*Page 1 line 19: you indicate that 70% of SLR is associated with a response to increased submarine melting. Could you clarify here and throughout the paper exactly what is meant by this? Is it that increased calving and submarine melting alone are accounting for 70% of SLR, or is it that increased calving and submarine melting together with decreased SMB due to dynamic thinning are accounting for 70%? This is an important distinction – the latter possibility would include a dynamics-SMB coupling whereas the former is pure dynamics.*

The reviewer's question is right to the point. In fact, we do not attribute 70% of SLR to increased submarine melting and 30% to increased surface melt. What we derived from our experiments is that changes in SMB alone (with constant ocean temperature T and subglacial discharge Q can only explain 30% of SLR simulated in the experiments where all three factors were taken into account. Therefore remaining 70 % are attributed to the glaciers response to changes in ocean temperature and Q. These two factors, assuming all other are kept constant, do cause increased submarine melt. However, in the transient experiments, there is a complex interplay between all three processes (surface and submarine melt, and calving). This is why in the revised manuscript we changed "changes in submarine melt" to "change in ocean temperature and subglacial discharge which control the submarine melt".

*Page 3 line 28: you have assessed the uncertainty for calving and melting parameters (at least for a single calving law) but in relation to climate scenarios you have only considered RCP8.5 and a single CMIP model on the SMB side – so I think you have not really quantified uncertainty related to climate change scenarios and maybe this statement should be removed.*

Agreed, we deleted the words climate scenarios and only refer to the proportion of calving and submarine melting and ocean warming.

*Page 5 line 10: 'initial boundary condition' – it would be clearer if you changed this to the 'boundary condition at the ice divide' or 'boundary condition at the top of the glacier catchment'.*

Agreed,done.

*Section 2.2: it would be good to acknowledge briefly the extent to which this plume model approach captures what is known about submarine melting. E.g. it does capture vertical variability in melt rate within a plume (Jenkins 2011), but it can't capture variability across the calving front due to presence/absence of plumes (Fried et al. 2015), and it can't capture melting outside of plumes due to melt-driven convection (Magorrian & Wells, 2016) or fjord-wide circulation (Slater et al. 2018).*

The plume actually simulate melt-driven convection if the subglacial discharge is small (Beckmann et al. 2018) but of course-as the reviewers mentions- not outside the plume. In the nature of a width-averaged 1D plume is of course it's limit in terms of  variability across the calving front. We added now: "The plume is a 1D model and therefore can neither simulate variability across the calving front (Fried et al., 2015), nor account for fjord wide circulation (Slater and Straneo, 2018) across and outside plumes. However, the width-averaged melt rates - as only required for the 1d glacier model- can be simulated with the 1D plume model (Beckmann et al., 2018)."

*Section 2.3: it might improve the readability of the paper if some of these technical details which are not central to the main messages of the paper (e.g. the definition of the total submarine melting) could be moved to supporting information – just a suggestion so feel free to ignore.*

We left this description in the main part, this this is the essential part of the glacier-plume model. And reviewers before have ask for the detailed equations e.g. the plume.

*Page 16 line 15: pvalue=0 is a bit meaningless – better to say p<0.01 or similar.*

Done.

*Page 18 line 19: I think it would be more natural to state the proportion of SLR which is attributable to dynamics (i.e. the dynamical response of Greenland's outlet glaciers can account for 5/13.8 = 36% of total sea level contribution from the Greenland ice sheet).*

This is not correct. 5 cm of SLR which we obtained by upscaling of dynamical glacier response is not a part of 13.8 cm SLR simulated in Calov et al. 2018 but additional SLR not accounted in Calov et al. (2018).

*Table 2: some of the column headings have a 'Delta' symbol in them which are not mentioned in the figure caption – are they meant to be there?*

We thank the reviewer for spotting this mistake. We deleted the Delta symbol, such that the caption entry corresponds to the table heading.

*Figure S2 – you say here you used Bedmachinev3 topography but in the main article you state you used Bedmachinev2 (page 8 line 18). Which was it?*

Yes we used Bedmhachinev2 throughout our experiments as described in the main text. We had started with our experimental setup when there wasn't the newest Bedmachinev3 available. This Figure only illustrates the sill depth determination with the best available data set (now, Fig. S8) to show the potential effect of the bathymetry on the reconstructed temperature profiles from reanalysis data (Fig. S3). Therefore we use Bedmachinev3, which contains the latest data on fjord bathymetry. We show that considering this sill (with the best available data) would shift the reanalysis temperature profile even further apart from the measurement close to the glacier and therefore does not improve the reconstruction of temperature profiles from reanalysis data sets. Thus our current method (deriving the temperature profile from the 400m-depth point) is an adequate approach. We add this part in the caption of the Figure (now Figure S8)

*Supplement – please improve figure captions throughout – at the moment they are sloppy. For example Fig S4 – annual subglacial discharge from what? Fig S5 has two missing references.*

Done.

References which are not in the paper
Fried, M. J., Catania, G. A., Bartholomaus, T. C., Duncan, D., Davis, M., Stearns, L. A., et al. (2015). Distributed subglacial discharge drives significant submarine melt at a Greenland tidewater glacier. Geophysical Research Letters, 42, 9328–9336. https://doi.org/10.1002/2015GL065806

Magorrian, S. J., & Wells, A. J. (2016). Turbulent plumes from a glacier terminus melting in a stratified ocean. Journal of Geophysical Research: Oceans, 121, 4670–4696. https://doi.org/10.1002/2015JC011160

Slater, D. A., Straneo, F., Das, S. B., Richards, C. G., Wagner, T. J. W., & Nienow, P. W. (2018). Localized plumes drive front-wide ocean melting of a Greenlandic tidewater glacier. Geophysical Research Letters, 45. https://doi.org/10.1029/2018GL080763

Stearns and Van der Veen (2018): Friction at the bed does not control fast glacier flow. Science,361. DOI: 10.1126/science.aat2217

**Reviewer 2**

Minor technical corrections:

*page 3, line 18: change "For" to "for"*

Done.

*page 7, line 15: there is a "dot" before "This"*

Since there is a comma after the equation (15), we believe the "dot" is correct, since it signifies the end of the sentence.

*page 7, line 20: there is a "dot" before "If"*

Agreed, done.

*page 8, line 14: "Goeltzer (2013) should have brackets*

Agreed, done.

*page 10/11, line 31: from 1000 to 3000 m or 1000 and 3000 m?*

[revised manuscript text omitted]

---

## Author Response (AR3)

We thank the reviewers for the constructive reviews and suggestions. The comments by the reviewers are in indented blocks and italic fonts. Our response follows each comment and changes in the manuscript are in quotation marks.

**Reviewer**

> *The authors have largely addressed my scientific concerns, and have more clearly discussed assumptions or caveats in the important places. I commend them for their efforts in revising the paper. I still think the paper could do with a thorough proof-read to catch typos and small errors as these are still rather numerous. Overall I am happy to recommend only minor/technical revisions.*
>
> *I also note that the line numbers in the author response file are a bit odd (they don't reset at the top of each page) - but hopefully it is still clear to which part my comments refer.*
>
> *Minor corrections:*
>
> *Section 2.1: it would be helpful to refer to Fig. 1 at appropriate points in this section - at the moment you don't refer to Fig. 1 until a few pages later but it is relevant here.*

Agreed, we now mention Fig. 1 in the introducing sentence of the section (p.4, line 20) and refer to it when the position of the grounding line x_gl and calving front x_cf is mentioned (p.5 line 11 and 21)

> *P7L3-5: this addition would be better somewhere around P4L15*

Agreed, we transferred that section accordingly. Nevertheless we kept an introductory sentence for this subsection describing the approach of submarine melt rate profile and thickness loss of the glacier.

> *P17L29-31: I am still a little confused about this partition between SMB and dynamics. You state here that most of the SLR due to SMB change alone occurs through dynamical processes. Presumably you mean that dynamic processes lead to thinning, which lowers the ice sheet surface and makes the SMB more negative in these regions? I would call this a surface elevation feedback. But in the next sentence you state that the surface elevation feedback has only a minor effect. Can you clarify exactly what you mean by surface elevation feedback? Does it include effects from ice dynamics or purely SMB? Better still, would it be possible to write down some equations (a bit like Eq. 17) to really make it obvious which factors you are referring to with which terminology?*

As we now described in the paper (see below) we decompose effect of SMB change on glacier's SLR contribution into "static", "dynamic" and "dynamic effect with elevation correction" components. Static effect is computed using the integral of SMB anomaly computed by the MAR model over the fixed present-day catchment and elevation of individual glaciers. Note that MAR calculates SMB under changing atmospheric conditions but considers only a constant ice-sheet surface elevation over time (no surface elevation feedback).

In the next experiments we use the same SMB forcing (not corrected for simulated elevation change) to run the glacier model. The difference in SLR contribution between this experiment and the "static" we attributed to the "dynamic response" of the glacier model ( yellow curve). Since SMB forcing is not spatially homogeneous and is more negative in the ablation zone, it leads to

thinning of the tongue of the glacier which can trigger dynamic processes like speedup, grounding line retreat and calving which, in turn, can lead to ice mass loss and thus SLR (Fig. 8, yellow curve).

As mentioned by the reviewer, a thinning glacier lowers it's ice surface and automatically is exposed to warmer temperature the can lead to more negative SMB (surface–elevation feedback). To consider this feedback the SMB input-data for the glacier model have to be corrected analogous to equation (17) on every time step.This was done in the experiment "dynamic effect with elevation correction"

The difference in SLR between the experiment with the elevation correction and the experiment without correction we attributed to the elevation correction effect. However, Fig 8. shows that SLR contribution by the glacier models forced by corrected (with surface elevation feedback) SMB forcing and not corrected (without surface mass balance -elevation feedback) SMB forcing shows only a negligible difference ( difference between orange and yellow curves). Thus, the main contribution to SLR from the glaciers thinning occurred via changes in dynamics (velocity and calving) rather than surface mass balance - elevation feedback.

We clarify this now in the text and refer to equation 17.:

*"The total effect of SMB change* on SLR is decomposed into "static" (brown), "dynamic" (yellow) and "dynamic effect with elevation correction" (orange).  "Static" effect was computed as the cumulative integral of SMB anomalies over the fixed present-day catchment and elevation of individual glaciers. As Fig. 8 b shows, this component is close to zero which is explained by the geometry of the glaciers'
catchment area, where the ablation area is much smaller than accumulation area. For some glaciers, the cumulative SMB over  the glacier's catchment even increased towards the end of 21 century due to increased precipitation over accumulation area (Fig S1, supporting information). The SMB forcing used to force the glacier model comes from the original MAR output  data (Fig. 2), where no surface-elevation feedback is considered since the ice-sheet surface is considered constant over time. In the "dynamic" (yellow) experiment we force the glacier model with the original MAR output-data (no surface-elevation feedback). Thus most of SLR due to SMB change alone occurs through the dynamical processes – thinning and acceleration of glaciers, which in turn affects the calving rate and grounding line retreat. In the third experiment (orange) we consider the surface-elevation feedback by correcting the SMB-forcing for the elevation change on every time step (eq. 18) The resulting additional SLR (Fig. 8 b, orange) is negligible compared to the dynamical response (yellow curve). Thus, the surface elevation feedback has only a minor effect on the glaciers response to SMB change which is not the case for the entire GrIS (Calov et al., 2018) where this effect is important."

> *P17L16-21: I wonder if it would make sense to swap Figs. 11 and 12? This paragraph could then come after you discuss uncertainties below, and it would make more sense to have P17L16-21 together with the last paragraph of this section (P18L9-22).*

We think the refers to page 15 and 16? However, we find it difficult to swap paragraphs and figures here. In the case of swapping, we would jump from median range scenarios to uncertainties (considering all scenarios). We prefer the story line of explaining the differences within the median-range scenarios, then all scenarios, an the uncertainties within all these scenarios.

> *P18L21: 'not statistically significant' - the p-value on Fig. 13b is 0.01 so apparently it is significant?*

*Yes, we corrected for that. For the case of the small slope (old Fig. 13a) it is however statistically not significant.*

> *Fig. 13: this is just an idea, but I think it would look nice to remove Fig. 13 and instead put the best fit lines on Fig. 11. The large spread between the lines would be a nice visual illustration of the point you are trying to make - that upscaling to the whole ice sheet is very sensitive to which glaciers you choose.*

This is true but we also want to make the point that our sample size is appropriate to consider upscaling. Bringing the slopes from Fig. 13 into Fig. 11 does not show the dependency of the choice of sample size at a first glance. Instead we leave figure 13 but only use one panel, to show the discrepancy between those two slopes.

*Fig. 6: could you add to the caption that these plots show the 'present day' or steady state after spin-up*

Done.

*Fig. 11 caption: you might as well state that the slope is 0.12 rather than 0.1 for consistency with the plot and the main text*

Done.

> *Fig. 12: the numbers on the axes are too small. The same could be said for a few other plots too - maybe check throughout?*

*We increased the font size of the axis numbers.*

> *Table 1: are these numbers rounded to the nearest degree celsius? If so, please state in the caption.*

Yes, now inserted into the caption.

*Table 2: why do the number of stable simulations in this table disagree with Table S3? E.g. Alison glacier has 54 stable simulations according to this table but 74 according to Table S3. Similar for other glaciers.*

We forgot to update this table in the SI when updating Table 2. Now the updated version of S3 is in the SI and the number of sums is correct.

*Figures 6-8: subplots need 'a, b, c...' labels*
Done

*Fig. 7 for Daugaard-Jensen: it looks like Daugaard-Jensen continues to retreat once it becomes land-terminating. Is this purely due to surface mass balance? If so, it seems quite fast compared to retreat of land-terminating glaciers in the present day?*

*Yes Daugaard Jensen does retreat once is becomes land terminating, due to a combination for SMB and dynamics. Once starting to retreat into the retrograde bed Daugaard-Jensen accelerates and thins. Fig 7 panel b shows now a bigger range of the glacier profile and it can be seen that the bedrock topography has a steep sill at around 80 km, where the glacier is buttressed, and only the*

*part of the glacier further downstream thins due to the increased velocity at the calving front. Thus a strong slope develops, increasing velocity and the strain rate, and thus resistive stress which promotes higher calving rates. Due to the steepness the glacier thins a lot and accelerates. The high velocities promote calving an the very thin glacier promotes more negative SMB, thus the glacier still retreats. It can be faster than present-day land terminating glaciers, since we are here at the end of the century under the highest emission scenario RCP8.5.*

*More generally, do you modify the freshwater depth fwd at all as the glacier retreats? Do you not end up with situations when the freshwater depth is greater than the ice thickness? And is this appropriate?*

*No we keep a constant freshwater depth, despite increasing surface runoff towards the end of the century. At the beginning of our experiment freshwater depths fwd << H.*
*Our calving condition leads to calving when the depth of the crevasse penetrates from the surface hs to sea level 0.*

$$h_s \leq d\left(R_{xx}, fwd\right)$$

*for zero resistive stress,*

$$h_s \leq fwd\, \rho_{freshwater}/\rho_{ice}$$
$$h_s \leq fwd$$

*For marine-terminating glacier the thickness H is always thicker then hs.*

$$H > h_s$$

*Thus glacier ice left is always thicker than fwd (fwd < hs << H), since thinner ice is already calved.*
*But in the case of land-terminating glacier surface elevation can be higher then the thickness for bedrock above sea level.*

$$H < h_s$$

*In this case the ice thickness can be thinner than the fresh water depth but than the calving condition is harder to reach since hs increases strongly with altitude. Thus, we might even underrepresented calving for land-terminating glacier in our experiments. However the minority of our glaciers become land-terminating and we believe that this will no effect our results.*

*Typos etc:*

*P1L24: 'response in increased' should be 'response to increased'*
*done*
*P4L17: 'marine terminated' should be 'marine terminating'*
*done*
*P5L18: U(x=0) = 0 should be in math font*
*done*
*P8L21: MIROC should be MIROC5*
*done*

*P8L22: 'toforce' should be 'to force'*
*done*
*P8L26: 'where' should be 'were'*
*done*
*P9L5: 'were used' should be 'which were used'*

*done*

*P9L11: BedMachinev2 - check capitalization*

*done*

*P9L27: 'boundary of prescribed' should be 'boundary of the prescribed'*

*done*

*P10L32: 'chapter' should be 'section'*

*done*

*P12L23: 'glacier-plum' should be 'glacier-plume'*

*done*

*P13L21: 'stand alone' should be 'standalone'*

*done*

*P14L29: 'experiment' should be 'experiments'*

*done*

*P15L9: 'examples' should be 'example'*

*done*

*P15L12: 'points on' should be 'points to'*

*done.*

*P16L7: 'glacier is depicted' should be 'glacier are depicted'*

*done.*

*P16L8: 'land-terminated' should be 'land-terminating'*

*done*

*P17L31: no full stop needed after 'yellow curve)'*

*done*

*P17L20 and P17L27: 'Jakobshaven' should be 'Jakobshavn'*

*done*

*P19L6: 'on SLR' should be 'for SLR'*

*done*

*P19L7: 'estimations' should be 'estimates' (in both cases)*

*done*

*P19L10: 'Jakbobshavn' should be 'Jakobshavn'*

*done*

*P19L13: 'account on' should be 'account for'*

*done*

*P19L20: 'over a threefold' should be 'over threefold'*

*done*

*P20L16: need a closing bracket after 168 mm*

*no, ..."from the lower sample size range" belongs within the bracket.*

*Fig. 7 caption: 'with over 80 km' should be 'of over 30 km'*

*changed too: "of over 60."*

*Fig. 12: no need for 'a' subplot label*

*done*

*Fig. 13: the label on the left plot should be 'a', not 'b'*

*done, now one Fig.*

[revised manuscript text omitted]